



# Evaluation of satellite-based aerosol datasets and the CAMS reanalysis over ocean utilizing shipborne reference observations

Jonas Witthuhn[1], Anja Hünerbein[1], and Hartwig Deneke[1]

[1]Leibniz Institute of Tropospheric Research, Leipzig, Germany

**Correspondence:** jonas.witthuhn@tropos.de

**Abstract.** Reliable reference measurements over ocean are essential for the evaluation and improvement of satellite- and model-based aerosol datasets. Within the framework of the Maritime Aerosol Network, shipborne reference datasets have been collected over the Atlantic ocean since 2004 with Microtops sun photometers. These were recently complemented by measurements with the multi-spectral shadowband radiometer GUVis-3511 during five cruises with the research vessel *Polarstern*. The AOD uncertainty estimate of both ship-borne instruments of $\pm0.02$ can be confirmed, if the GUVis instrument is cross-calibrated to the Microtops instrument to account for differences in calibration, and an empirical correction to account for the broad shadowband and the effects of forward-scattering is introduced. Based on these two datasets, a comprehensive evaluation of aerosol products from the Moderate resolution Imaging Spectroradiometer (MODIS) flown on NASA's Earth Observing System satellites, the Spinning Enhanced Visible and Infra-Red Imager (SEVIRI) onboard the geostationary Meteosat satellite, and the Copernicus Atmosphere Monitoring Service reanalysis (CAMS RA) is presented. For this purpose, focus is given to the accuracy of the aerosol optical depth (AOD) at $630\,\mathrm{nm}$ in combination with the Ångström exponent (AE), discussed in the context of the ambient aerosol type. In general, the evaluation of MODIS AOD from the official Level-2 aerosol products of C6.1 against the Microtops AOD product confirms that 76% of datapoints fall into the expected error limits given by previous validation studies. The SEVIRI-based AOD product exhibits a 25% larger scatter than the MODIS AOD products at the instrument's native spectral channels. Further, the comparison of CAMS RA and MODIS AOD versus the shipborne reference show similar performances of both datasets, with some differences arising from the assimilation and model assumptions. When considering aerosol conditions, an overestimation of AE is found for scenes dominated by desert dust for MODIS and SEVIRI products versus the shipborne reference dataset. This highlights the importance of considering aerosol type in evaluation studies for identifying problematic aspects.

## 1 Introduction

Aerosol particles directly influence the Earth's radiation budget through their interaction with solar and terrestrial radiation. The direct radiative effect of aerosol ($\mathrm{RE_{ari}}$) is defined as the change of radiative fluxes caused by aerosol particles (Boucher et al., 2013). Quantifying $\mathrm{RE_{ari}}$ on a global scale is crucial for understanding the role of aerosol in the climate system, and in particular the climate forcing by aerosol in the context of anthropogenic climate change. To quantitatively estimate $\mathrm{RE_{ari}}$ based on radiative transfer models, knowledge about the spectrally resolved optical properties of different aerosol types is essential.





The annually averaged shortwave $RE_{ari}$ over cloud-free ocean is estimated to lie in the range of -4 to -6 $\mathrm{Wm}^{-2}$, which can mainly be attributed to sea spray (Bellouin et al., 2005; Loeb and Manalo-Smith, 2005; Yu et al., 2006; Myhre et al., 2007). Besides maritime aerosol originating from sea spray, mineral dust emitted from the large desert areas (e.g. the Sahara desert) is a major contributor to both the short- and longwave $RE_{ari}$ over ocean (e.g., Tegen, 2003; Christopher and Jones, 2007; Nabat

et al., 2015).

Aerosol also influences the Earth's radiation budget indirectly through its role in cloud formation and by modulating the radiative effects of clouds through a number of possible pathways (e.g., by altering cloud albedo or cloud lifetime) (Denman et al., 2007). All indirect effects on the Earth's radiation budget due to aerosol–cloud interactions can be combined into the effective radiative forcing arising from aerosol (Boucher et al., 2013). Observations of aerosol load and optical properties with

global coverage are required to improve our understanding of climate-relevant aerosol processes, to investigate their role in anthropogenic climate change, and to improve quantitative estimates of this effective forcing.

Satellite remote sensing provides global observations of aerosol properties and the radiation budget (Chen et al., 2011; Kahn, 2012). These observations are key to quantify $RE_{ari}$, in particular over ocean, where only limited surface observations (e.g. from ship) are available (Haywood et al., 1999). Due to the sensitivity of the retrievals to factors such as instrumental

calibration and retrieval assumptions however, a critical evaluation of the accuracy of the resulting satellite datasets is essential for understanding their quality and limitations, e.g. by comparing these products with well-calibrated ground-based reference observations.

The most widely used satellite-based aerosol products are based on the Moderate resolution Imaging Spectrometer (MODIS) instrument flown on the polar-orbiting Terra and Aqua satellite platforms, which were launched in 1999 and 2002, respectively,

by the National Aeronautics and Space Administration (NASA), and continue operations to this day. These products were evaluated in numerous studies in their evolution from Collection 4 (C4) (e.g., Remer et al., 2005; Kleidman et al., 2005) to C5 and C5.1 (e.g., Levy et al., 2010; Bréon et al., 2011; Misra, 2015) and finally to C6 and C6.1 (e.g., Munchak et al., 2013; Levy et al., 2013; Livingston et al., 2014). Validation of the product quality over ocean was more limited compared to that over land, and has mostly relied on coastal or island sites with sun-photometer measurements (e.g., Abdou et al., 2005; Bréon et al., 2011;

Shi et al., 2011; Anderson et al., 2012; Wei et al., 2019). Ship or airborne reference observations were utilized less frequently (e.g., Smirnov et al., 2011; Adames et al., 2011; Schutgens et al., 2013). Levy et al. (2013) estimate an error of aerosol optical depth (AOD) over ocean within the error limits of $[+(0.04+0.1\mathrm{AOD}), -(0.02+0.1\mathrm{AOD})]$ for the C6 products. Considered in this paper are the products from both the MODIS Aqua and Terra instruments and refer to them as *MxD04_3K* and *MxD04_L2* for the high resolution 3km Remer et al. (2013) and lower resolution 10km Levy et al. (2015) swath products, respectively.

In addition to the widely used aerosol datasets available from the MODIS instruments, datasets based on geostationary satellite observations are of high potential interest for scientific applications. In particular, their high temporal resolution combined with their fixed field of view on the earth enables studies of the diurnal cycle and the temporal evolution of aerosol plumes. Hence, the aerosol product of Thieuleux et al. (2005), which is based on the Spinning Enhanced Visible and Infra-Red Imager (SEVIRI) onboard the geostationary Meteosat second generation (MSG) satellites operated by the European Organization for

the Exploitation of Meteorological Satellites (EUMETSAT), is also taken into consideration in this evaluation. It is available





at a temporal resolution of 15 min. Compared to the MODIS aerosol products, some limitations arise from the instrumental characteristics of the SEVIRI instrument, and thus have to be taken into account: the spatial resolution of SEVIRI is 3 km in nadir versus 1 km for MODIS, and only two spectral channels (630 and 810 nm) are utilized in the retrieval. A smaller set of 12 aerosol models is used as basis for the retrieval, and the product has received far less validation efforts (e.g., Bréon et al., 2011; Bernard et al., 2011). To our knowledge, it has not been validated previously with shipborne observations.

For many research purposes, aerosol properties from model-based reanalysis datasets are a promising alternative to the direct use of satellite-based aerosol products. In contrast to satellite products, aerosol properties from a reanalysis are available independent of cloud cover and satellite overpass time. The Copernicus Atmosphere Monitoring Service reanalysis (CAMS RA) is the latest global reanalysis of atmospheric composition produced by the European Centre for Medium-Range Weather Forecasts (ECMWF) and provides global information on aerosol optical properties. It relies on the data assimilation of satellite observations into ECMWF's Integrated Forcasting System (Inness et al., 2019). In the case of aerosol, it has to be realized that MODIS datasets are assimilated into CAMS RA, so that differences between both datasets are expected to be relatively small and will mainly show the influence of model assumptions and the assimilation system of the CAMS system.

In this study, two independent datasets of shipborne aerosol products are compared and used for an evaluation of both satellite products and the CAMS RA over ocean, with an additional focus on aerosol type. There is still a lack of shipborne spectral radiation measurements for this purpose (Brando et al., 2016). Furthermore, by separating the evaluation according to aerosol type, more insights can be gained into the limitations of the current satellite products. Also, further validation of the CAMS RA aerosol products with respect to aerosol type is needed (Inness et al., 2019). While the optical properties of maritime aerosol are considered to be relatively well understood, the optical properties of mineral dust are still the topic of ongoing research due to their complex, non-spheric shape (Dubovik et al., 2006; Mishchenko et al., 1999), which introduces significant uncertainty in their optical properties and remote sensing.

Compared to observations on land, shipborne observations are more challenging due to the continuously moving nature of the observational platform caused by waves. Observations of aerosol optical properties were established within the framework of the Maritime Aerosol Network (MAN) as a sub-project of the Aerosol Robotic Network (AERONET), based on the sunphotometer technique. Global observations from MAN are available since 2004, and utilize the hand-held Microtops II sunphotometers (referred to as Microtops in the following text). It thus relies on the skill of human observers to compensate for the ship movement (Smirnov et al., 2009). An automatic approach to derive aerosol optical properties over ocean using the shadowband radiometer technique was established within the framework of the OCEANET project (Macke, 2009). The shadowband radiometer GUVis-3511 (referred to as GUVis in the following) built by Biospherical Instruments Inc. was operating alongside other OCEANET instruments to provide observations during five Atlantic transit cruises of the German research vessel Polarstern since 2014 (Witthuhn et al., 2017).

Observations from both the GUVis and Microtops instruments on a number of Polarstern ship cruises over the Atlantic ocean are utilized in this study. The GUVis aerosol product has received a substantial update since the version presented in Witthuhn et al. (2017). The improvements are briefly discussed in the appendix section (Sect. A). The comparison of these shipborne





datasets to aerosol products from MODIS C6.1 and SEVIRI as well as the CAMS RA aerosol datasets is presented here, which were collocated to the ship's position along these cruises.

This paper has three principal goals:

1. Inter-comparison of both shipborne aerosol products in terms of their accuracy, with a particular focus on the verification of the uncertainty estimate of the GUVis dataset, and the usability of both datasets for the validation of satellite retrievals.

2. Evaluation of the satellite aerosol products from SEVIRI and MODIS over ocean using these shipborne datasets. A specific question is whether SEVIRI can offer additional information due to the diurnal cycle and temporal evolution of aerosol.

3. Evaluation of the CAMS RA as an alternative source of aerosol information to MODIS and SEVIRI for research purposes.

The two shipborne datasets serve as reference for the subsequent validation study. Since they are based on different techniques, an inter-comparison is presented first to point out their individual strengths and weaknesses. In this context, focus is given in particular to their suitability for satellite validation.

Within the second and third point, the estimated error limits proposed previously for the MODIS AOD products are investigated compared to the deviations found in this study. These findings are put into context to the results found for the

SEVIRI aerosol product, to observe how the limitations of the SEVIRI sensor influence the retrieval accuracy. Further, the benefit resulting from the increased time resolution of SEVIRI is investigated. Besides the accuracy of the AOD, the estimate of Ångström exponent (AE) is investigated, in particular in the context of characterizing the aerosol type. Both AOD and AE from the CAMS RA are compared to the satellite and shipborne datasets to identify differences due to the satellite retrievals, and to evaluate its performance during different aerosol situations.

The paper is structured as follows: First shipborne instrumentation and reference datasets are introduced (Sect. 2.1). A description of the satellite products and the CAMS RA are shown in Sect. 2.2 and Sect. 2.3. The methods utilized for aerosol classification, satellite data collocation and statistical measures for evaluation as well as the GUVis cross-calibration and aerosol forward scattering correction are reported in Sect. 3. The inter-comparison of the shipborne data and the comparison of the satellite products versus the shipborne reference is given in Sect. 4. Finally,the evaluation results are discussed in the con-

clusions and outlook sections (Sect. 5 and Sect. 6). In the appendix section (Sect. A) the update of GUVis irradiance processing algorithm is described.

## 2   Instruments and datasets

This section gives an overview of the shipborne instruments and reference datasets (Sect. 2.1) as well as the satellite (Sect. 2.2) and model reanalysis dataset (Sect. 2.3). All datasets are publically available, see the section on data availability at the end of

the article.





In this study, focus is given to the aerosol optical depth (AOD) and Ångström exponent (AE), the latter quantifying the dependency on wavelength $\lambda$ of the former quantity. Specifically, the AOD at $\lambda = 440\,\mathrm{nm}$ (for inter-comparison of the shipborne datasets) and at $\lambda = 630\,\mathrm{nm}$ (for comparison of shipborne and satellite data) are mainly considered here, while the AE $\alpha$ is calculated from the AOD $\tau_A$ at $\lambda_1 = 440$ nm and $\lambda_2 = 870\,\mathrm{nm}$ based on the Ångström relation as follows, unless otherwise noted:

$$\frac{\tau_{A,\lambda_1}}{\tau_{A,\lambda_2}} = \left(\frac{\lambda_1}{\lambda_2}\right)^{-\alpha}. \tag{1}$$

### 2.1 Shipborne instruments and datasets

Two aerosol datasets based on shipborne observations are considered here as ground-based reference: on the sunphotometer Microtops (Microtops II manufactured by Solar Light Inc.), and the shadowband radiometer GUVis (GUVis-3511 plus BioSHADE accessory manufactured by Biospherical Instruments Inc.).

Both instruments are well-suited for operation on moving platforms such as ships. Their measurement principles however are rather different. The Microtops is a sunphotometer, which has to be pointed manually at the sun. It measures the incident direct normal solar irradiance with a field of view of $2.5°$ (Porter et al., 2001). The shadowband radiometer utilizes an entrance optic with a global field of view combined with a shadowband that performs a $180°$ sweep, while the global irradiance is measured at a high temporal frequency of $10\,\mathrm{Hz}$.

While the global irradiance is observed with the shadowband in its low position between sweeps, the shadowband blocks a fraction of the incoming diffuse irradiance during its rotation, and will occlude the direct irradiance at a specific angle determined by instrument orientation and sun position. From the irradiance time series measured during the sweep, the global, diffuse and direct irradiance components can be inferred (Witthuhn et al., 2017).

Prior to the processing of the GUVis sweeps, the measured irradiance data has to be corrected, to compensate for the motion of the ship and the imperfect cosine response of the instrument. The actual cosine response of the entrance optic is measured by the manufacturer during lab calibrations, and can be corrected by applying correction factors depending on the spectral channel and sun elevation, if the orientation angles of the ship are known. The motion correction utilizes the method of Boers et al. (1998) based on the ship motion angles to correct the direct and diffuse irradiance components. The GUVis instrument has been calibrated in a laboratory at regular two-year intervals using a $1000\,\mathrm{W}$ FEL standard calibration lamp as absolute reference. The correction and processing of GUVis irradiance data as well as the calculation of AOD is described in detail in Witthuhn et al. (2017).

Given the direct normal irradiance obtained from both instruments and a given spectral band, the AOD can be calculated using the well-known Lambert-Beer law, and by subtracting optical depth contributions from Rayleigh scattering and gas absorption.

The concept of the "field of view" of a sun photometer is not directly applicable to a shadowband radiometer. Instead, there is the "shading angle" as described in Witthuhn et al. (2017), which is the minimum angle between the edges of the shadowband as viewed from the center of the global entrance optic. For the GUVis, the shading angle is about $15°$ (depending on shadowband position), and thus relatively large in comparison to the Microtops field of view.



The wide angle of the shadowband of the GUVis causes an underestimation of AOD caused by the influence of the forward scattering of the aerosol (Russell, 2004). The GUVis processing algorithm has received a substantial update (see Sect. A) to compensate at least partially for this effect. The reduction of measured irradiance during the shadowband sweeps is stronger in situations with increased aerosol forward scattering. Besides some other refinements, an offset was introduced for estimation of the blocked diffuse irradiance as part of the processing algorithm update in order to compensate for this effect (see Sect. A).

The technical specifications of the Microtops and GUVis instruments are summarized in Table 1. The configurations of both instruments allow a direct comparison of all spectral channels of the Microtops versus corresponding GUVis observations.

In the following, an overview of the shipborne datasets based on both instruments is presented:

(i) As the first dataset, all observations conducted during numerous cruises with the Microtops II sunphotometer in the framework of AERONET MAN since 2004 in the area of the Atlantic ocean are used, holding a total number of 19250 valid

data points. This dataset (referred to as MIC in the following text) also provides the diversity needed to investigate aerosol type–related effects for the evaluation of satellite products, and for the comparison with CAMS RA.

(ii) The second reference dataset (GUVis) is based on the GUVis shadowband radiometer. Observations with the GUVis were conducted within the framework of OCEANET (Macke, 2009) during Atlantic transect cruises with the German research vessel *Polarstern* operated by the Alfred-Wegener Institute since 2014. Until now, five cruises including the shadowband radiometer

observations have been performed, namely PS83, PS95, PS98, PS102 and PS113. A Microtops instrument from MAN has also been operated in parallel on all these cruises. This offers the opportunity to directly compare both datasets. A direct comparison of Microtops and GUVis AOD product has already been presented for PS83 in Witthuhn et al. (2017). For both shipborne AOD datasets, the uncertainty is estimated to be within ±0.02 (Smirnov et al., 2009; Witthuhn et al., 2017). Following the same procedure, this comparison is extended to all available cruises, with some minor changes to obtain more meaningful results.

The total number of valid GUVis observations is 10412.

In order to improve the agreement of the aerosol products of both instruments to acceptable limits, it has been found necessary to introduce a cross-calibration to the MIC instrument, and an empirical correction for aerosol forward-scattering, to account for differences arising from the limited accuracy of lab-based instrumental calibration, and the broad shadowband of the GUVis instrument. The correction is done fitting a linear regression curve (Eq. (10)) to the GUVis AOD (see Sect 3.4), sim-

ilar to the approaches adopted by (di Sarra et al., 2015) and (Wood et al., 2017). This enhanced dataset is denoted as GUVisE in this study.

(iii) The enhanced GUVis dataset (GUVisE) is combined with the Microtops dataset to obtain a merged surface product, to test whether the combination can lead to further improvements in accuracy. For this purpose, the mean of all GUVisE and Microtops AOD retrievals which are not flagged as outliers is calculated. This combined surface dataset (COMB) serves as the

third reference dataset for the evaluation of the satellite products. As shown in Table 2, the total amount of data points decreases to 1006 due to the combination.



## 2.2 Satellite aerosol products

Satellite based aerosol datasets over ocean considered here are obtained from both the MODIS and SEVIRI satellite instruments. The MODIS Collection 6.1 (C6.1) level-2 aerosol products *MxD04_L2* (Levy et al., 2015) and *MxD04_3K* (Remer et al., 2013) are used from both the Terra and Aqua satellites. The AOD(500 nm) and AE obtained from the SEVIRI instrument onboard the Meteosat Second Generation satellite introduced by Thieuleux et al. (2005) is also considered.

Both aerosol retrievals are based on the inversion of the measured reflectance at top of atmosphere to estimate the AOD at the instrumental spectral channels, using lookup tables of radiative transfer calculations. The accuracy of these estimates critically depends on realistic assumptions about the optical properties of aerosols assumed in the calculations. A larger number of channels enables a more accurate choice of aerosol type used by the retrieval, and is thus expected to increase the overall accuracy. In addition, factors such as the spatial resolution of the sensor, the viewing geometry, sensor calibration, as well as the accuracy of cloud screening will influence the overall accuracy. While the SEVIRI retrieval is based on only two wavelengths (630 nm and 810 nm) (Thieuleux et al., 2005), the MODIS retrieval utilizes seven spectral channels. In addition, it is continuously monitored with ground-based observations at AERONET stations (Levy et al., 2015). Therefore, it is expected that the AOD and AE obtained from MODIS will have a better accuracy than from SEVIRI when compared to the shipborne AOD and AE datasets. Further, a degraded accuracy for these aerosol properties in the presence of desert dust in both satellite products is expected, since dust particles are nonspherical, contrary to the assumption of sphericity made in both retrievals.

Besides the retrieval differences, MODIS and SEVIRI products are also different due to their satellite platform characteristics. MODIS is operated on both the Terra and Aqua satellites, which fly in a polar orbit. For studies targeting aerosol properties at a specific location, MODIS observations are only available for the two overpasses during daylight, compared to SEVIRI with a time resolution of 15 minutes. On the other hand, the geostationary orbit of MSG leads to lower spatial resolution of nadir 3 km for SEVIRI versus a 1 km nadir resolution of MODIS. In order to avoid cloud contamination in the aerosol product, the MODIS retrievals consider multiple pixels together with a strict cloud mask, leading to a decrease of the spatial resolution to 3 km for the high resolution aerosol product (*MxD04_3K*), and to 10 km for the standard aerosol product (*MxD04_L2*).

## 2.3 CAMS RA aerosol product

CAMS RA is the latest global reanalysis dataset of atmospheric conditions produced by ECMWF (Inness et al., 2019). Amongst other atmospheric constituents, it contains the spectral AOD at a temporal resolution of 3 h on a global grid of 0.7° (corresponding to a T255 spectral resolution). The advantage of utilizing CAMS RA over satellite observations is the availability of aerosol properties independent of factors such as cloud coverage or satellite orbit.

CAMS RA was developed based on the experiences gained with the former Monitoring Atmospheric Composition and Climate (MACC) reanalysis and the CAMS interim analysis (Inness et al., 2019). It relies on the assimilation of global observational datasets into the Integrated Forecast System (IFS) from various satellites to provide a global picture. In terms of aerosol properties, the AOD products of the MODIS C6 from both Terra and Aqua are assimilated. Before its failure in March 2012, retrievals from the Advanced Along-Track Scanning Radiometer (AATSR; Popp et al. (2016)) flown aboard the Envisat





mission were also being assimilated. The influence of this additional source of information for data assimilation on the accuracy is investigated in Sec. 4.3. Currently, the dataset covers the period 2003-2016, and will be extended in the following years. For the evaluation of the CAMS RA aerosol dataset an accuracy close to MODIS aerosol product is expected.

A first validation presented within Inness et al. (2019) emphasizes the high quality of AOD in the CAMS RA system, judged by a comparison to AERONET stations around the world. However, an overestimate of AE was shown during desert dust
events, and was attributed to problems in realistically representing the fine- and coarse-mode fractions for dust particles in the aerosol formulation of CAMS RA. Further evaluation with a focus on individual aerosol components as well as aerosol properties over ocean has been recommended (Inness et al., 2019).

## 3 Methods

This section gives an overview of the methods used for aerosol classification (Sect 3.1), for collocation of satellite and shipborne
measurements (Sect 3.2), and presents the statistical measures used for evaluation (Sect 3.3), as well as the correction approach adopted for the GUVis aerosol product for better comparability to MIC AOD (Sect 3.4).

### 3.1 Aerosol classification

Our study aims to compare shipborne and satellite AOD products also with respect to the role of aerosol types. A satellite–independent aerosol classification is applied, which is based on the empirical method presented in Toledano et al. (2007) for
Cimel instruments from AERONET. This method is also applicable to the Microtops AOD product, as it contains all required parameters. The aerosol classification is done by comparing the AOD at $\lambda = 440$ nm with the AE calculated based on the 440 and 870 nm channels (Eq.(1)). The pair of AOD and AE values is checked against empirical thresholds to identify the aerosol type as being one of maritime, desert dust, continental, biomass burning or mixed type. For example, AOD(440 nm) < 0.2 and AE between zero and two will identify the data point as maritime type (Toledano et al., 2007).
All results shown in this study separated by aerosol type (maritime, desert dust, continental, biomass burning, mixed) are based on this aerosol classification method. It should be noted that the shipborne observations at wavelengths of 440 and 870 nm are utilized for classification, even if figures and tables present AOD and AE for different channels (e.g., Fig. 7).

### 3.2 Collocation criteria

As common practice for spatiotemporal collocation with MODIS, a window size of 50x50 km and a time window of one hour is
recommended by Ichoku (2002) for sunphotometer observations. For the MODIS C6 validation by Levy et al. (2013), a spatial radius of 25 km and a temporal window of ±30 min has been used. Both the *MxD04_L2* and *MxD04_3K* products have been validated using a window of 5x5 pixel by Munchak et al. (2013), resulting in different window sizes of 50 km² and 15 km², respectively. The following collocation technique is utilized here to find the appropriate pixel of the satellite dataset, and to compare it to the shipborne data obtained at a certain position. First, eligible satellite images are selected using a time-frame of
±30 min around observations, and checking if the ship position is located within the field of view of the satellite image. Then,





the distance angles of all pixel coordinates to the ship position has been calculated. The satellite AOD is finally calculated as the median of all non–cloudy pixel values with a distance angle equal and below $0.2°$.

Choosing a distance angle threshold of $0.2°$ for the collocation of all satellite and model datasets to the shipborne observation assures that the same area around the reference observation is chosen regardless of satellite or model product, spatial resolution, and projection, and ensures comparability of results. This threshold results in a spatial radius of about $22\,\text{km}$.

Applying the collocation strategy introduced above to the 19250 MIC data points results in a total number of remaining 1517 data pairs for *MxD04_L2*, 1448 for *MxD04_3K*, 10061 for SEVIRI and 2474 for CAMS RA, as shown in Table 2.

After collocation with the GUVis dataset, consisting of a total number of 10412 data points, the resulting number of data pairs is 147 for *MxD04_L2*, 210 for *MxD04_3K*, and 1126 for SEVIRI. The collocation with CAMS RA results in 141 data pairs. The number of collocated data pairs is rather small, limiting the statistical significance of the comparison results.

The number of data pairs per aerosol type classified based on the shipborne reference data as described in Sect. 3.1 is given in Table 2. Since the observations are performed across the Atlantic ocean, the dominant aerosol conditions are mainly maritime or desert dust originating from the Sahara desert (see Fig. 1 and Table 2).

### 3.3 Limit of agreement method

To assess the agreement of two measures (X,Y) of the same quantity such as AOD from Microtops versus GUVis, or the shipborne dataset versus satellite products, linear regression statistics and the Pearson product-moment correlation coefficient R (referred to simply as the correlation in the following text) are calculated. Further, the analysis are extended with the so-called "limits of agreement" (LOA) method first introduced by Bland and Altman (1986). This method considers the mean of the differences of both quantities X - Y (i.e. the bias), and the LOA defined as the 95% confidence interval for those differences as additional parameters. As not stated otherwise, Y denotes the reference dataset for comparisons presented in this study.

For the evaluation of the uncertainty estimates for the shipborne observations, the method of Knobelspiesse et al. (2019) is adopted, weighting the difference X - Y (D) with their uncertainty estimate ($\sigma_X$,$\sigma_Y$):

$$(X - Y)\,/\,\sqrt{\sigma_X^2 + \sigma_Y^2}. \tag{2}$$

Thus, utilizing the LOA method together with the weighted difference, the uncertainty estimate can be confirmed if the uncertainty–weighted difference lies within the range of $\pm1.96$ for the 95% confidence interval (see Fig. 3). The percentage of outliers exceeding the limits of $\pm1.96$ is used as quantitative measure for the validation.

For the evaluation of the satellite products and CAMS RA, the bias and LOA (95% confidence interval) are used as a measure for the agreement to the shipborne reference datasets. Additionally, *Gfrac* defined as the percentage of data lying within expected error (EE) limits is calculated, in order to be consistent with other validation studies (e.g., Bréon et al., 2011). Expectations of the error are met, if 67% of data points of the satellite or model product fall into the EE range compared to the shipborne reference (Levy et al., 2013). Two EE limits are chosen here, originally presented for the MODIS aerosol product based on former validation studies e.g. by Abdou et al. (2005); Remer et al. (2008) and Livingston et al. (2014):

$$EE1 = \pm(0.03 + 0.05\text{AOD}) \tag{3}$$





and more recently in Levy et al. (2013):

$$\text{EE2} = [+(0.04 + 0.1\,\text{AOD}), -(0.02 + 0.1\,\text{AOD})]$$ (4)

EE1 is a general measure of agreement, since the boundaries are equally distributed around the reference dataset. EE2 has been specialized for the MODIS aerosol product, since a known overestimation is considered via different intercepts.

### 3.4  Cross-calibration and empirical correction of AOD

The relatively large differences originally observed in the comparison between Microtops and GUVis (Sect. 4.1), and their
changes from one cruise to another, lead to the hypothesis that the lamp-based calibration of the GUVis instrument might introduce significant uncertainties and be responsible for the differences, given the importance of calibration for the AOD accuracy (see Alexandrov et al., 2002; Witthuhn et al., 2017).

Despite the fact that the deviation of AOD between Microtops and GUVis due to forward scattering effects of aerosol is partially compensated by the processing update of GUVis (see Sect. A), a remaining linear dependence of the bias has been
observed (Sect. 4.1), which can most likely be attributed to the wide shadowband of the GUVis instrument, and the resulting difference in the field of view of both instruments. If AOD increases, this effect increases due to the enhanced circum-solar radiation. Although this effect does not have a major impact on the correlation in the direct comparison of Microtops and GUVis AOD datasets of this study (see Sect. 4.1), it introduces a substantial relative bias, and needs to be compensated for consistency of both shipborne datasets, and for the comparison to the satellite and model datasets. The compensation is done
using a linear scaling factor for measured AOD ($S$), as is explained later in this section.

To improve the consistency of the GUVis and MIC datasets, the following approach has been adopted to both transfer the calibration from the MIC instrument to the GUVis instrument, and to empirically correct for the effects of forward scattering. The first correction is accomplished following the method introduced by Alexandrov et al. (2002) for the Multi-Filter Rotating Shadowband Radiometer (MFRSR). The spectral direct irradiance measured by the GUVis can be represented by the equation:

$$I_i = C_i\,I_i^0\,\exp\left(-\frac{\tau_i}{\mu_0}\right),$$ (5)

where $I_i^0$ and $I_i$ are the spectral direct irradiance at top of atmosphere and surface, respectively, for a spectral channel $i$. The inverse of the airmass is denoted by $\mu_0$, the cosine of the solar zenith angle. $\tau_i$ is the atmospheric column extinction optical depth for a spectral channel $i$. Following Alexandrov et al. (2002), a correction factor $C_i$ for the calibration is introduced.
The absolute calibration of GUVis spectral channels is carried out in the laboratory to obtain the channel-specific calibration factors ($k_i$, [$\text{V W}^{-1}\text{m}^2\text{nm}^1$]) for the conversion of the measured voltage ($V_i$) to spectral irradiance ($I_i$):

$$I_i = \frac{V_i}{k_i}.$$ (6)

The relation of the calibration factor $k_i$ and the correction $C_i$ can be obtained from Eq.(5) as:

$$ks_i = k_i\,C_i,$$ (7)





where $ks_i$ denotes a corrected calibration factor.

    $\tau_i$ can be expressed as the sum of AOD $\tau_{A,i}$, and remaining contributions to the atmospheric optical depth ($\tilde{\tau}_i$) from Rayleigh scattering and gaseous absorption as:

$$\tau_i = \tau_{A,i} + \tilde{\tau}_i. \tag{8}$$

The AOD can now be obtained from Eq.(5) and Eq.(8) as:

$\tau_{A,i} = -\mu_0 \ln\left(\dfrac{I_i}{I_i^0}\right) + \mu_0 \ln(C_i) - \tilde{\tau}_i.$ $\hspace{4cm}$ (9)

This equation shows that the calibration correction factor $C_i$ introduces a change in AOD which is proportional to the product of the cosine of the solar zenith angle, and the logarithm of the correction factor. Introducing also a linear scaling factor $S_i$ for the AOD to account for the effects of aerosol forward scattering (see Wood et al., 2017; di Sarra et al., 2015), the following correction equation is used here in a bilinear fit, using $\mu_0$ and the GUVis-based AOD $\tau_{\mathrm{GUV},A,i}$ as dependent variables, and the

MIC-based AOD $\tau_{\mathrm{MIC},A,i}$ as independent variable:

$$\tau_{\mathrm{MIC},A,i} = \mu_0\, c_i + S_i\, \tau_{\mathrm{GUV},A,i}. \tag{10}$$

    Thus, the scaling factor $S_i$ and the calibration correction factor $C_i = \exp(c_i)$ can be obtained simultaneously from this bi linear fit.

    In the approach adopted for this study, the factor $C_i$ has been determined independently for each of the five *Polarstern* cruises

(PS83, PS95, PS98, PS102, PS113), in order to account for potential temporal changes in calibration between the different ship cruises, while a single constant value is assumed for $S_i$. The correction factors obtained by multi-linear regression based on Eq. (10) $(C_{ij}, S_i)$ are listed in Table 3. Excluding individual cruises from the regression has been found to cause only negligible influence on the remaining coefficients, confirming the stability of this correction approach. In addition, adding either a constant or quadratic correction term such as used by di Sarra et al. (2015) does not lead to a significantly improved fit quality, and has

thus not been used.

    The final procedure adopted here for the correction of GUVis AOD is done in the following steps:

(i) First, the closest GUVis and MIC data points regarding time of measurement are selected for comparison within a time frame of 30 min.

(ii) If the deviation of the AOD pair exceeds the uncertainty estimate of $\pm 0.02$ of both instruments, the data pair is flagged

as an outlier.

(iii) The fit coefficients $(C_{ij}, S_i)$ are calculated based on Eq. (10) from the GUVis and MIC AOD without considering outliers. In this fit, multiple values of $C_{ij}$ are obtained for separated cruises $j$, whereas a single value of $S_i$ is assumed for all data.

(iv) Based on both correction coefficients, a corrected AOD is calculated from the GUVis measurements.

The cross-calibrated and scaled dataset is denoted in the following text as *enhanced* dataset GUVisE.





# 4 Results and discussion

This section presents and discusses the results of this study. First, the shipborne reference datasets are compared (Sect. 4.1). Second, the satellite aerosol products are evaluated against the shipborne reference datasets (Sect. 4.2). Lastly, the evaluation of the CAMS RA aerosol data is presented in Sect. 4.3.

## 355 4.1 Shipborne datasets comparison

An evaluation of the AOD product of the GUVis shadowband radiometer compared to the Microtops sunphotometer as reference was previously described by Witthuhn et al. (2017), considering one cruise of the research vessel (RV) *Polarstern* (PS83). This study extends the comparison to include four additional cruises with the RV *Polarstern* (comprising PS83, PS95, PS98, PS102 and PS113) (see Fig. 2). Regarding the comparability of both datasets, certain shortcomings are expected, as already

mentioned. (i) Since the radiometers of both instruments utilize different calibration methods, and the spectral response of comparable channels might slightly differ, a deviation due to calibration is expected. (ii) Due to the different measurement methods of the sunphotometer and shadowband radiometer, and in particular the wide shadowband and the resulting differences in the field of view, an underestimation of AOD is expected for the GUVis instrument, related to forward scattering of aerosols (Russell, 2004).

Given the importance of calibration for the AOD accuracy of the GUVis (see Witthuhn et al., 2017), only the calibration difference of both instruments is corrected for first, based on the method presented in Alexandrov et al. (2002) (see Sect. 3.4). The correction factor $C_{ij}$ for each spectral channel $i$ and each cruise of RV *Polarstern* $j$ is given in Table 3. The Microtops observations can serve as a reliable reference for calibration, given that the lamp-based calibration of the GUVis instrument might introduce significant uncertainties and the Microtops calibration is considered as consistent and trustworthy, due to

traceability to the mature AERONET retrieval and calibration process. In the following, all versions of the GUVis datasets are calibration–corrected towards the Microtops by the method presented in Sect. 3.4. The correction of AOD with the linear scaling factor $S$ is only applied to GUVisE.

  The extended comparison is presented in the top part of Table 4. The GUVis irradiance data is first processed with the original algorithm used in Witthuhn et al. (2017). The correlation (R>0.95) found for all spectral channels comparing GUVis

(old processing) and MIC generally confirms the findings of Witthuhn et al. (2017). However, the goal of an outlier ratio below 5% (see Table 4) as well as the weighted LOA within ±1.96 to verify the uncertainty estimate of ±0.02 for GUVis is missed with the old processing algorithm. As expected, an underestimation of AOD measured by the GUVis is reflected in the negative bias of -0.02. Since the observations are performed over ocean, the dominant aerosol conditions are maritime or desert dust from the Sahara desert (see Fig. 1 and Table 2), which significantly differ in their forward scattering behaviour. Comparing

sunphotometers with a narrow field of view to measurements with shadowband radiometers with a wide shading angle the influence of the forward scattering of the aerosol causes an underestimation of AOD of the shadowband radiometer (Russell, 2004). This has previously been confirmed by di Sarra et al. (2015) for the MFRSR as well as for the autonomous marine hyperspectral radiometers presented by Wood et al. (2017).





The GUVis processing algorithm has received an substantial update to improve the data quality and to compensate for the underestimation of aerosol forward scattering. This update is described in detail in the appendix section (Sect. A). The GUVis AOD data of all cruises with the RV *Polarstern* have been reprocessed with the new algorithm, and the resulting improvement of the measured AOD compared to the Microtops is also shown in Table 4. The correlation of GUVis AOD compared to MIC increases from >0.954 to >0.988 for all channels, indicating that any non–linear deviations due to aerosol forward scattering and other effects (di Sarra et al., 2015) have been substantially reduced. The underestimation of AOD is still present, indicated by a negative bias of -0.02. The uncertainty estimate of ±0.02 can be verified for spectral channels with wavelengths larger than 500 nm, since the statistics show a weighted LOA within ±1.96 (see Sect. 3.3). The uncertainty of GUVis AOD increases with decreasing wavelengths, indicated by the increase of outlier percentage and LOA.

The difference (D) of GUVis and MIC AOD shows a higher linear correlation as |R(D)| increases from >0.4 to >0.6 going from the old to new processing. As also shown by Fig. 3 panel (a), the underestimation of GUVis AOD increases linearly with increasing AOD. This linear dependence is here attributed to the difference of field of view of both instruments. If AOD increases, the effect of the circum-solar radiation due to differences in the field of view will increase. Since GUVis utilizes a broad shadowband resulting in a shading angle of 12° to 15° compared to the field of view of Microtops of 2.5°, this effect results in an underestimation of AOD for the GUVis radiometer.

The lower part of Table 4 shows the results of the comparison of the enhanced GUVis dataset GUVisE and MIC, which includes both the calibration and forward-scattering corrections. The expected uncertainty of ±0.02 is verified again, as the values of the weighted LOA are all within ±1.96 (see Sect. 3.3). This is also shown in Fig. 3 panel (b), where the LOA falls within the uncertainty limits. In addition, the outlier percentage is close to zero, indicating a close agreement of MIC and GUVisE. The correlation increases to >0.992 for all comparable channels. The MIC and GUVisE datasets are thus consistent, due to their strong linear correlation, and their agreement within the individual uncertainty limits. Therefore, the datasets are used as reliable ground–based reference datasets in the following.

## 4.2 Satellite product evaluation

The MODIS aerosol products have been extensively validated in a large number of previous studies (e.g., Abdou et al., 2005; Bréon et al., 2011; Shi et al., 2011; Anderson et al., 2012). In contrast to most of these studies, the satellite observations are here compared over ocean to collocated shipborne observations of the Microtops sunphotometer. Figure 4 visualizes the comparison, showing results grouped in bins of 0.1 in AOD, and for an AOD at 550 nm. The validation with respect to the EE2 limits shows that the MODIS aerosol product meets the goals set by the 67% confidence interval compared to the Microtops dataset. As expected, the SEVIRI aerosol product shows a higher deviation versus Microtops than MODIS, as the SEVIRI product utilizes a less complex scheme of aerosol models and only two spectral channels. The EE2 limits have been adjusted to account for a general overestimation of 0.02 in AOD by the MODIS product, which is also observed here. The SEVIRI AOD shows an even stronger tendency to overestimate AOD in comparison to the MIC reference dataset.

Comparing the MODIS- and SEVIRI-based AOD values to the GUVisE dataset, an overestimation of satellite-based AOD versus GUVisE is observed. Fig. 4 panel (a) shows a positive bias of satellite AOD for an AOD value of 0.3, turning towards a





negative bias for higher AOD. This could be an artifact of the correction method for the GUVisE dataset, but a similar behavior also appears in the comparison to Microtops (panel (c)), although it is far less pronounced. This suggests that the reasons for

this behavior lies on the satellite side, and might be an artifact arising from the utilization of different aerosol models. While the Microtops dataset is rather diverse in observed aerosol conditions, the GUVisE dataset contains mostly maritime and desert dust situations. This is one potential explanation why the non-linearity is more pronounced when compared to the GUVisE dataset.

Comparing both satellite products to COMB (Fig. 4 panel (b)), a clear improvement is found compared to the GUVisE data.

MODIS and SEVIRI agree equally well with COMB, and show the same dependence of the bias as pointed out in panel (a). Since COMB is the combined product of MIC and GUVisE, it also contains only maritime and desert dust situations like GUVisE. Additionally, the COMB dataset consists of less data points than GUVis. Hence, it lacks statistical significance, so the difference in the results needs to be considered with caution.

The comparisons displayed in Fig. 4 show individually collocated satellite data versus Microtops. Fig. 5 presents the same

comparison, but simultaneous availability of data from all datasets (SEVIRI, MODIS, MIC) is required to preclude differences arising from a different sampling of cases. Therefore, the accuracy of SEVIRI and MODIS are directly comparable with respect to the MIC reference. This comparison shows that both AODs retrieved from SEVIRI and MODIS agree well with the shipborne reference, although the non-linear behavior of over- and underestimation is more pronounced for the SEVIRI retrieval. Since 550 nm is not a native spectral channel of SEVIRI, increased deviations in AOD are expected due to the uncertainty of the

AE calculated with the two spectral channels 630 and 810 nm and used for extrapolation. Therefore, a strong improvement of agreement is found comparing the 630 nm AOD to the shipborne reference in Fig. A3. Considering the AOD at 630 nm for the comparison, the SEVIRI performance is similar to MODIS, and even slightly better when compared to COMB and GUVisE, since the SEVIRI AOD is generally slightly lower than the MODIS AOD.

The results shown in Fig. 4, Fig. 5 and Fig. A3 are also presented in Table 5. Generally, the satellite-based AOD is higher

compared to the shipborne reference datasets, as is reflected in the bias > 0 for all comparisons. MODIS aerosol products show the highest linear correlation, which also slightly exceed those for the CAMS RA. *MxD04_3K* correlation is slightly higher than that for the *MxD04_L2* product, and the LOA is nearly equal for both MODIS products. Thus, this finding does not confirm the expectation of higher noise in the 3 km versus the 10 km product of MODIS expressed in Levy et al. (2015). Our comparison of *MxD04_3K* and *MxD04_L2* indicates a slightly lower AOD calculated in the former product, which would

presumably represent maritime conditions better. The analysis has also been repeated separately for the MODIS datasets based on the Terra and Aqua satellites (not shown), but only minor differences in the evaluation statistics for the individual satellites was found. Thus, only the combined MODIS dataset from Terra and AQUA are presented here. It also has to be stressed that the considered dataset is still relatively small compared to other validation studies, and should be repeated if more reference data becomes available. Nevertheless, the correlations found here agree well with the findings of Levy et al. (2013) for the MODIS

C6.1 aerosol products (0.93) and for the 550 nm channel, and exceed the findings of Bréon et al. (2011) for the SEVIRI aerosol product at 630 nm over ocean (0.795 versus 0.9 in this study), indicating a significantly better performance over ocean than





over land. The results for SEVIRI and MODIS show a similar agreement of the AOD compared to the reference data, but with a larger scatter of ±0.14 LOA for SEVIRI versus ±0.11 LOA for MODIS AOD.

Besides the AOD, the AE of the satellite products are evaluated as a quantity characterizing the spectral dependence of AOD, as shown in Fig. 6. The panels display the comparison of the difference of AE versus the different shipborne reference datasets as a scatter plot, indicating the bias, as well as the LOA and EE limits. The EE for AE is estimated to be ±0.4 for the MODIS products (Levy et al., 2010, 2013), and the same EE is applied for the SEVIRI AE product. In general, the MODIS AE agrees with this estimate of the EE limits, but shows a tendency to overestimate the shipborne AE, as reflected by the positive bias. The bimodal behaviour of AE of the MODIS products found for C5 in Levy et al. (2010) is not reproduced here, which agrees

with the findings for C6.1 presented in Levy et al. (2013). Also, MODIS AE meets the expected Gfrac of >67% for an EE of ±0.4, as was already found in Levy et al. (2013).

    The results for the SEVIRI AE show a general overestimation versus MIC, indicated by the positive bias. Furthermore, a bimodal behaviour of AE is found similar to that reported for the C5 MODIS products in Levy et al. (2010). The SEVIRI-based AE mostly lies close to two values: AE close to zero is associated with the models of oceanic and maritime aerosol used

by the retrieval (O99, M99, (Shettle and Fenn, 1979)). Another large fraction of the dataset is related to purely tropospheric aerosol models (type T99, T90, T50, (Shettle and Fenn, 1979)), covering AE from 1.29 to 1.61. Another frequent assignment of aerosol model are those for very small particles which cover the AE range from 1.8 to 2.4 (Thieuleux et al., 2005). Therefore, it can be concluded that SEVIRI retrieval of AE cannot realistically capture the variability in the AE which is observed from shipborne products, likely due to the limitation of using only two relatively closely spaced spectral channels of SEVIRI. Thus,

calculation of AOD extrapolated to wavelengths outside the range of the native channel 630 nm and 810 nm from the SEVIRI product based on the AE may lead to high uncertainties.

    It should be noted that the results comparing AE from satellite to MIC can be reproduced with the COMB dataset (Fig. 6 panel (a) and panel (b)), although the number of collocated measurements is small. Thus, this study should be extended with data from additional cruises in future.

After the general evaluation of the satellite-based AOD and AE products, their representation for different aerosol conditions is investigated. Since the data has been acquired over the Atlantic ocean, the most prominent conditions contain maritime or desert dust aerosol. To examine the representation of AOD and AE with respect to aerosol type, the layout presented in Toledano et al. (2007) is used for example in Fig. 7 and Fig. 8. Instead of the AOD at 440 nm, the wavelength of 630 nm is chosen, to match the SEVIRI channel at 630 nm. The aerosol type is classified by applying the Toledano et al. (2007) scheme to the MIC

data. Points related to a certain aerosol type are combined in the form of a covariance ellipse which spans 67 % of the related data points.

    Fig. 7 shows that overall, the AOD of the different products and instruments lie very close together, with a slight tendency to overestimate the AOD for desert dust (only MODIS products) or maritime aerosol types. In general, the satellite-based datasets overestimate the AE. These results confirm the statistics discussed before. The satellite ellipses are tilted compared to the MIC

ellipses, as a result of the assumed relation of AOD and AE in the retrieval, which is determined by the choice of aerosol model. This effect is strongly visible for SEVIRI in desert dust situations, because the AE is calculated from the 630 nm and





810 nm channels only. Satellite AOD in maritime conditions exhibits a stronger overestimation as also shown in Table A1 and Table A2. This effect might be related to the coarser spatial resolution of the satellite pixels or undetected cloud contamination (see Sect. 2.2). The spatial-mean mean AOD inferred from satellite pixels can deviate from the AOD which is retrieved from
slant transmission in case of MIC, due to the mismatch of spatial scales.

The most prominent feature of Fig. 7 is the deviation in AE for desert dust conditions. When compared to the shipborne products, satellite products show an AE which is more than two times larger for MODIS, and more than three times larger than for SEVIRI. This relates to a lack of realistic mineral dust models in the satellite retrievals. Since the MODIS product uses a larger set of spectral channels and aerosol models, it is able to estimate AE more accurately. This emphasize, that the Ångström
behaviour is not applicable for desert dust conditions, at least with a limited set of spectral channels.

Although, the AE is still the method of choice for extrapolating the AOD at the desired wavelengths to validate or increase observation capabilities, as is done in several studies (Kleidman et al., 2005). Therefore, one should be aware that during desert dust conditions, only the AOD from satellite products is accurate at available spectral channels only and that extrapolating AOD at other wavelengths using AE may lead to unexpected high uncertainties.

Figure 8 confirms above findings. In this figure, each dataset has been collocated individually to the MIC reference to increase the diversity of conditions. No noteworthy discrepancies are found for other aerosol types than maritime and desert dust. With the exception of the positive bias in AOD especially at lower values of AOD, and the overestimation of AE in particular for desert dust,the satellite aerosol products are found to agree closely to MIC, in particular for continental as well as biomass burning aerosol.

The previous statistics confirm that the AOD retrieved from satellite agrees well with the shipborne reference, but slightly overestimate AOD in general and especially at low AOD. AE is also overestimated for maritime and desert dust aerosol. Therefore, AOD is only represented well for the native spectral channels of the satellite instruments. The estimation of the spectral behaviour of AOD remains challenging, due to the lack of realism of the aerosol models (MODIS and SEVIRI), or the number of spectral channels available (SEVIRI). These findings are in particular applicable for conditions dominated by
mineral dust.

While the MODIS aerosol product is clearly the product of choice for many applications, i.e. for data assimilation and climate studies, due to its accuracy, availability and global coverage, the SEVIRI aerosol product is still of scientific interest due to its high temporal resolution of 15 min Bréon et al. (2011). The high temporal resolution however only adds information compared to products from polar-orbiting satellites, if the temporal variations of aerosol properties since the last overpass of a polar-
orbiting satellite exceed the error limits of the retrieval. Thus, it is not clear how much information can actually be gained from the higher temporal resolution of SEVIRI, as it is expected that AOD variation are generally small on the time scale of hours. To further investigate this point, MODIS collocations with the shipborne datasets are used to serve as random samples to study the AOD variability between successive overpasses. For each pixel of follow up MODIS images the corresponding SEVIRI AOD for every available SEVIRI image between both MODIS images was acquired to calculate the AOD variation. Relative to the
linear regression line of the follow up MODIS AOD, the standard deviation (STD) of SEVIRI AOD was calculated. Therefore, STD is a measure of the additional variation of AOD which cannot be seen in a MODIS only AOD product. Fig. 9 shows the



STD calculated for different time intervals between follow up MODIS images. The mean STD of AOD within six hours is slightly larger than 0.02. The STD is compared to the mean EE2 calculated using Eq.(4) and mean SEVIRI AOD, indicated by the green dashed line in Fig. 9. If the STD is larger than EE2 it points out a situation where AOD variation cannot be captured
by MODIS and can be called significant. SEVIRI aerosol measurements add information to the general AOD monitoring only if the AOD variation is significant. As Fig. 9 reveals, this is only true for slightly above 8% of all situations, knowing that, in general, Terra and Aqua satellites overpass the same region every three hours. This emphasises that, in terms of climate studies or data assimilation, the significant higher temporal resolution of SEVIRI does not lead to improvements for the majority of situations, unless the accuracy of this product could be significantly improved. In fact, such an improvement could in future be
possible with the third generation of Meteosat (MTG). Nevertheless, with the high temporal resolution, the SEVIRI product may be needed for many applications, such as case studies of dust or smoke plume development, where high variability of AOD is expected.

### 4.3 CAMS RA evaluation

Alongside the evaluation of satellite aerosol products described above, results for the CAMS RA AOD are presented in Table 5.
In comparison to MIC as reference dataset, the Table 5 shows that CAMS RA AOD agrees closely to MIC, since the correlation is 0.92 and the bias is about zero. The LOA of $\pm 0.13$ is similar to the one found for the products of SEVIRI and MODIS compared to MIC, with values ranging from $\pm 0.12$ to $\pm 0.15$. The correlation of the AOD difference R(D) is close to zero, indicating that the difference of CAMS RA and MIC AOD is not linearly related. Compared to the statistics calculated for the satellite products, shows that CAMS RA performance is clearly superior to the SEVIRI dataset, since the correlation is
increased from 0.90 to 0.92 as well as the LOA from 0.15 to 0.13. Further, the overestimation of AOD found for the satellite products, indicated by the bias of about 0.03, is not present in the CAMS RA AOD. This emphasize that although the MODIS AOD is assimilated, the overestimation of AOD is compensated in CAMS RA. This effect is clearly shown in Fig. 10, together with a tendency of CAMS RA towards an underestimation of AOD for larger values of AOD. As the evaluation shown in Table 5 compares each aerosol dataset individually against each reference dataset, it offers the largest amount of collocated data
and is best for obtaining the individual statistics. The performance of each product is however not directly inter-comparable among the datasets, due to potential differences arising from different sampling of conditions. To address this point, Table 6 is given, which shows the same statistics as in Table 5, but requires simultaneous availability of all datasets for enabling a direct inter-comparison. Table 6 confirms the conclusions drawn from Table 5. (i) CAMS RA compensates for the overestimation of AOD provided by MODIS aerosol products. The bias calculated for CAMS RA equals zero in this dataset. (ii) the CAMS RA
AOD shows larger scatter, indicated by larger LOA and less correlation than the MODIS products. Therefore, in terms of consistency, the MODIS aerosol products show better performance than CAMS RA. (iii) CAMS RA performance is clearly superior to the SEVIRI dataset, with an increased correlation and no overestimation of AOD.

In terms of assimilated aerosol observations, data from MODIS and AATSR are used by the IFS for CAMS RA starting from the year 2003 until March 2012, when the ENVISAT mission ended due to loss of contact to the satellite. After March
2012, only the MODIS AOD is used (Inness et al., 2019). Table 7 shows the evaluation of CAMS RA versus MIC for the





different time periods, to investigate potential differences in quality. Since CAMS RA can provide AOD regardless of cloud cover and satellite orbit, a comparison of CAMS RA AOD to MIC conditioned on the availability of collocated MODIS data is also shown. The results show no significant difference in terms of correlation, bias or LOA. Without AATSR, MODIS is the only contributor for data assimilation in terms of AOD. Comparing the results of CAMS RA with and without AATSR, the

performance of CMAS RA with additional AATSR data is increased, indicated by increased correlation from 0.87 to 0.90 and lower LOA, dropping from 0.18 to 0.14. This shows that the AATSR observations lead to an improvement of the representation of aerosol in CAMS RA. Inness et al. (2019) suspected an slight increase of CAMS RA AOD without AATSR, which cannot be observed in this study. As the analysis presented here is based on a limited number of data points, it is unclear whether these findings are statistically significant, and the discussed tendencies should be considered with caution.

To evaluate CAMS RA AOD with respect to the representation of aerosol type, Table 7 lists evaluation statistics calculated only for maritime and only for desert dust aerosol type, in comparison to the statistics of the whole dataset of CAMS RA versus MIC. For maritime aerosol, CAMS RA AOD has lowest correlation of 0.7. The values of LOA are lowest for maritime, which is expected since this measure favors lower AOD and maritime aerosol situations are generally connected to low AOD values. A slight overestimation of 0.01 is shown by the bias considering only maritime aerosol situations. Compared to MODIS for

maritime aerosol type, the overestimation found for CAMS RA AOD in Table 7 is less significant as shown in Table A1. This emphasizes that CAMS RA exceeds the MODIS accuracy for maritime aerosol conditions in terms of AOD. For desert dust conditions, the correlation of CAMS RA to MIC (0.85) is similar to the one found for *MxD04_L2* (0.86) in Table A2, although the correlation of *MxD04_3K* is largest with 0.92. As for maritime aerosol, the overestimation of AOD is compensated in the CAMS RA aerosol product. This emphasizes that the CAMS RA aerosol product is comparable in accuracy to the MODIS

products in maritime and desert dust situations.

Inness et al. (2019) reported an overestimation of AE in CAMS RA compared to AERONET stations of about 5-20%. From the comparison to MIC presented in Fig. 6 (panel (f)), the same conclusion can be drawn based on our dataset, showing a positive bias of 0.17. Compared to MODIS, similar values are found for Gfrac, but while the MODIS AE scatters more equally around the reference AE, CAMS RA AE is clearly distributed above zero. Also, the AE difference of CAMS RA and

MIC shows a increased linear dependency indicated by increased correlation R(D). This indicates that similar to the SEVIRI product, certain aerosol models are favored in the processing. Nevertheless, the overall scatter of AE indicated by the values of LOA is lower for CAMS RA.

As stated by Inness et al. (2019), the overestimation of AE results from a deficit in the handling of the coarse dust fraction in the model. The total AOD calculated in CAMS RA is composed of less dust than in its predecessor versions, which explains the

higher overall AE. Nevertheless, the comparison of CAMS RA AE with respect to aerosol type in Fig. 11 reveals that the AE for desert dust agrees best with the MIC reference, compared to the satellite products. The slightly better representation compared even to MODIS indicates that the representation of the spectral dependence of AOD for dust is most realistic in CAMS RA. For maritime aerosol, CAMS RA AE shows a similar overestimation compared to the satellite products, but with less scatter. This emphasises a more consistent representation of maritime aerosol in CAMS RA as compared to satellite products. The

CAMS RA AE representation versus MIC in Fig. 8 shows a close agreement for all aerosol types, except an overestimation of



AE in maritime conditions, and a tendency for overestimation during dust conditions with low AOD. In general, the AE and AOD of CAMS RA is similar or in some instances even exceeds the accuracy of the satellite retrievals including MODIS in comparison to the reference data presented in this study.

## 5 Conclusions

Within this paper, a comprehensive evaluation of MODIS and SEVIRI AOD products as well as the representation of AOD in the CAMS reanalysis has been presented with shipborne reference datasets. For this purpose, available Microtops observations from MAN across the Atlantic ocean were utilized, and complemented by a unique set of shipborne aerosol products collected during five Atlantic transit cruises of RV *Polarstern* with the multi spectral shadowband radiometer GUVis-3511.

Three separate aspects have been investigated within the study:

(i) First, the two shipborne datasets were inter-compared to verify their consistency. Extending the comparison presented in Witthuhn et al. (2017), the AOD derived from the GUVis and Microtops instruments from five cruises with RV *Polarstern* were compared. A substantial update of the GUVis processing algorithm is shown to address several shortcomings identified in the prior version. To improve upon the lamp-based instrumental calibration of the GUVis, the method of Alexandrov et al. (2002) has been applied to obtain a cross-calibration based on the MIC observations. In addition, an underestimation of AOD

by the GUVis instrument compared to the MIC has been observed, which is related to strong forward scattering by aerosol, and arises from the broad shadowband and the much wider effective field of view, compared to the MIC observations (Russell, 2004). Combining the cross-calibration with an empirical correction following the approach of di Sarra et al. (2015) and Wood et al. (2017), a correlation >0.992 is found for all spectral channels. The uncertainty estimate of ±0.02 for the GUVis AOD is shown to be valid after applying these two corrections.

Compared to the manually operated Microtops instrument, an important advantage of the GUVis dataset is its high temporal resolution as well as the uniformity of sampling. These automated shipborne measurements lead to a larger collocated dataset for satellite evaluation, which in turn leads to more robust evaluation statistics. They also offer the chance to conduct such observations on more cruises, as they greatly reduce the amount of effort to operate the instrument.

(ii) Second, the shipborne datasets have been utilized to evaluate the MODIS *MxD04* and SEVIRI AOD products. The

satellite products differ in temporal and spatial resolution as well as in number of spectral channels available from the satellite instruments. The AOD has been compared at 550 nm (used in previous validation studies) and at 630 nm, the latter being a native channel of the SEVIRI instrument, enabling a consistent and fair evaluation of the aerosol products from both satellite sensors. For non-native channels, interpolation of AOD based on the AE using the Ångström relation have been used. The AOD at these two wavelengths have been compared with collocated Microtops measurements and show similar agreement, although

the comparison to SEVIRI AOD shows larger scatter (about 25%) and therefore less correlation than the one to MODIS.

Previous evaluation studies of the MODIS aerosol products have utilized the EE limits at 550 nm (defined as spanning at least 67% of the data) of ±(0.03+0.05 AOD) (EE1) (e.g., Abdou et al., 2005; Remer et al., 2008; Livingston et al., 2014) and $[+(0.04+0.1\,\mathrm{AOD}),-(0.02+0.1\,\mathrm{AOD})]$ (EE2) by Levy et al. (2013) over ocean. The EE1 limits are missed slightly by the



67% criterion, while the EE2 limits are confirmed by this study. The EE2 limits account for a general overestimation of AOD by the MODIS satellite products of about 0.02, which is close to the value of 0.03 found here. The SEVIRI aerosol product also meets the EE2 limits for the interpolated AOD at 550 nm.

Since SEVIRI has only channels at wavelengths of 630 and 810 nm, which lie relatively close together, the accuracy for calculating the AE is significantly degraded. The representation of the spectral dependence of AOD therefore is superior in the MODIS products, due to the large set of available spectral channels, combined with the mature set of aerosol models used in the retrievals. This manifests itself in a consistent accuracy of AOD for all available spectral channels utilizing the AE for the MODIS product, which is not the case for SEVIRI.

Evaluating the satellite products with a focus on aerosol type reveals that the main challenge arises from the identification of realistic aerosol models for use in the retrieval (for both MODIS and SEVIRI), and from the limited number of spectral channels (for SEVIRI). Therefore, the AOD and AE from the SEVIRI product should not be used to extrapolate the AOD to wavelengths outside the available spectral channel range. Given the large number of channels, the MODIS AOD at non-native wavelengths is significantly more accurate than that of the SEVIRI product, but still relies on the underlying aerosol model, which can introduce uncertainties depending on aerosol conditions. In particular, the AE calculated from satellites during aerosol conditions dominated by mineral dust aerosol shows values which are two times (MODIS) and three times (SEVIRI) larger than the AE from shipborne products.

Avoiding spectral conversions, our results confirm that satellite products can serve well to provide a global view of AOD e.g. for climate studies. This finding is consistent with results of former validation studies for the MODIS instrument (e.g., Munchak et al., 2013; Levy et al., 2013; Livingston et al., 2014). The quality of MODIS products is continuously monitored over land by comparison with products from worldwide AERONET stations.

In most situations, temporal variations of AOD within a window of six hours are smaller than the uncertainty limits of the satellite products. Hence, the better time resolution of SEVIRI and other geostationary satellite sensors offers only minor benefits compared to the use of polar-orbiting satellite platforms, given its increased uncertainties. Nevertheless, the SEVIRI AOD product can still provide valuable information on the temporal evolution of AOD fields for specific cases with high temporal variability such as dust storms.

(iii) Finally, the aerosol fields obtained from the CAMS RA have been evaluated versus collocated Microtops measurements. The performance of CAMS RA is rather close to that of the MODIS product. The differences of MODIS and CAMS RA arise mainly from the model handling of different aerosol types: while an overestimation of AOD observed for MODIS for maritime and desert dust aerosol is compensated in CAMS RA, the overall consistency of MODIS AOD exceeds CAMS RA AOD, indicated by larger correlation of MODIS AOD to the reference datasets.

Finally, it has to be noted that the evaluation presented here is still based on a relatively small set of collocated shipborne and satellite observations. For more meaningful results, a significantly larger shipborne dataset would be desirable.



## 6 Outlook

Ground-based and shipborne observations will continue to play an important role for monitoring and investigating aerosols at a global scale. Applications range from the evaluation and monitoring of satellite products to independent studies targeting radiative closure and aerosol processes, which cannot be resolved by satellite datasets. Shipborne observations of aerosol

optical properties with the Microtops sunphotometer will continue within MAN, and will be complemented by the GUVis shadowband radiometer on future OCEANET cruises. The GUVis measures direct and diffuse irradiance simultaneously. It is thus well suited to extend the aerosol products by additional parameters such as single scattering albedo and asymmetry parameter utilizing the diffuse to direct ratio as outlined by Herman et al. (1975) and applied in a number of previous studies (e.g., Petters et al., 2003; Kassianov et al., 2007). It also offers the chance of an evaluation of the direct radiative effect of

aerosol.

Current efforts are also directed to operate state-of-the art Cimel sunphotometers on shipborne platforms (Yin et al., 2019). While the automatic operation of those instruments on a moving platform still poses a significant challenge, the high accuracy offered by sunphotometers combined with recent advances in navigation and alignment sensors make this seem a promising approach for the future. Fully automated Cimel sunphotometer observations on ship including the capability of sky scans will

open up the full potential of the well-developed AERONET aerosol product for studies over ocean.

Alongside these ongoing effort in shipborne observations, a number of promising new satellite missions will be launched within the next years, whose validation will increase the demand for reliable reference datasets.

(i) With the launch of Meteosat third generation (MTG) operated by EUMETSAT, SEVIRI on MSG will be replaced by the Flexible Combined Imager. This will lead to observations with an increased spatial resolution comparable to that of MODIS,

but with the benefits of the geostationary satellite perspective (in particular in terms of temporal resolution). The set of available spectral channels will also increase, including channels at 440 and 510 nm wavelength. Since the AOD retrieved at SEVIRI's native spectral channels already has been shown to have satisfactory accuracy here, the availability of MTG observations should increase the accuracy of aerosol products to the level of MODIS, including significant improvements in aerosol model selection and AE calculation within the retrievals. The high temporal resolution of MTG will thus provide novel information

on the spatio-temporal distribution of aerosols with MODIS-like accuracy, which will be valuable for studies targeting air quality or aerosol transport. These data are also expected to be useful for data assimilation into CAMS RA, and can provide information on temporal changes beyond the current time resolution of CAMS RA.

(ii) The Earth Cloud Aerosol and Radiation Explorer (EarthCARE) satellite will be launched by ESA in 2021. This polar orbiting satellite mission utilizes a combination of instruments including the Multi-Spectral Imager (MSI) spectral radiometer

system, which utilize spectral channels in the visible and near infrared region similar to the SEVIRI instrument. In addition, the Atmospheric Lidar (ATLID) will provide vertical profiles of extinction at 355 nm, and thereby reveal new information on the vertical distribution of aerosols and thin clouds. This unique feature will benefit for scientific studies targeting aerosols including their radiative effects. The synergy of the MSI and ATLID instruments will open up new opportunities for the retrieval and classification of aerosol properties, and will provide new insights on the vertical distribution of aerosol optical



properties. Based on our findings, it seems particularly important to combine MSI and ATLID information to constrain the spectral dependence of aerosol properties, due to the limitations reported here arising for the SEVIRI wavelengths.

(iii) Following on from the EUMETSAT Polar System program (Metop), the second generation of European polar orbiting spacecraft (EPS-SG) will continue the meteorological observations in the morning orbit from 2022 onward. The Multi-Viewing Multi-Channel Multi-Polarisation Imaging (3MI) instrument on board this mission utilizes 12 spectral channels from 410 to

2130 nm and with a nadir resolution of 4 km. Together with information on light polarization, these observations will provide unique observations for the estimation and characterization of aerosol optical properties at a global scale.

With all these upcoming satellite observations, the consistency of the different aerosol products will become an important aspect for future analyses, in particular with respect to aerosol type. Reliable ground-based reference datasets will continue to play an important role for their evaluation, and for reconciling the unavoidable discrepancies between datasets.

*Data availability.* The datasets from all instruments used in this study are public available:

GUVis data of AOD and is available from the Pangaea database (https://doi.org/10.1594/PANGAEA.872377).

Microtops data is available over the MAN AERONET website (https://aeronet.gsfc.nasa.gov/new_web/maritime_aerosol_network.html).

MODIS aerosol products called *Effective_Optical_Depth_Average_Ocean* are available through the Level-1 and Atmosphere Archive & Distribution System (LAADS) Distributed Active Archive Center (DAAC) (https://ladsweb.modaps.eosdis.nasa.gov).

The SEVIRI aerosol products are available from the ICARE Data and Services Center (http://www.icare.univ-lille1.fr).

CAMS RA data can be acquired from the ECMWF public dataset catalogue (https://apps.ecmwf.int/datasets).

*Author contributions.* Jonas Witthuhn developed and implement processing scheme and conducted AOD calculation from the GUVis-3511 shadowband radiometer. Jonas Witthuhn calculated the evaluation statistics and prepared the manuscript. Anja Hünerbein and Hartwig Deneke contributed expertise about satellite remote sensing, provided helpful advice on the work and paper, and contributed to writing the

manuscript.

*Competing interests.* The authors declare no competing interests.

*Acknowledgements.* We thank Stefan Kinne and Alexander Smirnov for their effort in maintaining and organisation of Microtops observations on RV *Polarstern*, as well as all scientists operating the Microtops during the cruises.

Thanks are also due to the Alfred Wegener Institute for Polar and Marine Research (AWI) for the opportunity to operate the GUVis

instrument during the research cruises PS83, PS95, PS98, PS102 and PS113 across the Atlantic Ocean on RV *Polarstern*.

We thank the ICARE Data and Services Center for providing access to the data used in this study.



The Terra/MODIS and Aqua/MODIS Aerosol 5-Min L2 Swath 10 km and 3K Swath 3 km datasets (Levy et al., 2015) were acquired from the Level-2 and Atmosphere Archive & Distribution System (LAADS) Distributed Active Archive Center (DAAC), located in the Goddard Space Flight Center in Greenbelt, Maryland (https://ladsweb.nascom.nasa.gov/).

This paper contains modified Copernicus Atmosphere Monitoring Service information [2019], neither the European Commission nor ECMWF is responsible for any use that may be made of the information it contains.



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





**Table 1.** Technical specifications of the GUVis and Microtops instruments.

| Characteristics | GUVis-3511 | Microtops II |
|---|---|---|
| spectral channels | 18x [310-1640 nm] | 5x [380-870 nm] |
| | 1x unfiltered | 1x 940 nm (water vapor) |
| FWHM | 10 nm | 10 nm (4 nm at 380 nm) |
| Measurement frequency | 15 Hz | - |
| Sweep period | 40 s | - |
| Time resolution | 1 min | variable (>10 min) |
| viewing/shading angle | 13-15° | 2.5° |
| weight | 5 kg | 0.6 kg |
| dimensions | 24x24x36 cm | 10x20x4 cm |



**Table 2.** Number of available data points in the MIC, GUV, and COMB datasets, given as total number, and separated by aerosol type. The type classification follows the scheme of Toledano et al. (2007). Collocated data points for the comparison with satellite and model datasets (see text for collocation criteria) are also given.

| Class | MIC | CAMS | *MxD04_L2* | *MxD04_3K* | SEVIRI |
|---|---|---|---|---|---|
| maritime | 12749 | 1478 | 1090 | 1064 | 7218 |
| desert dust | 5085 | 730 | 296 | 261 | 2023 |
| biomass burning | 225 | 44 | 14 | 6 | 116 |
| continental | 931 | 174 | 92 | 93 | 599 |
| mixed | 252 | 48 | 23 | 22 | 100 |
| no class | 8 | 0 | 2 | 2 | 5 |
| total | 19250 | 2474 | 1517 | 1448 | 10061 |
| Class | GUVis | CAMS | *MxD04_L2* | *MxD04_3K* | SEVIRI |
| maritime | 6179 | 93 | 92 | 131 | 793 |
| desert dust | 2552 | 31 | 27 | 46 | 240 |
| biomass burning | 0 | 0 | 0 | 0 | 0 |
| continental | 152 | 1 | 0 | 0 | 24 |
| mixed | 32 | 2 | 0 | 1 | 2 |
| no class | 1497 | 14 | 28 | 32 | 67 |
| total | 10412 | 141 | 147 | 210 | 1126 |
| Class | COMB | CAMS | *MxD04_L2* | *MxD04_3K* | SEVIRI |
| maritime | 698 | 84 | 73 | 86 | 355 |
| desert dust | 278 | 23 | 16 | 13 | 88 |
| biomass burning | 0 | 0 | 0 | 0 | 0 |
| continental | 5 | 1 | 0 | 0 | 3 |
| mixed | 0 | 0 | 0 | 0 | 0 |
| no class | 0 | 0 | 0 | 0 | 0 |
| total | 981 | 108 | 89 | 99 | 446 |





**Table 3.** Per–channel coefficients obtained for the cross-calibration of the GUVis to the MIC instrumental channels (determined per cruise), expressed as relative correction $C$ to the most recent laboratory calibration, and as absolute calibration coefficient $k$, together with the empirical scaling coefficient $S$ to correct the AOD for the forward scattering contribution in the GUVis observations (determined for all cruises). See text for details on their estimation.

| Channel nm | $C_{PS83}$ - | $k_{PS83}$ $\frac{V\,m^2\,nm}{W}$ | $C_{PS95}$ - | $k_{PS95}$ $\frac{V\,m^2\,nm}{W}$ | $C_{PS98}$ - | $k_{PS98}$ $\frac{V\,m^2\,nm}{W}$ | $C_{PS102}$ - | $k_{PS102}$ $\frac{V\,m^2\,nm}{W}$ | $C_{PS113}$ - | $k_{PS113}$ $\frac{V\,m^2\,nm}{W}$ | $S$ - |
|---|---|---|---|---|---|---|---|---|---|---|---|
| 380 | 1.02 | 1.42 | 0.97 | 1.44 | 1.03 | 1.32 | 0.96 | 1.31 | 0.99 | 1.29 | 1.12 |
| 440 | 0.98 | 7.52 | 0.93 | 7.48 | 0.96 | 6.90 | 0.92 | 6.87 | 0.95 | 6.79 | 1.13 |
| 500 | 1.02 | 20.91 | 0.99 | 21.33 | 1.03 | 19.55 | 0.97 | 19.63 | 1.00 | 19.83 | 1.14 |
| 630 | 0.99 | 43.57 | 0.98 | 43.83 | 1.01 | 39.90 | 0.96 | 39.95 | 0.98 | 40.09 | 1.13 |
| 675 | 1.02 | 51.64 | 1.01 | 51.96 | 1.05 | 46.74 | 1.01 | 46.90 | 1.01 | 47.32 | 1.13 |
| 810 | 0.97 | 48.86 | 0.98 | 48.43 | 0.99 | 43.47 | 0.95 | 43.60 | 0.94 | 43.92 | 1.11 |
| 870 | 0.96 | 48.86 | 0.98 | 48.43 | 1.00 | 43.47 | 0.96 | 43.60 | 0.94 | 43.92 | 1.10 |





**Table 4.** Statistics comparing the old and new processing algorithms of the GUVis observations, as well as the calibration– and forward-scattering–corrected GUVis dataset versus the MIC dataset. Availability of Microtops and GUVis channels are indicated with (x: available; i: internally interpolated; -: interpolated using AE). Statistics include the number of datapoints N, Pearson correlation coefficient (R), the Pearson correlation coefficient of difference (R(D)), and bias plus fraction of outliers based on the limit of agreement (LOA) method for 95% confidence interval (see Sect. 3.3 for explanation of R(D), bias and LOA).

| Channel [nm] | availability MIC | availability GUVis | N - | R - | R(D) - | bias ± LOA - | bias ± LOA weighted | Outlier % |
|---|---|---|---|---|---|---|---|---|
| | | | | GUVis data (old processing) | | | | |
| 380 | x | x | 993 | 0.954 | -0.43 | -0.03±0.10 | -1.03±3.49 | 19.94 |
| 440 | x | x | 993 | 0.965 | -0.58 | -0.03±0.09 | -1.12±3.23 | 17.42 |
| 500 | i | x | 993 | 0.965 | -0.66 | -0.03±0.09 | -1.07±3.32 | 13.39 |
| 630 | - | - | 993 | 0.969 | -0.67 | -0.03±0.09 | -0.97±3.13 | 12.19 |
| 675 | x | x | 993 | 0.968 | -0.67 | -0.02±0.09 | -0.86±3.15 | 10.98 |
| 810 | - | - | 993 | 0.970 | -0.62 | -0.03±0.08 | -0.92±2.89 | 10.78 |
| 870 | x | x | 993 | 0.970 | -0.62 | -0.03±0.08 | -0.91±2.86 | 10.47 |
| | | | | GUVis data (new processing) | | | | |
| 380 | x | x | 1061 | 0.988 | -0.66 | -0.02±0.06 | -0.78±2.23 | 11.12 |
| 440 | x | x | 1061 | 0.989 | -0.69 | -0.02±0.06 | -0.87±2.16 | 10.74 |
| 500 | i | x | 1061 | 0.990 | -0.73 | -0.02±0.06 | -0.69±2.07 | 9.52 |
| 630 | - | - | 1061 | 0.991 | -0.74 | -0.02±0.06 | -0.67±1.96 | 9.43 |
| 675 | x | x | 1061 | 0.992 | -0.75 | -0.02±0.05 | -0.56±1.86 | 8.11 |
| 810 | - | - | 1061 | 0.992 | -0.68 | -0.02±0.05 | -0.59±1.71 | 7.26 |
| 870 | x | x | 1061 | 0.992 | -0.67 | -0.02±0.05 | -0.55±1.66 | 6.69 |
| | | | | Enhanced GUVis data (GUVisE) | | | | |
| 380 | x | x | 1006 | 0.992 | -0.13 | 0.00±0.04 | 0.04±1.42 | 0.99 |
| 440 | x | x | 1006 | 0.993 | -0.12 | 0.00±0.04 | 0.03±1.32 | 0.70 |
| 500 | i | x | 1006 | 0.994 | -0.14 | 0.00±0.03 | 0.04±1.20 | 0.10 |
| 630 | - | - | 1006 | 0.995 | -0.17 | 0.00±0.03 | 0.06±1.07 | 0.00 |
| 675 | x | x | 1006 | 0.996 | -0.13 | 0.00±0.03 | 0.04±0.96 | 0.00 |
| 810 | - | - | 1006 | 0.995 | -0.11 | 0.00±0.03 | 0.03±0.95 | 0.10 |
| 870 | x | x | 1006 | 0.996 | -0.11 | 0.00±0.03 | 0.03±0.93 | 0.00 |





**Table 5.** Statistics comparing the GUVisE, COMB and MIC reference AOD at wavelengths of 550 and 630 nm versus the CAMS, SEVIRI and *MxD04* AOD products. N denotes the number of collocated data point. Listed are also the correlation (R), coefficients of a linear regression, followed by the correlation of the difference (R(D)) and the bias/fraction of data based on the limit of agreement (LOA) method for 95% confidence interval. G1 and G2 indicate the percentage of data points laying within the expected error limits EE1 and EE2.

| Reference | Instrument | N | channel | R | linear regression | R(D) | bias ± LOA | G1 | G2 |
|---|---|---|---|---|---|---|---|---|---|
| - | - | - | nm | - | - | - | - | % | % |
| GUVisE | CAMS RA | 107 | 550 | 0.91 | Y = 0.89 X-0.01 | 0.06 | 0.03±0.10 | 52 | 71 |
| | | | 630 | 0.91 | Y = 0.90 X-0.00 | 0.03 | 0.02±0.09 | 66 | 78 |
| | SEVIRI | 457 | 550 | 0.80 | Y = 0.58 X+0.03 | 0.47 | 0.05±0.20 | 51 | 60 |
| | | | 630 | 0.78 | Y = 0.59 X+0.03 | 0.42 | 0.03±0.19 | 58 | 67 |
| | *MxD04_3K* | 62 | 550 | 0.94 | Y = 0.78 X-0.00 | 0.50 | 0.04±0.14 | 55 | 73 |
| | | | 630 | 0.94 | Y = 0.79 X+0.00 | 0.45 | 0.03±0.14 | 66 | 77 |
| | *MxD04_L2* | 65 | 550 | 0.93 | Y = 0.67 X+0.02 | 0.69 | 0.06±0.19 | 51 | 68 |
| | | | 630 | 0.93 | Y = 0.67 X+0.02 | 0.66 | 0.05±0.19 | 55 | 68 |
| COMB | CAMS RA | 107 | 550 | 0.91 | Y = 0.88 X-0.01 | 0.10 | 0.03±0.09 | 57 | 77 |
| | | | 630 | 0.92 | Y = 0.90 X-0.00 | 0.05 | 0.02±0.09 | 68 | 82 |
| | SEVIRI | 439 | 550 | 0.81 | Y = 0.56 X+0.03 | 0.54 | 0.05±0.19 | 48 | 61 |
| | | | 630 | 0.80 | Y = 0.57 X+0.03 | 0.49 | 0.04±0.18 | 58 | 69 |
| | *MxD04_3K* | 58 | 550 | 0.98 | Y = 0.91 X-0.02 | 0.31 | 0.03±0.07 | 66 | 79 |
| | | | 630 | 0.98 | Y = 0.93 X-0.02 | 0.23 | 0.03±0.06 | 74 | 86 |
| | *MxD04_L2* | 62 | 550 | 0.94 | Y = 0.71 X+0.01 | 0.67 | 0.05±0.15 | 53 | 68 |
| | | | 630 | 0.94 | Y = 0.71 X+0.01 | 0.64 | 0.04±0.15 | 60 | 77 |
| MIC | CAMS RA | 2472 | 550 | 0.92 | Y = 0.92 X+0.01 | -0.00 | 0.00±0.13 | 59 | 66 |
| | | | 630 | 0.93 | Y = 0.95 X+0.01 | -0.06 | 0.00±0.13 | 63 | 70 |
| | SEVIRI | 10060 | 550 | 0.90 | Y = 0.81 X-0.00 | 0.21 | 0.03±0.15 | 56 | 68 |
| | | | 630 | 0.89 | Y = 0.87 X-0.00 | 0.06 | 0.02±0.14 | 63 | 73 |
| | *MxD04_3K* | 704 | 550 | 0.95 | Y = 0.87 X-0.00 | 0.30 | 0.03±0.12 | 65 | 78 |
| | | | 630 | 0.95 | Y = 0.87 X-0.00 | 0.28 | 0.03±0.12 | 67 | 77 |
| | *MxD04_L2* | 924 | 550 | 0.93 | Y = 0.86 X+0.00 | 0.22 | 0.03±0.13 | 65 | 76 |
| | | | 630 | 0.93 | Y = 0.86 X-0.00 | 0.20 | 0.02±0.13 | 66 | 77 |





**Table 6.** Statistics comparing the MIC AOD at 550 and 630 nm as reference versus the CAMS, SEVIRI and *MxD04* AOD products. Same statistics used as in Table 5, but requiring simultaneous availability of all products.

| Instrument | N | channel | R | linear regression | R(D) | bias ± LOA | G1 | G2 |
|---|---|---|---|---|---|---|---|---|
| - | - | nm | - | - | - | - | % | % |
| CAMS RA | 68 | 550 | 0.92 | Y = 0.88 X+0.02 | 0.12 | 0.00±0.13 | 71 | 76 |
| | | 630 | 0.92 | Y = 0.90 X+0.02 | 0.06 | 0.00±0.12 | 72 | 78 |
| SEVIRI | 68 | 550 | 0.91 | Y = 0.96 X-0.02 | -0.14 | 0.03±0.13 | 44 | 62 |
| | | 630 | 0.91 | Y = 1.02 X-0.02 | -0.27 | 0.01±0.13 | 56 | 63 |
| *MxD04_3K* | 68 | 550 | 0.96 | Y = 0.91 X+0.00 | 0.17 | 0.02±0.10 | 76 | 85 |
| | | 630 | 0.95 | Y = 0.91 X+0.00 | 0.16 | 0.01±0.10 | 75 | 85 |
| *MxD04_L2* | 68 | 550 | 0.95 | Y = 0.91 X+0.00 | 0.15 | 0.02±0.10 | 75 | 82 |
| | | 630 | 0.95 | Y = 0.91 X+0.00 | 0.14 | 0.01±0.10 | 69 | 82 |


**Table 7.** Statistics comparing the MIC AOD at 550 and 630 nm as reference versus CAMS AOD for selected conditions (see text for details). Same statistics used as in Table 5.

| CAMS RA selection | N | channel | R | linear regression | R(D) | bias ± LOA | G1 |
|---|---|---|---|---|---|---|---|
| - | - | nm | - | - | - | - | % |
| with MxD04_L2 | 293 | 550 | 0.92 | Y = 0.92 X+0.02 | -0.01 | 0.00±0.13 | 58 |
| | | 630 | 0.93 | Y = 0.96 X+0.01 | -0.09 | -0.00±0.13 | 62 |
| no MxD04_L2 | 2181 | 550 | 0.92 | Y = 0.92 X+0.01 | -0.00 | 0.00±0.14 | 59 |
| | | 630 | 0.93 | Y = 0.94 X+0.01 | -0.05 | 0.00±0.13 | 63 |
| with AATSR | 941 | 550 | 0.90 | Y = 0.88 X+0.03 | 0.06 | -0.00±0.14 | 57 |
| | | 630 | 0.91 | Y = 0.90 X+0.02 | 0.01 | -0.00±0.13 | 60 |
| no AATSR | 190 | 550 | 0.87 | Y = 0.89 X+0.02 | -0.06 | 0.00±0.18 | 49 |
| | | 630 | 0.86 | Y = 0.93 X+0.02 | -0.14 | -0.00±0.18 | 51 |
| only maritime | 1478 | 550 | 0.72 | Y = 0.48 X+0.04 | 0.51 | 0.01±0.08 | 69 |
| | | 630 | 0.72 | Y = 0.50 X+0.03 | 0.47 | 0.01±0.07 | 74 |
| only desert dust | 730 | 550 | 0.85 | Y = 0.83 X+0.06 | 0.03 | -0.00±0.20 | 46 |
| | | 630 | 0.86 | Y = 0.86 X+0.06 | -0.00 | -0.01±0.20 | 47 |
| all | 2474 | 550 | 0.92 | Y = 0.92 X+0.01 | -0.00 | 0.00±0.13 | 59 |
| | | 630 | 0.93 | Y = 0.95 X+0.01 | -0.06 | 0.00±0.13 | 63 |

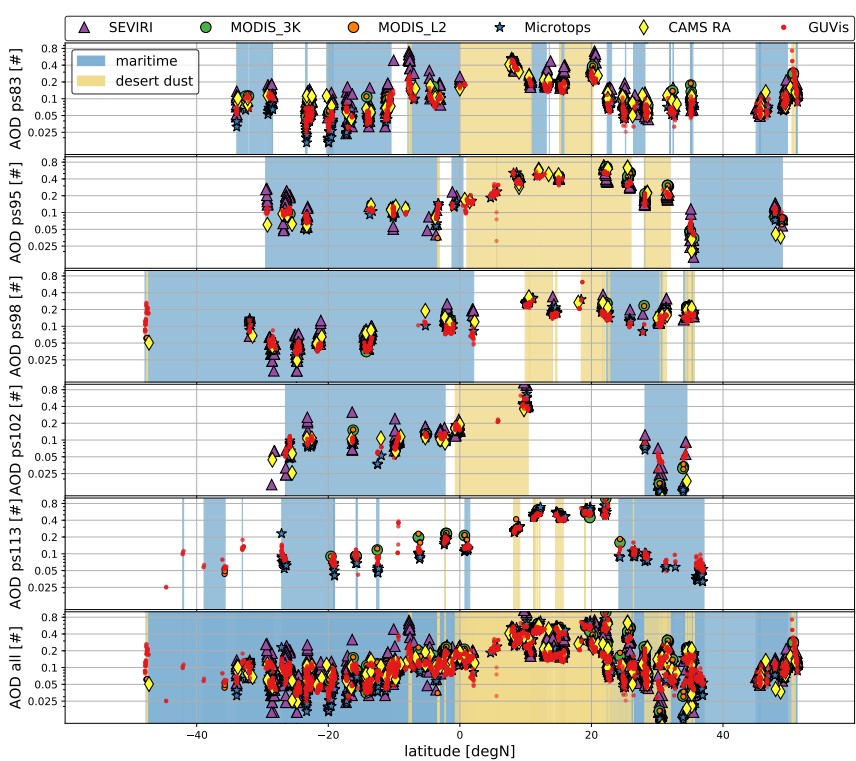

**Figure 1.** Zonal cross-section of AOD at 630 nm estimated from GUVis measurements during the *Polarstern* cruises PS83, PS95, PS98, PS102 and PS113, together with collocated AOD obtained from Microtops, satellite products and CAMS RA. Along this cross-section across the Atlantic ocean, the dominant aerosol type is either maritime (blue shaded region) or desert dust (yellow shaded region) while passing the Sahara desert. The Aerosol classification is based on the method of Toledano et al. (2007) and GUVis products.

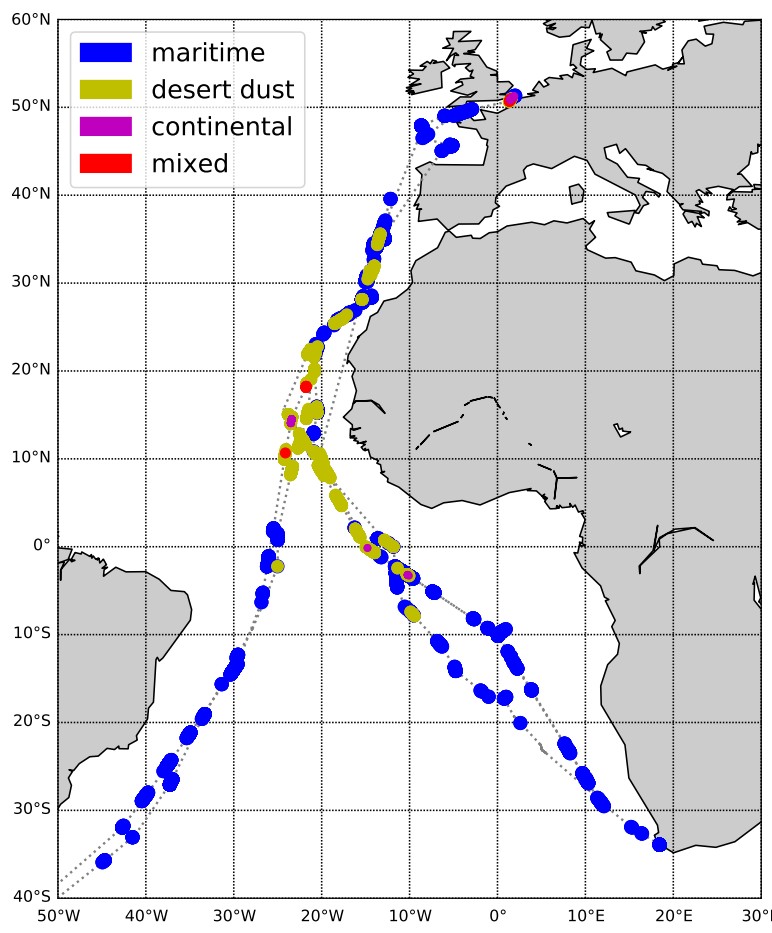

**Figure 2.** Cruise tracks of the Polarstern during PS83, PS95, PS98, PS102 and PS113. The resulting aerosol type classification obtained from the GUVis observation is shown by color coding.





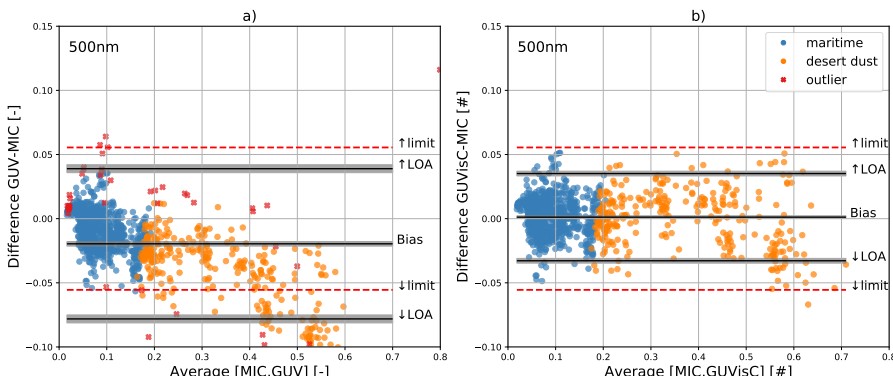

**Figure 3.** (a) Difference of the AOD from GUVis - MIC datasets, plotted versus their mean. (b) Diffence of AOD from GUVisE - MIC datasets, plotted versus their mean. Blue and orange dots indicate maritime and desert dust respectively. This data is considered as valid, while red dots are flagged as outliers (which exceed the uncertainty estimate in at least one of the considered spectral channels). The black lines indicate the bias and the upper and lower limit of agreement (LOA), which should contain 95% of data points (see Sect. 3.3). The gray–shaded areas indicate the uncertainty estimate (95% confidence limit) of bias and LOA.





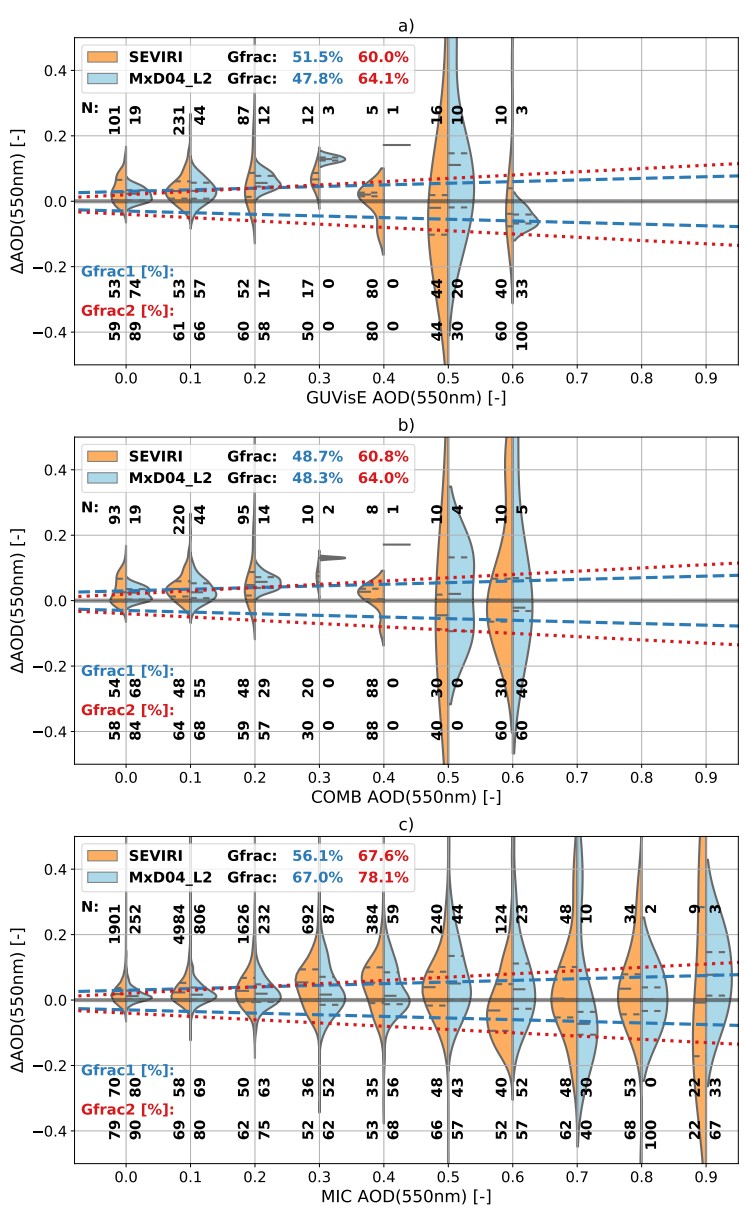

**Figure 4.** Comparison of AOD at 550 nm from the (a) GUVisE, (b) COMB, and (c) MIC reference datasets versus the SEVIRI and MODIS AOD products. The two sided violin-plots indicate the distribution of the difference for bins of 0.1 in AOD. The blue dashed and red dotted lines indicate the expected error limits for the MODIS AOD products. It is expected that at least 67% of data points fall into the expected error limits. Gfrac1 and Gfrac2 are the actual percentage of data points lying within the error limits, calculated for each bin, and as total for both the SEVIRI and MODIS AOD products.

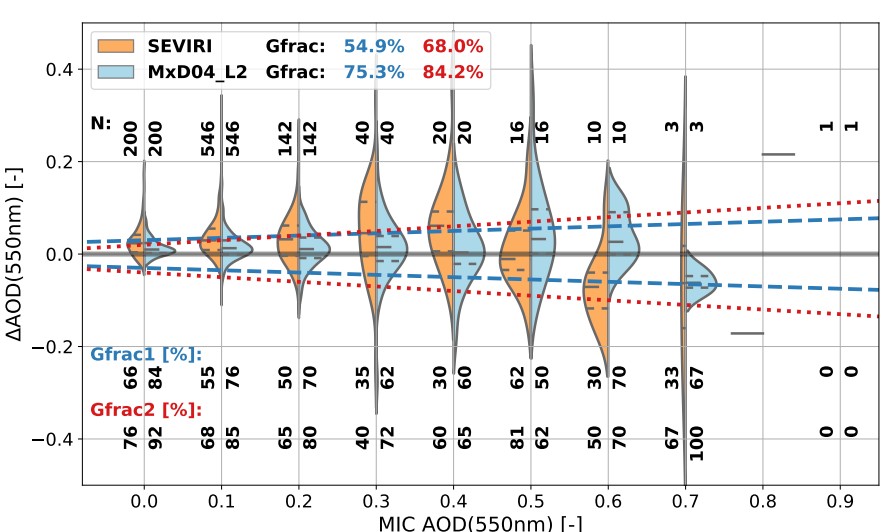

**Figure 5.** Same as Fig 4, but only using MIC as reference dataset. As additional constraint, availability of data from both the SEVIRI and MODIS datasets is required.





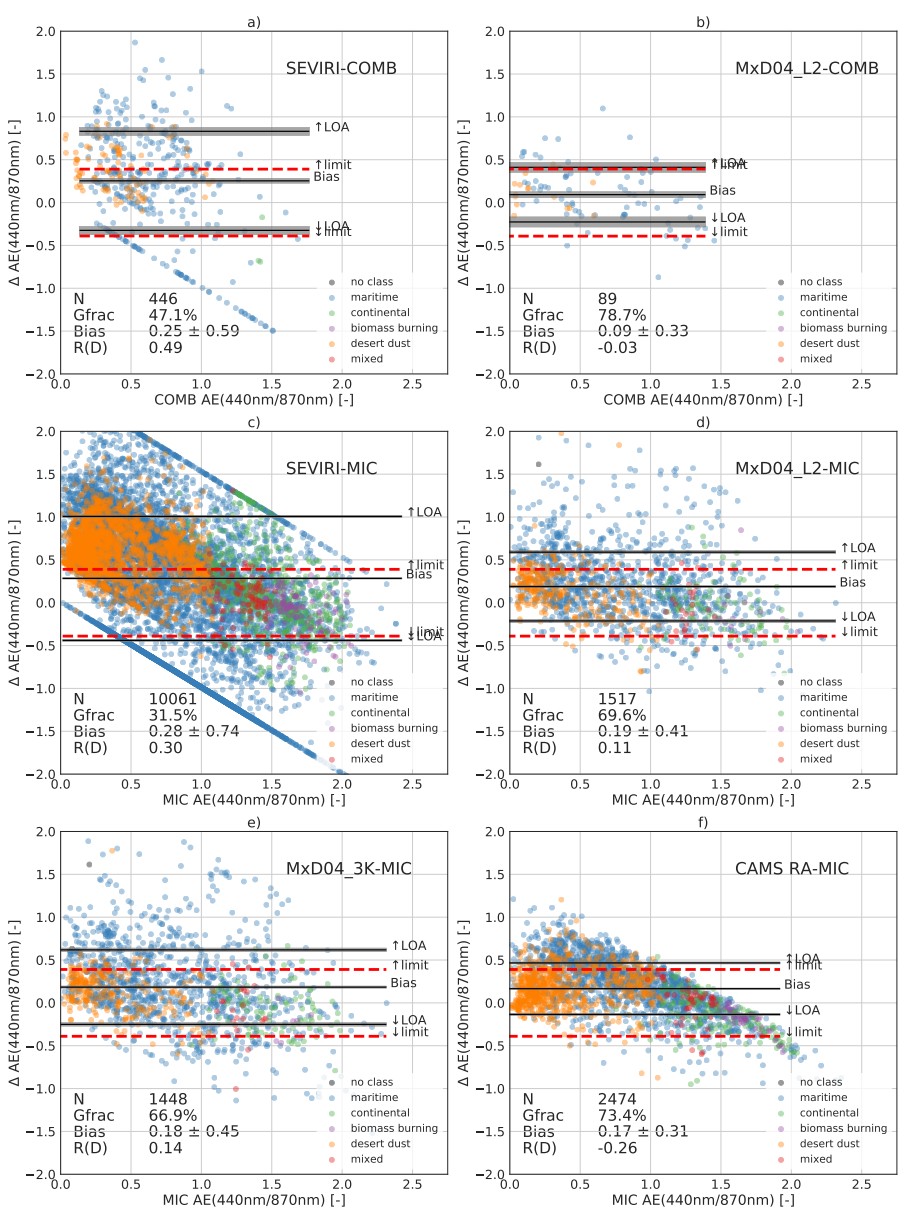

**Figure 6.** Comparison of AE from the COMB or MIC reference datasets versus the SEVIRI, MxD04 and CAMS RA datasets. The aerosol type classified with the reference dataset is indicated by the color of each point. Red dashed lines indicate the estimated error limits for AE (±0.4) of the MODIS products (Chu, 2002). 67% of AE data points are expected to fall into these limits. LOA (outer black lines) are based on 67% confidence intervals. The bias is given by the middle black line, and is calculated as the mean of the difference. The statistics state the number of measurements (N), percentage of data within expected error limits (Gfrac), bias ± LOA, and correlation of the difference (R(D)).





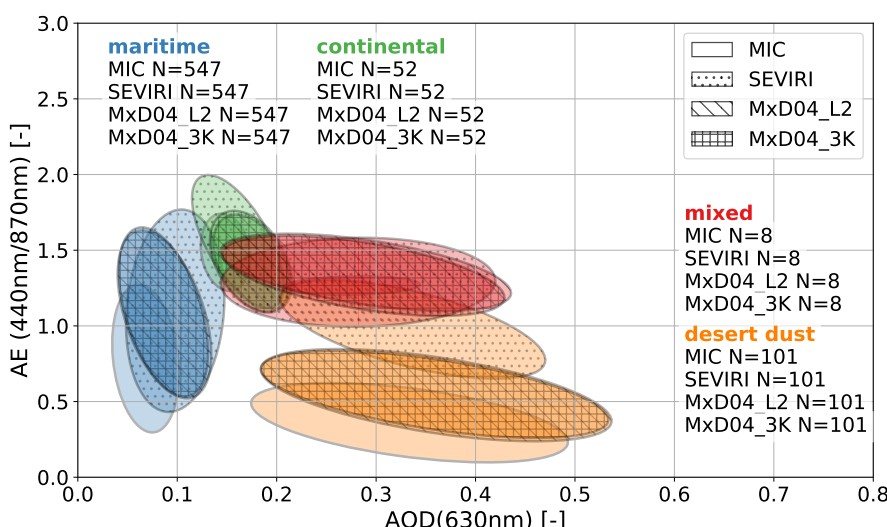

**Figure 7.** Comparison of AE calculated from AOD at the wavelengths of 630 nm and 810 nm versus AOD at 630 nm, calculated from the Microtops, SEVIRI and MODIS products. Simultaneous data availability from satellites and MIC is required, so that each instrumental data points has a corresponding counterpart from the other instruments. The data points are grouped by aerosol type (classified with MIC), and visualized as covariance ellipsoids for a 67 % confidence interval.



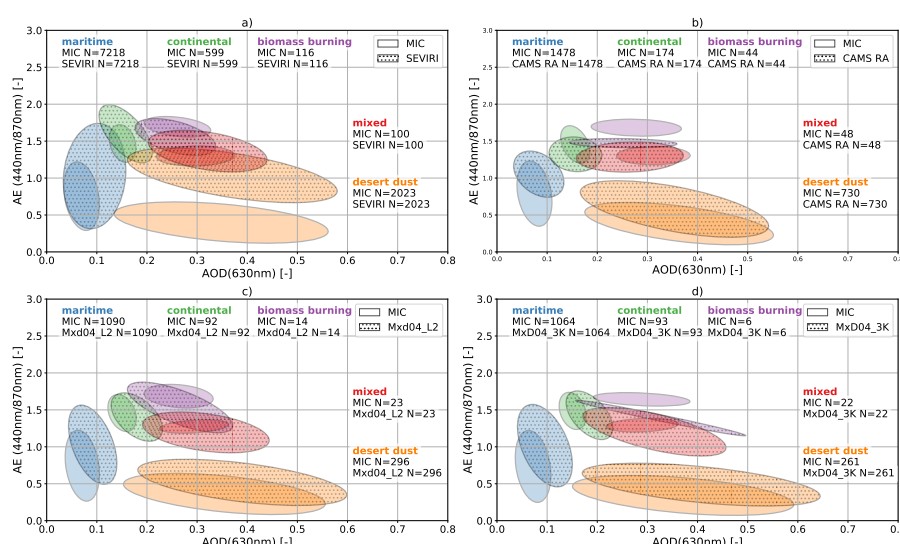

**Figure 8.** Same as Fig. 7, but the requirement simultaneous data availability from all data points was dropped, and the figure also CAMS RA data points.

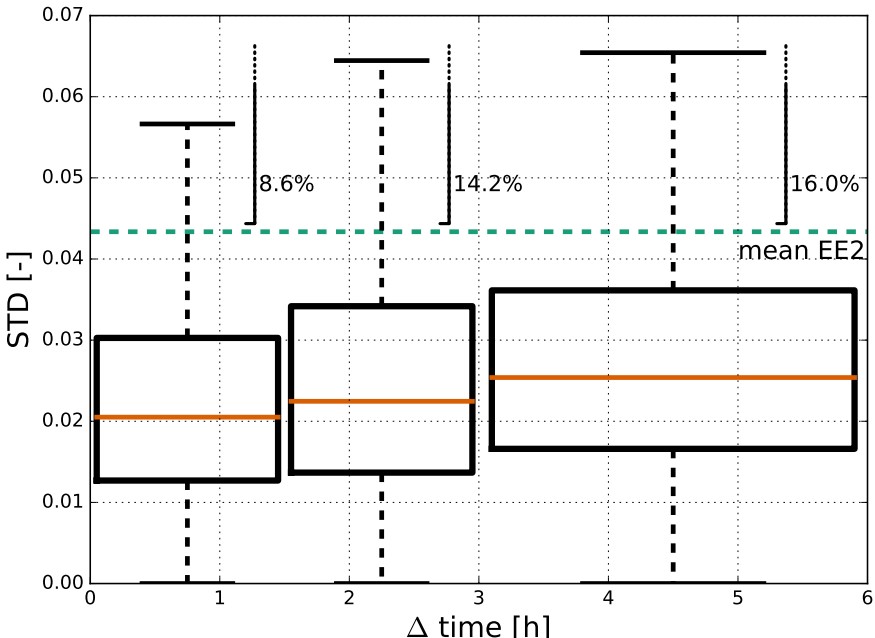

**Figure 9.** Temporal variation of AOD expressed as the standard deviation (STD) of SEVIRI AOD between MODIS overpasses, as a function of time lag between retrievals ($\Delta$ time). The mean of the expected error limits (EE2) of MODIS $[+(0.04 + 0.1\,\mathrm{AOD}), -(0.02 + 0.1\,\mathrm{AOD})]$ are calculated from the mean SEVIRI AOD of all datapoints and shown as the green dashed line. A variation in AOD can be considered significant, if the magnitude of STD exceeds the error limits. The precentage of significant situations are denoted for each time interval.



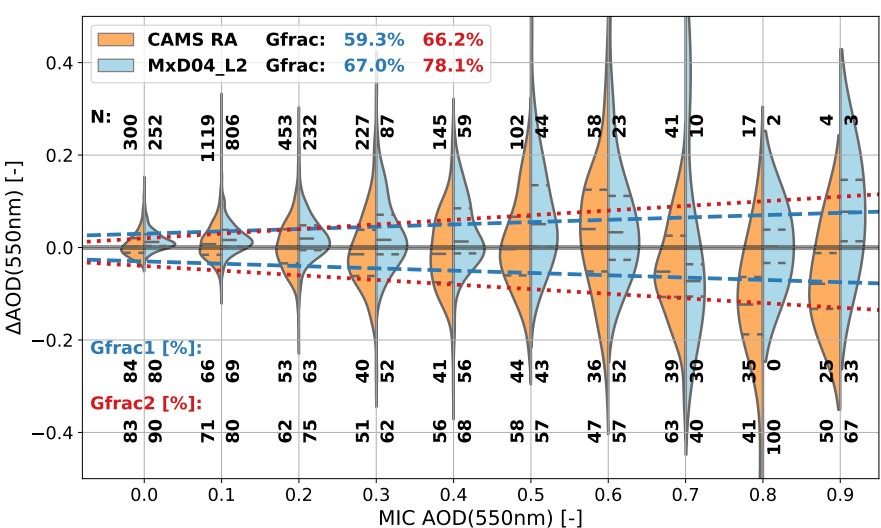

**Figure 10.** Same as Fig 4 but comparing CAMS RA and MODIS *MxD04_L2* AOD to MIC AOD as reference.





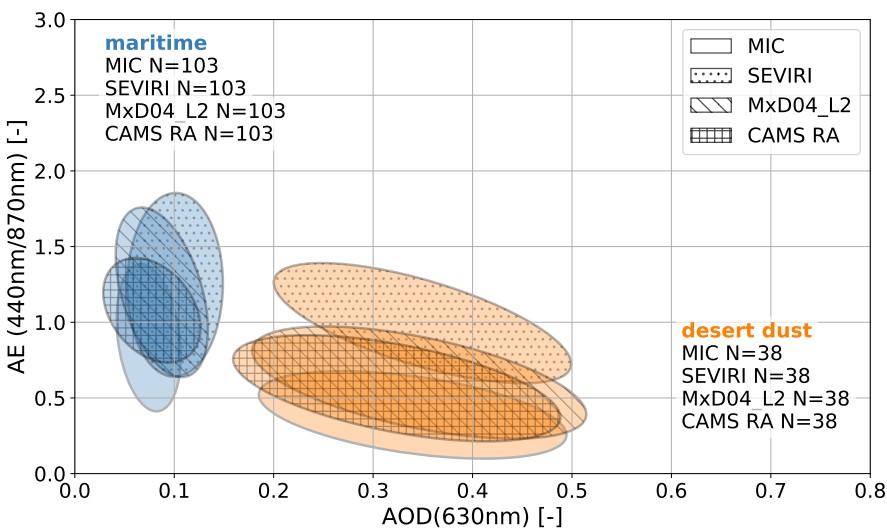

**Figure 11.** Same as Fig. 7, but comparing CAMS RA AOD instead of *MxD04_3K* AOD.





## Appendix A: GUVis processing update

In this appendix, the improvements of the GUVis processing algorithm of the shadowband sweep irradiance data is described.

The processing of the sweep time-series of the GUVis instrument is required to extract the global, direct and diffuse components from the measured spectral irradiance components. As described in detail in Witthuhn et al. (2017), the accurate estimation of the blocked diffuse irradiance while the direct sun is also blocked by the shadowband (i.e. the shadow of the band falls onto the detector) during the sweep is a fundamental challenge which has to be solved by the processing algorithm.

Figure A1 and A2 illustrate an idealized and a measured shadowband sweep together with the processing algorithm. Between shadowband sweeps, the band is stowed in a parking position out of sight of the hemispheric field of view of the sensor, so the global irradiance ($F_{glo}$) is observed by the radiometer (periods (a) and (g) in the figures). The amount of blocked diffuse irradiance can be directly inferred from the sweep data while the sensor is not shaded from direct sunlight by the band, by considering the reduction in measured irradiance (periods (b) and (f) in the figures). While the sun is partially or completely

blocked by the band due to the shadow falling onto the sensor (periods (c) to (e) in the figures), the reduction of the irradiance recorded by the shadowband compared to the global irradiance consists both of the blocked direct irradiance component ($F_{dir}$), plus a blocked fraction of the diffuse irradiance ($F_{dif,b}$). It is necessary to separate both parts in order to be able to calculate $F_{dir}$. The relation of the irradiance components is as follows:

$$F_{glo} = F_{dir} + F_{dif}, \tag{A1}$$

with $F_{dif}$ being the diffuse irradiance component which is partially blocked by the shadowband during the sweep, and can be separated into a blocked ($F_{dif,d}$) and a non-blocked part contributing to the observations ($F_{dif,o}$):

$$F_{dif} = F_{dif,b} + F_{dif,o}. \tag{A2}$$

As $F_{dif,b}$ cannot be inferred directly from the measurement, it is estimated by linear extrapolation of the measured irradiance data during the sweep while the sun is not blocked by the shadowband (periods (b) and (f)). As the 30 samples before and after

the shadow of the band transitions across the sensor are used, accurate knowledge of the time when the shadow starts to shade the sensor is required (termed point of contact from here on). The identification of the point of contact is accomplished in our processing algorithm by considering the slope of the measured irradiance data using empirical thresholds. It has to be realized that the change of slope before and after reaching the points of contact (thus at the transition from (b) to (c)) depends strongly on the present atmospheric situation, shape of the circum-solar radiation and shadowband geometry. In particular, this change

is not as sharp as indicated by Fig. A1, but shows a smooth transition as visible in Fig. A2.

The GUVis processing algorithm has received a substantial update compared to the version introduced in Witthuhn et al. (2017) to address several shortcomings. The following improvements were made:

(i) The identification of the point of contact is done by considering the slope of the measured irradiance during the complete shadowband sweep. Since the measured irradiance drops sharply once the sensor is partially shaded by the shadowband, a

threshold can be used for the slope to identify the point of contact. This threshold was choosen by using a constant absolute value in the old processing, which sometimes resulted in an inconsistent identification of the point of contact, in particular





during low sun or high AOD situations. In the revised processing, a relative threshold is used, which is calculated relative to the difference of measured global irradiance and the minimum measured irradiance of the sweep. This leads to a more reliable identification of the point of contact, as well as less scatter of the irradiance components of successive sweeps and during the

daily cycle.

(ii) The measured irradiance during one sweep sometimes contains high-frequency variations of the irradiance, e.g. caused by small clouds or the smoke plume of the ship. Affected sweeps are identified by a pre-processing filter and excluded from further processing. The pre-processing filter is applied by calculating the variance for the selected interpolation data (30 data points before and after the points of contact). The variance is compared to an fixed threshold value of $0.002^2$. In the old processing,

if the threshold was exceeded either using the data before or after the points of contact, the sweep is dropped completely. For the updated processing algorithm, if the threshold is exceeded, the data point with the largest deviation are removed from the interpolation. Data points are removed either until the variance criterium is met and the processing continues, or the number of data points used for interpolation is less than 21, in which case the sweep is excluded.

(iii) As mentioned before, the BioSHADE accessory for the GUVis utilizes a broad shadowband with a shading angle of

about $15°$. The shading angle is comparable to the field of view of a sunphotometer, which has a field of view of about $2.5°$ for the Microtops instrument (Porter et al., 2001). Comparing AOD based on the GUVis and Microtops, the difference in the field of view will lead to an underestimation of AOD retrieved with the GUVis, as has been reported also for the multi-filter rotating shadowband radiomeer (MFRSR) in a comparison to AERONET sunphotometer in the study of di Sarra et al. (2015). The underestimation is attributable to the forward scattering contribution of aerosol scattering as investigated by Russell (2004).

The underestimation of AOD is substantial for large shadowband shading angles, and especially large for aerosol particles with strong forward scattering (e.g. desert dust) (Ge et al., 2011). To at least partly compensate for this effect, an offset has been introduced in the linear extrapolation of the blocked diffuse irradiance. The offset depends on the slope of the interpolation data before and after the points of contact, which is steeper during stronger forward scattering aerosol situations. The offset is calculated from the difference of the irradiance at the point of contact, and the extrapolated irradiance using the interpolation

data at the time of the point of contact. Therefore, the offset is larger during strong forward scattering aerosol situations, since the increase in forward scattering leads to a steeper drop before the point of contact. Thus, the offset compensates for the underestimation of AOD due to aerosol forward scattering.

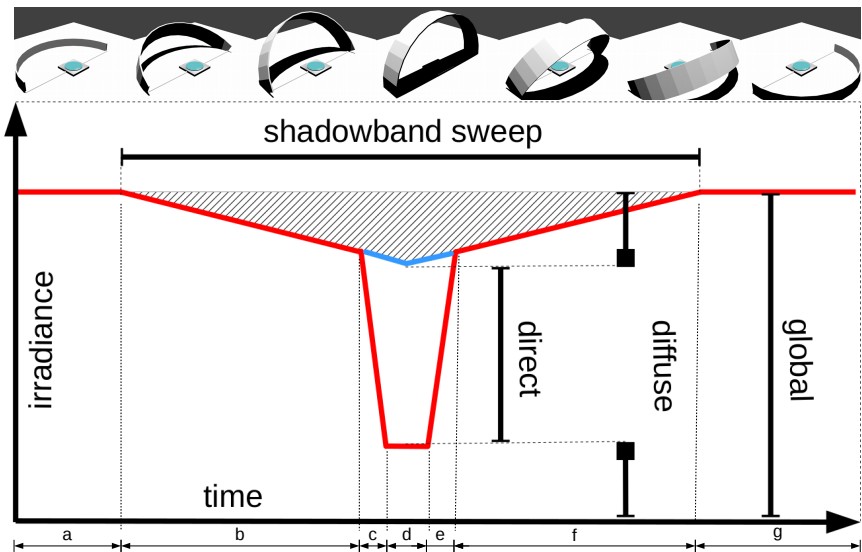

**Figure A1.** Schematic illustration of a shadowband sweep measured by the GUVis shadowband radiometer. The red line indicates the mearued irradiance. The figures on top illustrate the shadowband position relative to the sensor during the sweep. The hatched area indicates the diffuse irradiance blocked from the sensor during the sweep. The blue line indicates the unknown blocked diffuse irradiance when the direct irradiance is at least partially blocked by the shadowband. It has to be estimated by the processing algorithm in order to accurately estimate the direct irradiance. The letters a to g indicate different periods during the sweep as follows: (a, g) shadowband in parking position, out of sight of the hemispheric field of view of the sensor; the measured irradiance corresponds to the global irradiance. (b, f) the shadowband is moving, but the direct irradiance of the sun is not blocked from the sensor. (c, e) the direct irradiance is partially blocked by the shadowband, as the band shades the sensor. (d) the direct irradiance is completely blocked by the shadowband.

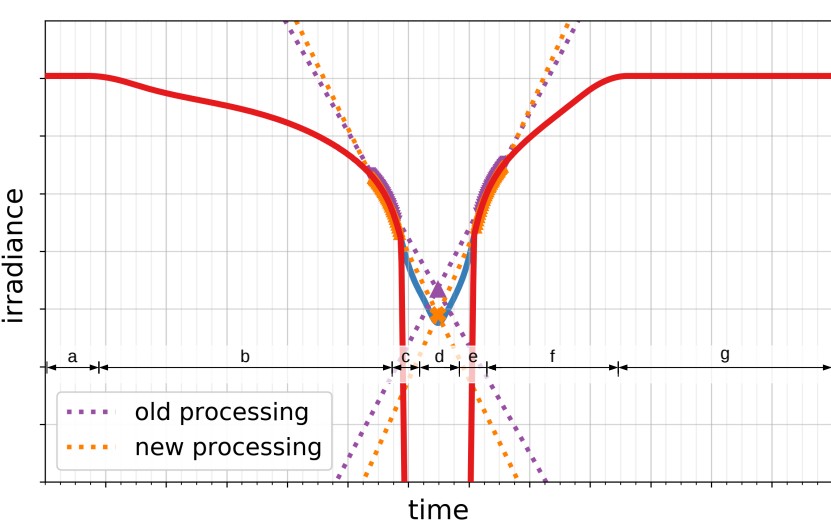

**Figure A2.** Like Fig. A1, but with a measured irradiance time series (red line) of a shadowband sweep. The blue line indicates the unknown amount of blocked diffuse irradiance, which has to be estimated by the processing algorithm. The dotted lines indicate the extrapolation lines of the old (purple) and new (orange) processing algorithm to estimate the blocked diffuse irradiance.

**Figure A3.** Same as Fig 4, but for AOD at 630 nm, which is a native spectral channel of the SEVIRI instrument.

4000
4000




**Table A1.** Like Table 5 but for maritime aerosol only.

| Reference | Instrument | N | channel | R | linear regression | R(D) | bias ± LOA | G1 | G2 |
|---|---|---|---|---|---|---|---|---|---|
| - | - | - | nm | - | - | - | - | % | % |
| GUVisE | CAMS RA | 80 | 550 | 0.73 | Y = 0.55 X+0.02 | 0.39 | 0.03±0.07 | 55 | 71 |
| | | | 630 | 0.74 | Y = 0.59 X+0.02 | 0.31 | 0.02±0.06 | 70 | 79 |
| | SEVIRI | 343 | 550 | 0.69 | Y = 0.43 X+0.03 | 0.56 | 0.04±0.09 | 52 | 62 |
| | | | 630 | 0.66 | Y = 0.44 X+0.03 | 0.49 | 0.03±0.09 | 61 | 68 |
| | MxD04_3K | 50 | 550 | 0.86 | Y = 0.60 X+0.01 | 0.58 | 0.03±0.06 | 62 | 78 |
| | | | 630 | 0.85 | Y = 0.58 X+0.01 | 0.60 | 0.03±0.06 | 72 | 82 |
| | MxD04_L2 | 49 | 550 | 0.85 | Y = 0.61 X+0.01 | 0.53 | 0.03±0.06 | 59 | 76 |
| | | | 630 | 0.86 | Y = 0.60 X+0.02 | 0.59 | 0.03±0.06 | 63 | 76 |
| COMB | CAMS RA | 83 | 550 | 0.77 | Y = 0.59 X+0.02 | 0.38 | 0.03±0.06 | 59 | 75 |
| | | | 630 | 0.79 | Y = 0.63 X+0.02 | 0.33 | 0.02±0.06 | 70 | 82 |
| | SEVIRI | 350 | 550 | 0.73 | Y = 0.46 X+0.02 | 0.57 | 0.04±0.09 | 49 | 62 |
| | | | 630 | 0.71 | Y = 0.49 X+0.02 | 0.48 | 0.03±0.08 | 60 | 71 |
| | MxD04_3K | 52 | 550 | 0.92 | Y = 0.67 X+0.01 | 0.63 | 0.03±0.05 | 67 | 81 |
| | | | 630 | 0.91 | Y = 0.66 X+0.01 | 0.64 | 0.02±0.05 | 75 | 87 |
| | MxD04_L2 | 52 | 550 | 0.90 | Y = 0.66 X+0.01 | 0.57 | 0.03±0.06 | 58 | 73 |
| | | | 630 | 0.91 | Y = 0.66 X+0.01 | 0.60 | 0.03±0.05 | 63 | 85 |
| MIC | CAMS RA | 1476 | 550 | 0.72 | Y = 0.48 X+0.04 | 0.52 | 0.01±0.08 | 69 | 73 |
| | | | 630 | 0.72 | Y = 0.50 X+0.03 | 0.47 | 0.01±0.07 | 74 | 78 |
| | SEVIRI | 7217 | 550 | 0.63 | Y = 0.34 X+0.04 | 0.64 | 0.03±0.11 | 60 | 71 |
| | | | 630 | 0.59 | Y = 0.33 X+0.04 | 0.60 | 0.03±0.10 | 66 | 76 |
| | MxD04_3K | 526 | 550 | 0.74 | Y = 0.56 X+0.02 | 0.40 | 0.02±0.07 | 68 | 81 |
| | | | 630 | 0.73 | Y = 0.53 X+0.02 | 0.41 | 0.02±0.07 | 71 | 80 |
| | MxD04_L2 | 660 | 550 | 0.74 | Y = 0.55 X+0.02 | 0.41 | 0.02±0.07 | 70 | 81 |
| | | | 630 | 0.74 | Y = 0.55 X+0.02 | 0.43 | 0.02±0.06 | 71 | 82 |





**Table A2.** Like table 5 but for desert dust aerosol only.

| Reference | Instrument | N | channel | R | linear regression | R(D) | bias $\pm$ LOA | G1 | G2 |
|---|---|---|---|---|---|---|---|---|---|
| - | - | - | nm | - | - | - | - | % | % |
| GUVisE | CAMS RA | 23 | 550 | 0.75 | Y = 0.75 X+0.06 | 0.02 | 0.02$\pm$0.18 | 43 | 74 |
| | | | 630 | 0.77 | Y = 0.73 X+0.07 | 0.09 | 0.01$\pm$0.17 | 52 | 70 |
| | SEVIRI | 87 | 550 | 0.57 | Y = 0.32 X+0.18 | 0.59 | 0.09$\pm$0.40 | 43 | 54 |
| | | | 630 | 0.55 | Y = 0.31 X+0.18 | 0.59 | 0.07$\pm$0.39 | 48 | 60 |
| | *MxD04_3K* | 8 | 550 | 0.90 | Y = 0.95 X-0.04 | -0.13 | 0.07$\pm$0.15 | 25 | 50 |
| | | | 630 | 0.91 | Y = 0.98 X-0.04 | -0.16 | 0.05$\pm$0.14 | 50 | 62 |
| | *MxD04_L2* | 12 | 550 | 0.76 | Y = 0.47 X+0.16 | 0.62 | 0.13$\pm$0.33 | 17 | 42 |
| | | | 630 | 0.76 | Y = 0.47 X+0.17 | 0.61 | 0.12$\pm$0.34 | 25 | 42 |
| COMB | CAMS RA | 23 | 550 | 0.79 | Y = 0.77 X+0.05 | 0.03 | 0.03$\pm$0.17 | 48 | 83 |
| | | | 630 | 0.80 | Y = 0.75 X+0.07 | 0.11 | 0.01$\pm$0.16 | 61 | 83 |
| | SEVIRI | 86 | 550 | 0.60 | Y = 0.34 X+0.17 | 0.60 | 0.09$\pm$0.39 | 42 | 52 |
| | | | 630 | 0.58 | Y = 0.33 X+0.17 | 0.59 | 0.07$\pm$0.38 | 49 | 62 |
| | *MxD04_3K* | 6 | 550 | 0.70 | Y = 0.80 X+0.07 | -0.19 | 0.05$\pm$0.18 | 50 | 67 |
| | | | 630 | 0.75 | Y = 0.87 X+0.04 | -0.22 | 0.03$\pm$0.16 | 67 | 83 |
| | *MxD04_L2* | 10 | 550 | 0.69 | Y = 0.41 X+0.21 | 0.60 | 0.13$\pm$0.37 | 30 | 40 |
| | | | 630 | 0.69 | Y = 0.41 X+0.21 | 0.61 | 0.12$\pm$0.38 | 40 | 40 |
| MIC | CAMS RA | 730 | 550 | 0.85 | Y = 0.83 X+0.06 | 0.03 | 0.00$\pm$0.20 | 46 | 59 |
| | | | 630 | 0.86 | Y = 0.86 X+0.06 | -0.00 | -0.01$\pm$0.20 | 47 | 60 |
| | SEVIRI | 2023 | 550 | 0.83 | Y = 0.80 X+0.03 | 0.06 | 0.05$\pm$0.25 | 38 | 54 |
| | | | 630 | 0.83 | Y = 0.84 X+0.04 | -0.03 | 0.02$\pm$0.24 | 49 | 62 |
| | *MxD04_3K* | 156 | 550 | 0.92 | Y = 0.84 X+0.03 | 0.24 | 0.05$\pm$0.21 | 56 | 67 |
| | | | 630 | 0.92 | Y = 0.83 X+0.03 | 0.25 | 0.04$\pm$0.21 | 54 | 69 |
| | *MxD04_L2* | 223 | 550 | 0.86 | Y = 0.81 X+0.04 | 0.13 | 0.05$\pm$0.23 | 52 | 62 |
| | | | 630 | 0.86 | Y = 0.81 X+0.04 | 0.10 | 0.04$\pm$0.23 | 52 | 65 |