# Peer review of "Evaluation of satellite-based aerosol datasets and the CAMS reanalysis over ocean utilizing shipborne reference observations"

_Atmospheric Measurement Techniques, 2019_

## Referee Comment (RC1) · Anonymous Referee #4 · 25 Oct 2019

The paper discusses aerosol optical depth observations from ships. The calibration and retrieval techniques for a shadowband radiometer are revised to improve agreement with simultaneous sun-photometer measurements, removing a fractional bias. The new dataset is used to evaluate products from the MODIS and SEVIRI sensors. The former is, unsurprisingly, found to be more precise, but both overestimate the Angstrom exponent, with SEVIRI being out by up to an order of magnitude. The CAMS aerosol reanalysis is similarly evaluated, finding it eliminates much of the bias in the underlying MODIS data.

The paper is suitable for publication in this journal. Current satellite retrievals show

signficant disagreement as to the average AOD over remote ocean and the data provided by this study should be invaliable in resolving that discrepancy. I hope the authors can place their data in a publicy available repository — I am eager to use it in the evaluation of my own satellite products!

I have a few minor few comments that warrant the authors' attention:

- Though I am fond of your analysis in Fig. 9, I disagree with the scope of your conclusions with respect to the information provided by SEVIRI.

    – By using the name of the sensor to refer to a specific dataset, you imply that your conclusions apply to *all* SEVIRI aerosol products. If one could produce a more accurate aerosol product from SEVIRI, that would provide useful information. You don't present sufficient evidence that all possible SEVIRI products provide minimal additional information.

    – Your wording is fairly definitive: 'only offer minor benefits compared to the use of polar-orbiting satellite platforms'. The circumstances where aerosol changes rapidly, such as plumes or the passing of a frontal system, are scientifically very interesting and exactly the sort of circumstances that geostationary imagery are absolutely vital in understanding. Geostationary observations might not add much to our understanding of the climatology of AOD, but this doesn't mean that they only provide minor benefits; they provide targetted benefits.

    – You only evaluate the representivity of observations between the two MODIS overpasses. This omits the periods of boundary layer growth and collapse in the morning and evening, which current polar orbiting satellites do not observe.

    There is no need to perform additional analysis, but your conclusions should be reworded to be clearer about their breadth.

- I am surprised by the repeated implications that laboratory lamp calibration is inadequate. Calibration in a controlled environment is usually held up as the gold standard of observational atmospheric science. Did the authors mean to imply that such calibrations are insufficient to produce a scientifically valid product (e.g. 'limited accuracy')? I would find that difficult to believe.

  I suspect what was meant is that there is an intrinsic difference between what a sun-photometer and shadowband radiometer measure. That limits the extent to which they could ever agree without additional correction methods, such as those outlined in this paper.

- I'm not convinced by the explanation in §4.2 of the narrow, highly biased observations of AOD $\simeq 0.3$ in Fig. 4 as I can't see why the choice of aerosol type would only affect one range of AODs. Are there an anomalously small number of collocations in those conditions or are they clustered in a small area? If you loosen your quality control conditions, does the distribution more closely resemble the typical behaviour?

- Are the outliers identified at page 11 (line 344) excluded from further analysis? That seems satistically suspect, as we expect large deviations to occur occassionally by random chance.

- I found it strange that Fig. 1 implies that only maritime and dust aerosols were observed while Fig. 2 shows that mixed and continental were ocassionally encountered as well.

- Fig. 7 is a compelling way to present the limitations in the retrieval of Angstrom exponent. In a future paper, it would be interesting to see a study of the implications of your results on the Aerosol Index, which is widely used as a proxy for cloud condensation nuclei in studies of aerosol-cloud interactions.

[Figure]

- At L540, is an increase from 0.90 to 0.92 really evidence of a 'clearly superior' product? That doesn't seem a particularly significant shift.

- On page 21, the EarthCARE lidar isn't itself that 'unique'. It's unique that said lidar is being flown collocated with an imager and radar.

- Your discussion about CAMS in §4.3 would be improved if you mention that the inputs to a renalysis system must be bias corrected before input to ensure a stable assimilation of the data. Hence, the reduction in bias is to be expected (but remains evidence of the utility of the CAMS product).

- In Fig. 3, is the sharp transition from maritime to dust aerosols at 0.18 a true feature of your data (which would be concerning) or a feature of plotting the orange points over the blue ones? If the latter, perhaps add some transparency, so the transition is easier to see?

- In point (ii) of the appendix, you change the method for filtering perturbed observations. What motivated this choice? In undergraduate labs, I teach my students to throw out any observation for which the method was suspect as making a correction involves a number of assumptions. Why do you feel the need to keep some corrupted observations here?

I also include some technical comments and corrections. P1L2 means line 2 of page 1.

P1L16 similar performances for both datasets

P2L39 e.g. from ships are available

P2L57 Does 'earth' need to be capitalized?

P3L79 complex, non-spherical shape

P4L109 These findings are understood in the context of the results found for the SEVIRI aerosol product to observe

P4L119 Add a space after the comma.

P4L124 are publicly available

P5L133 reference: the sunphotometer

P8L251 of $\pm 30$ min have been used

P9L257 distance angle less than $0.2°$

P9L272 the analyses are

P10L303 Perhaps add 'to ensure the' after 'compensated for'? It means something slightly different but is what I think you meant to say here.

P12L373 Add a space after 'Table'.

P16L513 The wrong style of reference is used.

P16L518 I don't know what you meant to say by 'follow up'.

P20L627 has channels only at

P20L640 products can provide a

P21L672 with the next few years collocated with an imager and radar.

Tab.5 collocated data points. Listed

Fig.1 The aerosol classification method

Fig.8 requirement for simultaneous . . . figure also shows CAMS RA

---

## Referee Comment (RC2) · Anonymous Referee #3 · 29 Oct 2019

This paper aims to evaluate the satellite (MODIS, SEVIRI) and reanalysis (CAMS) retrievals of AOD and Angstrom exponent over ocean by comparing them with moving ship-borne observations using Microtops sunphotometers and multi-spectral shadowband radiometer GUVis-3511 during several cruises in the Atlantic Ocean. The results are re-evaluated for defined aerosol types, mostly maritime and desert dust. Overall, the manuscript is well written and organized, although some improvements may be attained in the discussion of the results. However, the manuscript is rather long enough and some parts may be significantly shortened without any effect in the general discussion and importance of the results, since there are several repetitions throughout the manuscript. There is a rather long discussion of aerosol direct and indirect effects

in the beginning of the Introduction that is beyond the scope of the current research. I understand that authors initially discuss the role of aerosols on global climate and the necessity of accurate measurements of them, in a way to reduce the uncertainty in their climate response, but this part may be shortened in one paragraph (for example the first three paragraphs could be shortened and merged into one). Although a very analytic description is provided for satellite products, GUVis measurements, collocation procedures and so on, there is luck of information about uncertainties in the Microtops-II AOD retrievals, which may be high if the instrument is not exactly oriented to the sun's disk. Usually, 3-5 sets of measurements are taken from Microtops in order to select the best one via techniques described in previous papers (e.g. Sharma et al., 2014, Aerosol Air Quality Research; Tiwari et al. 2018, Environ. Science Pollution Res.). In addition, during the W-ICARB cruise campaign over the Bay of Bengal, there was a comparison between Microtops-II and MODIS AODs revealing a very good agreement between them, which may be mentioned in the paper and discussed against the current findings (Kharol et al., 2011; Annales Geophysicae). Section 4.2 is composed of numerous relatively short paragraphs, whose meanings are not so distinguishable. This creates some difficulties in reading and understanding exactly the major issue (spirit) of each paragraph. Taking also into account the several repetitions, this becomes more problematic. What I recommend is to merge the paragraphs into longer ones discussing a define issue, for example results of the presented figures and tables and/or discussion on these results. Special care should be taken throughout the manuscript on avoiding several repetitions. Some of these are emphasized below. Minor comments/corrections Line 51: Levy et al. (2013) estimated . . . Line 73: Double use of "system" at the end of this sentence does not make good sense and should be revised. Lines 205-206. I recommend to remove this sentence from this part of the manuscript. In case the reader would expect a better accuracy from MODIS, what's the reason to read the results of this study? Lines 362-364 and lines 380-381. These sentences are just a repetition and one should be removed. Line 447. . . .is presented here. Line 476-480. Since the data . . . MIC data. Such statements have been repeated

several times in the manuscript and may be removed or significantly shortened. Line 487. This sentence, even rephrased has been stated several times in the manuscript. 493-494. Since the MODIS . . . accurately. Similarly, this has been stated several times in the manuscript. Line 541. This emphasizes. . . Line 562. A slight increase. . . Line 589. This is a similar statement as in line 571.

---

## Referee Comment (RC3) · Stefan Kinne (Referee) · 10 Nov 2019

Review of paper:

Evaluation of satellite-based aerosol datasets and the CAMS reanalysis over ocean utilizing shipborne reference observations by J. Witthuhn et al.

Positives - needed demonstration of (MAN) reference data over ocean - efforts to evaluate more than just the aerosol column amount (AOD)

Concerns - shadowband (GUV) data seem to lack processing maturity to serve as reference - comparisons to two satellite data and one assimilation lack interpretation

[Figure]

- aerosol typing via AOD and Angstrom should be redone – more tailored of oceanic reg.

General comments

The paper evaluates the aerosol properties over oceans of two satellite retrievals (MODIS, SEVIRI) and one modeling effort (CAMS) that assimilated MODIS data. The evaluation is based on matches (in time and location) to shipborne samples of the direct solar attenuation and of solar scattering during multiple latitudinal Atlantic crossings with the German POLARSTERN research vessel. The accuracy of (handheld MICRO-TOPS) direct solar attenuation data is out of question and serves as the main reference in the evaluation. In contrast, the usefulness of the complementary shadowband radiometer (GUV) appears more limited (completely cloud-free situations are needed) and the retrieval algorithm to retrieve the direct solar spectral irradiance component has not reached the needed maturity (still many ad hoc adjustments are needed to match solar attenuations of simultaneous MICROTOPS measurements). Thus recent efforts to improve the shadow-band references (GUV, GUVisE) are more something for the Appendix. Focusing on the sunphotometer data there are many nice aspects addressed (although other satellite data sets and models could be included – possibly in an additional paper). I am not so happy about the chosen aerosol classification (especially since many source types like biomass are unlikely to be observed over oceans). Based on the sunphotometer data for AOD and Angstrom value (use AE only if AOD550 is larger than 0.15) I would separate into 4 categories (see comments to Figure 1) and work from there. I really like the plots that investigate biases are function of AOD. That said I am really disappointed that this detail is missed in the statistical tables, as common features of (1) AOD overestimate at lower AOD and (2) AOD underestimate at larger AOD lead the authors to claim via a linear fit 'overall good agreement' and 'low biases'. The paper is great contribution but I think it needs a major revision prior to a publication.

Detailed comments
1/17 if AE is overestimated then offer an explanation such as that large mineral dust size-events are missed or that the compositional mix is constrained by model assumptions

1/21+ the radiative effect/forcing introduction (although I am not sure if it is needed) could/should be sharper: In terms of radiative forcing (that is the impact by anthropogenic aerosol at all-sky conditions at TOA) the indirect effect RFaci overall has larger contributions than the RFari. And anthropogenic aerosol impacts are primarily caused by extra smaller aerosol sizes so that (mostly natural components) of seasalt and dust which are mainly observed over oceans are almost irrelevant even for RFari. The impact of these two components on Raci (seasalt to increase CCN, dust to increase IN) has been demonstrated but the overall impact strength remains unclear (for more on radiative effects and forcing have a look at the MACv2 paper).

2/37+ if you mean CERES (broadband radiation) data then say so, even though then the link of aerosol retrievals (they make a compositional and size assumptions to yield AOD estimates) to broadband fluxes is not straight forward and need further rad.transfer modeling.

2/39 Exactly, this is why you want the ship data: to evaluate and to test (retrieval) model assumptions.

2/55 you may also consider to evaluate to (the newly re-processed) MISR data, to VIIRS data (designated as MODIS follow on), to SLSTR (the ATSR follow-on) data and to GRASP-type (e.g. MERIS) data.

2/76 A separate evaluation for fine (and mostly anthropogenic) and coarse-mode AOD (as offered for the MICROTOPS data via the SDA approach) - in place of an Angstrom parameter evaluation - would elevate a marine aerosol retrieval or modeling evaluation: Most global models and satellite retrieval models use bi-modal (by size) scheme, while Angstrom parameters depend spectral choice and becomes extremely noisy and probably meaningless at low AOD values in already one of the two spectral bands needed.

5/131 I would have used a different description of the Angstrom parameter : ANG = -ln(AOD1/AOD2) / ln(wavel1/wavel2) . . . as the negative spectral slope in ln/ln-space

5/138 The AOD shadow-band radiometer requires more actions (leveling corrections) and is more restrictive (completely cloud-free over the time of a scan).

6/180 The empirical correction is covered in detail but only applied to the shadowband (GUV) instrument (and not to the sunphotometer). Considering that the GUV match statistics is so much sparser then for the sunphotometer (with useful data evaluation data at this stage only through sunphotometer matches) so that in the end only sunphotometer data are used in the evaluation, I wonder if all that GUV detail (table and figures) is not better added to the Appendix. In terms of the forward scattering correction I am worried about a linear approach - which ignores a dependence on aerosol size (over oceans certainly at larger AOD) size.

6/179 COMB is a subset of GUV, where GUV outliers (inconsistent to MIC) removed - correct?

7/194 mention also the AOD (I assume 550) wavelength of MODIS

7/208 there are three major solar retrieval problems for dust outflow: (1) dust is solar spectrally flat (like clouds) and large AOD events may be removed as clouds → retrieved dust AOD too low (2) dust size is underestimated and with it the size related 'dust absorption' (with low SSA values) → retrieval AOD will be too high (3) non-sphericity has increased side-scatter so that a retrieval model with coarse-mode and fine-mode spheres will interpret the extra side-scatter artificially with extra fine-mode spheres → wrong composition and likely size underestimate (see under 2).

7/219 . . . but cloud coverage may be wrong in global modeling. . .aside that the AOD might be different in clear-sky and cloud-sky regions of modeling

7/226 in CAMS only AOD is assimilated and in its bias correction the compositional mixture of the forecast model is maintained . . . so the composition may get worse (in

comparison to a forecast without assimilations). By the way, the use of the (relatively sparse) ATSR data added little in a combined (ATSR & MODIS) assimilation - as the volume of MODIS data dominated.

8/228 the offered aerosol classification (based on AOD and ANG) is rather general and prone to failure if one of the two needed AOD values becomes low. For those events any ANG value becomes meaningless. And with respect to aerosol typing aerosol is always a mixture of many components. And unless surface winds are very calm, even marine aerosol has Angstrom parameters between 0.7 and 1.0. An ideal thing would be if simultaneous info on aerosol absorption would be available (as demonstrated in MACv2). To extract info an components, I would start with an AOD separation into fine-mode (r<0.5um) and coarse-mode (r>0.5um) contributions via the fine-mode AOD fraction (easily derived from the AOD spectral dependence) to separate seasalt/dust from pollution/wildfire aerosol (in a quantitative way!). The MAN data-base (where MICROTOPS data are stored) offers such derived (fine-mode-fraction) estimates!

8/230 the AE overestimate in CAMS can be just related to the improper component mixture (too much fine-mode contributions) in the forecast model, which an (total) AOD assimilation cannot fix .. and in case of AOD adjustment during the assimilation process will pull away from likely expectations (such that mix of assimilated products is often worse than that of the forecast model)

8/240 aerosol is always a mixture of many components and what the simple 'Toledano' scheme does is simply identifying a likely dominant component in a qualitative way. I disagree though that marine aerosol has Angstrom parameters larger than 1.2 unless there are contributions from pollution and or biomass burning. AE values larger than 1.2 at should be linked fine-mode aerosol with contribution from sulfate / nitrate (less absorption), pollution and aged biomass burning (medium absorption) and fresh biomass burning (highly absorption and AE >1.5).

8/245 all classes are 'mixed' . . . but in terms of the mixed category here, how useful is

a 'mixed' category if it could be a 10/90% or a 90/10% mixture?

9/260 for more events (I think) the <22km co-location requirement could be relaxed over ocean

9/262 why only so few matches between MIC CAMS-RA (as modeling has AOD data at any location and time)?

10/298 Well it is sad that GUV retrieved AOD are often inconsistent to (trusted) MI-CROTOPS data – as apparent more algorithm work is needed.

10/306 To achieve GNU vs MICROTOPS consistency a rather complex procedure is applied to GNU data based on comparisons to almost simultaneous MICROTOPS samples. Hereby the described method seems a bit arbitrary to me. To account for the forward scattering into the GNU 15deg field of view (isn't 15 deg just the width of the band so in effect the missed diffuse radiation is even larger than 15 deg?) a constant C is defined . . . while in reality this forward scattering contribution depends (at completely cloud-free conditions) on aerosol size (!) and multiple scattering potential (AOD, airmass) . . .so a constant seems a poor choice. And then there is even a second scaling factor for a temporal (do you mean sun-elevation?) correction!

12/376 a fixed bias of -0.02 (with the new processing) seems strange as I would expect a bias related to missed forward scattering to be function of AOD and size..? Actually Figure 3 shows such an AOD dependence of the bias. Generally: the entire correction explanations of 3.4 and of 4.1 (till line 400) including the comparisons of Table 4 are technical details which seem better suited for the Appendix as the focus should be on the comparison of satellite (and CAMS) data to trusted references.

12/383 . . . the forward scattering missed radiation problem is more an issue for larger size and AOD (e.g. especially relevant for larger mineral dust sizes). You actually show in Figure 3a that this bias is near zero at low AOD and stronger at larger (likely dust) AOD . . . in that sense it is misleading to talk about a -0.02 bias (but I said this before).

[Figure]

13/412 how was the SEVIRI AOD at 550nm derived (was the MICROTOPS Angstrom parameter used?)

13/417 The larges MODIS AOD outliers in the 0.3 bin vs GUVisE may be less meaningful since the sample number is fairly low. This raises the question, why the GUVisE data-set is so small and why this data-set only addresses selective dust and marine cases. Does it mean that the retrieval and correction method very frequently fails?

14/424 I do not see major differences between 4a and 4b unless low (and less meaningful) statistics is considered. the low GUV sample number remains a mystery to me (a few 'marine' and 'dust' cases).

14/435 you could compare if the SEVIRI AOD550 data would be different if the MICROTOPS 440/870 Angstrom had been used instead (also the SEVIRI band are spectrally much broader with some water vapor contamination especially in 810nm band).

14/439 I would say Figure 5 shows (opposite to your assessment) that SEVIRI AOD is usually higher than MODIS (at low to median AOD) and that at high AOD SEVIRI likely has a low bias, as (large AOD dust events with no solar spectral dependence) are probably removed during cloud screening.

14/445 Can you say this when samples of the two different MODIS products are different?

15/471 Why did you apply SEVIRI Angstrom and not MICROTOPS Angstrom data to interpolate from 630 to 550nm?

15/474 Why would I want to use COMB if I have MIC (and much better statistics)?

15/478 the MICROTOPS 440/870 Angstrom is more reliable than the 670/870 Angstrom. (The label in Figure 7 says 440/870 . . . but the captions say differently?). Anyway, you do make sure not to include AE data if AOD at 870nm is below 0.05 . . . otherwise derived Angstrom parameters are likely close to 'garbage'.

16/489 I agree that at low AOD satellite tend to overestimate AOD due to a more relaxed or insufficient cloud-screening (sometime there are many low clouds which are hardly visible)

16/495 the CAMS forecast model (and thus also MODIS AOD data assimilations) have a too high fine-mode fraction for the dust outflow region. I am not sure if the problem of the satellite retrievals is related to non-spherical shapes of dust: Assuming spherical shapes in retrieval models the higher side-scattering of non-spheres is compensated (then incorrectly) by an extra fine-mode fraction.

17/530 Frequently the AOD over oceans is quite stable ... as long as there are no changes in near surface winds and unless you enter/exit a big plume, as the Saharan dust outflow. Then AOD variations are much stronger than suggested here (possibly these cases were removed in this study a mixed aerosol type to not acceptable aerosol type). In particular for process studies these aerosol gradients (as few as they may be) are of interest.

17/549 the positive bias at low AOD is still there! You cannot use negative biases at higher AOD to suggest a lack of bias.

18/558 The (to MODIS) added use of ATSR data had a negligible impact, since MODIS data were dominating in volume.

18/575 I would not expect significant differences between MODIS and CAMS except that the aerosol typing (e.g. AE) becomes worse in CAMS, as AE in CAMS in model prescribed.

19/610 this is not too convincing as along as a MICROTOPS reference is needed to assure accuracy.

19/620 an important result is that AOD biases change with AOD strength

20/629 have you determined the MODIS Angstrom (if so which wavelengths) or was simply the available product from MODIS used?

20/641 I disagree. There are biases and AOD retrievals make assumptions, which are not validated and often can be poor - affecting in turn the assigned AOD.

21/644 I agree that AERONET sky capabilities and the use of available inversion data would be great. The MFRSR type GUV however is a different approach. AERONET inversions show that sky data offer aerosol size (-distribution) detail, while info on aerosol absorption requires large AOD cases. Anyway, those inversion capabilities with GUV need to be demonstrated first.

22/. . . A nice touch to mention EarthCare . . . although 3MI might be an aerosol satellite dedicated satellite sensor, which likely will launch and provide data sooner.

Figures and tables

table 1 nice

table 2 I would rename the categories: maritime → maritime background, mixed and continental → continental transport, desert dust → mineral dust transport. I would get rid of this biomass (near sources) component since aged transported biomass is more like continental transport. I also wonder why there are so many 'no class' events for GUVis – apparent outliers as they do not appear in COMB. It is disappointing that COM matches are only 5% compared to the MIC matches

table 3 the C-values without a few exceptions are very close to 1.0, so in effect only the lab calibration factors seem to matter (I am not sure how this V to radiance conversion was done, though). And S to correct for the forward scattering into the field of view should be a function of AOD and size (which is also a function of AOD since large dust AOD usually carry larger sizes).

table 4 I would move this into the Appendix.

table 5/6/7 Maybe all the data can be compacted into a single table. I would work with 550 nm statistics only (and use Microtops data to convert SEVIRI to 550nm). I would focus on MIC data and I would separate by 3 regions (dust outflow, continental outflow

[Figure]

and maritime background) and there into lower and higher AOD. Also I would focus on bias and I do not know how to judge correlation. With low AOD cases (and more noise) correlations are meaningless.

figure 1 great overview. But I cannot believe a dust domination in the southern hemisphere (as for the last cruise near 8S) - unless close to the Namib. I also would separate into transported dust (AE <0.5, AOD >0.15), maritime background (AOD <0.15), continental transport (AE >1, AOD >0.15) and mixed (AE 0.5 to 1.0, AOD >0.15) unless close to the Namib. For a clearer presentation I would compare the different reference data in one plot and MIC vs satellite and CAMS in another plot

figure 2 redo with the suggested typing of Figure 1 comments

figure 3 /4 something for the Appendix (why are counts of COMB sometimes higher than for GUVisE in figure 4?)

figure 5 nice detail ... but just because SEVIRI has likely (and missed dust event related) biases at higher AOD, while most cases (at low AOD) are larger than MODIS you cannot conclude that the SEVIRI overestimate (compared to MIC) is lower

figure 6 seeing this plot, who wants to use the Angstrom parameter from satellite remote sensing and even modeling? SEVIRI seems to use a very simple Aerosol model without allowing for large and small Angstrom parameters. Similarly CAMS is constrained by composition, which does not allow for large Angstrom parameters. On the other hand very high Angstrom parameters with MICROTOPS may be associated with low AOD. Thus I recommend to show Figure 6 only for MIC data, when AOD at 870 is at least 0.1.

Actually I call a redo of Figure 6 via a Figure 5-type plot where Angstrom data are compared as a function of AOD bins.

figure 7/8/11 I would redo the aerosol types – see comments to Figure 1 (I assume all component samples fit within circles)

figure 9 interesting aspect

figure 10 nice

resource

I placed relevant monthly 1x1 lat/lon gridded data of the MACv2 aerosol climatology on ftp in ascii and netcdf (for details on file naming look at README)

ftp://ftp-projects.zmaw.de/aerocom/climatology/MACv2_2018/550nm/for_tropos/

there you find expected monthly average for the 450nm/850nm Angstrom parameter and aerosol single scattering properties at 450, 550, 650 and 850nm … maybe this helps as general reference

Review of paper:

**Evaluation of satellite-based aerosol datasets and the CAMS reanalysis
over ocean utilizing shipborne reference observations**
*by J. Witthuhn et al.*

***Positives***
- needed demonstration of (MAN) reference data over ocean
- efforts to evaluate more than just the aerosol column amount (AOD)

***Concerns***
- shadowband (GUV) data seem to lack processing maturity to serve as reference
- comparisons to two satellite data and one assimilation lack interpretation
- aerosol typing via AOD and Angstrom should be redone – more tailored of oceanic reg.

***General comments***

The paper evaluates the aerosol properties over oceans of two satellite retrievals (MODIS, SEVIRI) and one modeling effort (CAMS) that assimilated MODIS data. The evaluation is based on matches (in time and location) to shipborne samples of the direct solar attenuation and of solar scattering during multiple latitudinal Atlantic crossings with the German POLARSTERN research vessel.

The accuracy of (handheld MICROTOPS) direct solar attenuation data is out of question and serves as the main reference in the evaluation. In contrast, the usefulness of the complementary shadowband radiometer (GUV) appears more limited (completely cloud-free situations are needed) and the retrieval algorithm to retrieve the direct solar spectral irradiance component has not reached the needed maturity (still many ad hoc adjustments are needed to match solar attenuations of simultaneous MICROTOPS measurements). Thus recent efforts to improve the shadow-band references (GUV, GUVisE) are more something for the Appendix.

Focusing on the sunphotometer data there are many nice aspects addressed (although other satellite data sets and models could be included – possibly in an additional paper). I am not so happy about the chosen aerosol classification (especially since many source types like biomass are unlikely to be observed over oceans). Based on the sunphotometer data for AOD and Angstrom value (use AE only if AOD550 is larger than 0.15) I would separate into 4 categories (see comments to Figure 1) and work from there. I really like the plots that investigate biases are function of AOD. That said I am really disappointed that this detail is missed in the statistical tables, as common features of (1) AOD overestimate at lower AOD and (2) AOD underestimate at larger AOD lead the authors to claim via a linear fit 'overall good agreement' and 'low biases'.

The paper is great contribution but I think it needs a major revision prior to a publication.

***Detailed comments***

1/17    if AE is overestimated then offer an explanation such as that large mineral dust size-events are missed or that the compositional mix is constrained by model assumptions

1/21+   the radiative effect/forcing introduction (although I am not sure if it is needed) could/should be sharper: In terms of radiative forcing (that is the impact by anthropogenic aerosol at all-sky conditions at TOA) the indirect effect RFaci overall has larger contributions than the RFari. And anthropogenic aerosol impacts are primarily caused by extra smaller aerosol sizes so that (mostly natural components) of seasalt and dust which are mainly observed over oceans are

**Fig. 1.**

---

## Author Comment (AC1) · 7 Feb 2020

Dear Editor and Reviewers,

We thank the editor and the three reviewers for their detailed reviews and thoughtful suggestions. We largely agree with their comments and have tried to address their concerns in the revised paper. In the following text, we give a point-by-point reply to the reviewer's comments. If changes are given in the answers with line, figure or table numbers, those numbers refer to the discussion article. Also the latexdiff file highlights the changes between the discussion article and the revised manuscript.

[Figure]

In order to separate the reviewer's comments and the author's response, we have printed the comments in black, and our response in blue.

We highly appreciate the detailed comments and suggestions, which have helped to improve the manuscript.

Sincerely, on behalf of all authors

Jonas Witthuhn

e-mail: jonas.witthuhn@tropos.de

Overview of changes made to the manuscript:

- Sect. 1:

    – The first three paragraphs have been merged to provide a shorter and more comprehensive introduction about why spectral aerosol observations over ocean are necessary.

- Sect. 2.1:

    – Included short description of MAN Microtops measurement protocol.
    – Included note of calibration procedures for GUVis and Microtops.
    – Included note of post processing in the GUVis introduction and restructured segments.
    – Rephrased the COMB dataset description.

- Sect. 2.2:

    – Added the wavelengths of the MODIS aerosol product.
    – Added note about referring to *MxD04* (MODIS) or *SEV_AER-OC-L2* (SEVIRI) when writing about MODIS or SEVIRI aerosol products.
    – Added a note on the increased side-scatter effect of non-spherical particles.

- Sect. 2.3:

    – Question accuracy of CAMS RA AOD under cloudy sky conditions.

- Sect. 3.1:

    – Change the thresholds of the aerosol classification method as suggested by Stefan Kinne. This effects all figures and tables related to aerosol type by changing number of datapoints but do not change the main conclusions.

[Figure]

- – Emphasize that the presented aerosol classification is an estimate of the dominant aerosol type of the current (mixed) aerosol situation.

- Sect. 4.2:

  - – This section receives a major rework to account for changes in aerosol classification and the presentation of the statistics in Table 5., as well as to avoid several repetitions.
  - – Added a note about the incompleteness of the analysis of the AOD variation between MODIS overpasses, and the additional value of high temporal resolution observations from SEVIRI, since morning and evening hours with potential aerosol growth are omitted.

- Sect. 4.3:

  - – This section receives a rework due to Tables 6 and 7 are omitted or merged to Table 5 in the revised paper.

- Sect. 5:

  - – Added a sentence that it has been shown, that the bias of SEVIRI AOD is dependend on AOD.
  - – Reworded the paragraph about benefits of SEVIRI temporal resolution to emphasize more the targeted applications such as studies about aerosol plumes or frontal zones.

- Table 2 is updated due to the changes made to the aerosol classification.

- Table 4 is updated since outliers are no longer omitted from the calculation of GUVisE.

- Table 5 receives a major update as suggested by Stefan Kinne.

- – Results are presented for the comparison to MIC only.

- – The table focuses on 550 nm only.

- – The table includes the information about aerosol type and in case of CAMS RA additional information with and without AATSR.

- Figure 1 shows all aerosol types.

- Figure 3 shows all aerosol types.

- Figure 4 and similar Figures: added solid lines to connect the median values of each bin for clearer visualisation of the change in bias with increasing wavelength.

- All figures and tables are updated after changing the aerosol classification as suggested by Stefan Kinne.

- Minor changes and corrections to wording, grammar and typos throughout the manuscript as suggested by the reviewers.

Response to RC1 from Referee #4:

*The paper discusses aerosol optical depth observations from ships. The calibration and retrieval techniques for a shadowband radiometer are revised to improve agreement with simultaneous sun-photometer measurements, removing a fractional bias. The new dataset is used to evaluate products from the MODIS and SEVIRI sensors. The former is, unsurprisingly, found to be more precise, but both overestimate the Angstrom exponent, with SEVIRI being out by up to an order of magnitude. The CAMS aerosol reanalysis is similarly evaluated, finding it eliminates much of the bias in the underlying MODIS data.*

*The paper is suitable for publication in this journal. Current satellite retrievals show significant disagreement as to the average AOD over remote ocean and the data provided by this study should be invaluable in resolving that discrepancy. I hope the authors can place their data in a publicy available repository — I am eager to use it in the evaluation of my own satellite products!*

- Thank you for your recognition of the work in this paper. The processed data of all Polarstern cruises with the shadowband radiometer have been published at the PANAGEA data platform, where it can be freely accessed (without the need of a login): https://doi.pangaea.de/10.1594/PANGAEA.910535. We have also updated the link in the 'Data availability' section.

*I have a few minor few comments that warrant the authors' attention:*

- *Though I am fond of your analysis in Fig. 9, I disagree with the scope of your conclusions with respect to the information provided by SEVIRI.*

  – *By using the name of the sensor to refer to a specific dataset, you imply that your conclusions apply to all SEVIRI aerosol products. If one could produce*

*a more accurate aerosol product from SEVIRI, that would provide useful information. You don't present sufficient evidence that all possible SEVIRI products provide minimal additional information.*

- You are completely right. To be more specific, we have added the following text at the beginning of Sect.2.2 after the introduction of used satellite products: *"In the following text unless otherwise stated, the terms "MODIS aerosol products" or "MODIS retrieval" refer to the* MxD04_L2 *and* MxD04_3K *products, and similarly, the term "SEVIRI aerosol product" to the ICARE* SEV_AER-OC-L2 *aerosol product."*

– *Your wording is fairly definitive: 'only offer minor benefits compared to the use of polar-orbiting satellite platforms'. The circumstances where aerosol changes rapidly, such as plumes or the passing of a frontal system, are scientifically very interesting and exactly the sort of circumstances that geostationary imagery are absolutely vital in understanding. Geostationary observations might not add much to our understanding of the climatology of AOD, but this doesn't mean that they only provide minor benefits; they provide targeted benefits.*

- Thanks for pointing this out. We did not mean to disparage the value of geostationary measurements. As you have pointed out, this section was written with climatology studies in mind. To clarify this aspect, we have now reworded the paragraph in the conclusions section as follows: *"Hence, the better time resolution of SEVIRI and other geostationary satellite sensors offers minor benefits for climatological studies compared to the use of polar-orbiting satellite platforms, given its increased uncertainties. The SEVIRI AOD product provide valuable information on the temporal evolution of AOD when the aerosol changes rapidly. Specific cases with high temporal variability are dust storms, plumes of volcanic ash or the passing of frontal systems."*
– *You only evaluate the representivity of observations between the two MODIS overpasses. This omits the periods of boundary layer growth and collapse in the morning and evening, which current polar orbiting satellites do not observe.*

- That's indeed a limitation which deserves to be mentioned. For that reason, we have adopted your wording and added the following text at the end of Sect.4.2: *"We are aware that the analyses presented here do not provide a complete picture of the AOD variability over the full diurnal cycle. It was only possible to analyse the variability between daytime overpasses of MODIS. Continuous evaluation of the daily cycle of AOD are only possible with geostationary satellites such as SEVIRI."*

*There is no need to perform additional analysis, but your conclusions should be reworded to be clearer about their breadth.*

• *I am surprised by the repeated implications that laboratory lamp calibration is inadequate. Calibration in a controlled environment is usually held up as the gold standard of observational atmospheric science. Did the authors mean to imply that such calibrations are insufficient to produce a scientifically valid product(e.g. 'limited accuracy')?*
*I would find that difficult to believe. I suspect what was meant is that there is an intrinsic difference between what a sun-photometer and shadowband radiometer measure. That limits the extent to which they could ever agree without additional correction methods, such as those outlined in this paper.*

- You are right, the repeated use of the term 'lamp-based calibration' suggested that a calibration using a lamp was insufficient. This is indeed not true and not what we wanted to express in the text. Therefore we have removed the 'lamp-based' specification and now simply refer to 'calibration' in section 3.4 and 4.1. In

addition, we now give a short description of the calibration procedures of GU-Vis and Microtops in section 2. Nevertheless, several previous studies state that Langley-calibrations have a lower uncertainty than lamp-based calibartions. Hence, we think that it would be beneficial to also calibrate the GUVis instrument using the Langley-technique since this should lower the calibration uncertainty compared to a laboratory lamp-calibration (**??**).

- *I'm not convinced by the explanation in §4.2 of the narrow, highly biased observations of AOD ≃0.3 in Fig. 4 as I can't see why the choice of aerosol type would only affect one range of AODs. Are there an anomalously small number of collocations in those conditions or are they clustered in a small area? If you loosen your quality control conditions, does the distribution more closely resemble the typical behaviour?*

- Thanks to your comment, we recognize that the wording of these two paragraphs is misleading. We wanted to draw attention to the overestimation of satellite AOD during situations with low AOD values <0.4, where AOD ≃0.3. Panels (a) and (b) of Figure 4 show this overestimation most prominently. We referred to aerosol type with regards to GUVisE and COMB data, since both datasets consist mainly of cases with maritime and desert dust. Therefore the overestimation of AOD on the satellite side is strongly visible in panels (a) and (b) since we attribute it to limitation in the satellite retrivals, which requires parameterization of the surface reflection properties. We replaced these two paragraphs with the following text: "*The SEVIRI retrieval shows an even stronger tendency to overestimate AOD in comparison to the MIC reference dataset. The bias of satellite AOD also shows a dependence on the magnitude of the AOD. A positive bias (overestimation) is mostly found in situations with AOD values below 0.5, and decreases for larger AOD. This behavior is most evident in Fig. 4 panel (a) and panel (b) as the reference datasets are GUVisE and COMB. A similar behavior also appears in the comparison to Microtops (panel (c)), although it is far less pronounced. Since the*

*satellite instruments measure reflected radiance the reflecting properties of the ground used in the retrievals influence the retrieved AOD. Especially for clean atmosphere e.g. low AOD the influence of such parameters (e.g. surface albedo) is strong, since the values measured reflectance at TOA are close to the values of surface reflectance. For larger AOD values the uncertainty of those character­izations shrinks, therefore the overestimation of AOD decreases. Since GUVisE and COMB datasets contain more maritime and desert dust cases than the MIC, this behavior is strongly visible.*"

• *Are the outliers identified at page 11 (line 344) excluded from further analysis? That seems statistically suspect, as we expect large deviations to occur occa­sionally by random chance.*

- We indeed initially excluded outliers to produce the COMB dataset, where GUVis and MIC are combined. We saved the outliers for further analysis, but regarding the low number it does not change the results. You are right, the outliers should not be excluded for the regression to produce GUVisE. We have run the calcula­tions again with included outliers. The results did not change much and do not affect our conclusions. You can check the differences by looking at Table 4 in the latexdiff file.

• *I found it strange that Fig. 1 implies that only maritime and dust aerosols were observed while Fig. 2 shows that mixed and continental were occasionally en­countered as well.*

- In the discussion paper, we had restricted Fig.1 to show only maritime and desert dust, since the plot is already very crowded. Based on your comment, we have now updated Figure 1 to also show mixed and continental aerosol.

• *Fig. 7 is a compelling way to present the limitations in the retrieval of Angstrom exponent. In a future paper, it would be interesting to see a study of the impli-*

=""
</>

*cations of your results on the Aerosol Index, which is widely used as a proxy for cloud condensation nuclei in studies of aerosol-cloud interactions.*

- Thank you for this suggestion, a study considering the uncertainty of the AI would indeed be interesting.

• *At L540, is an increase from 0.90 to 0.92 really evidence of a 'clearly superior' product? That doesn't seem a particularly significant shift*

- We wanted to point out that CAMS RA performance is better than the performance of the SEVIRI product in all statistical measures considered here, even if 'clearly superior' might be misinterpreted to indicate a larger difference. We have now modified this sentence to: "*The CAMS RA outperforms the SERVIRI aerosol dataset in all presented statistical measures at least slightly (e.g., correlation 0.90 versus 0.92 or LOA of 0.13 versus 0.15)*."

• *On page 21, the EarthCARE lidar isn't itself that 'unique'. It's unique that said lidar is being flown collocated with an imager and radar.*

- An additional unique aspect is the high-spectral resolution lidar, which will be able to directly measure extinction, in contrast to CALIOP. We have now reworded the sentence as follows: "*The combination of both instruments on a single satellite and the use of a high-spectral resolution lidar enabling direct observations of the aerosol extinciton at 355nm is a unique feature and will benefit scientific studies targeting aerosols including their radiative effects*."

• *Your discussion about CAMS in §4.3 would be improved if you mention that the inputs to a reanalysis system must be bias corrected before input to ensure as table assimilation of the data. Hence, the reduction in bias is to be expected (but remains evidence of the utility of the CAMS product).*

- Thank you, we have now reworded the paragraph slightly to include this statement:

  " *Further, the dependency of the bias on AOD reduced for the CAMS RA product as it shows low bias values for both low and high AOD values. This is expected since the MODIS AOD bias must be corrected before assimilation into the reanalysis product.*"

- *In Fig. 3, is the sharp transition from maritime to dust aerosol at 0.18 a true feature of your data (which would be concerning) or a feature of plotting the orange points over the blue ones? If the latter, perhaps add some transparency, so the transition is easier to see?*

- The points overlap only slightly, since the AOD on the x-axis is an average value. The sharp transition originates from the aerosol classification method we are using. Based on **?**, and the suggestions of Stefan Kinne (see reviewer comment RC3) we chose the following thresholds to identify aerosol types: maritime background (AOD < 0.15), mineral dust transport (AE < 0.5, AOD > 0.15), continental transport (AE > 1, AOD > 0.15), and mixed (0.5 < AE < 1, AOD > 0.15) type. Therefore, the sharp transition is a result of the method. We are aware that this empirical classification of aerosol type has limitations. While a better aerosol classification based on more complex methods could be developed, we content that for the purpose of our study, this simple empirical approach is sufficient.

- *In point (ii) of the appendix, you change the method for filtering perturbed observations. What motivated this choice? In undergraduate labs, I teach my students to throw out any observation for which the method was suspect as making a correction involves a number of assumptions. Why do you feel the need to keep some corrupted observations here?*

- We found that the former filter criterion was too strict in excluding data from the whole sweep. Since we are only interested in the direct irradiance, it is sufficient

to have good observations of the global irradiance before and after the sweep, and the moment when the shadowband fully shades the diffusor, and right before or after the shadow falls on the diffuser. A full sweep needs 40 seconds and during most of the time, the shadowband shades some part of the sky nowhere close to the sun. In the former (strict) filter method, the whole sweep was skipped if too large variations occured during a sweep. Now the algorithm focuses only on the relevant parts of the sweep (when the shadowband is close to the sun) to assess whether it should be skipped or not. We have found that this new filter works well for identifying perturbed scenarios while providing stable data even during short sunny periods.

*I also include some technical comments and corrections. P1L2 means line 2 of page1.*

- *P1L16 similar performances for both datasets*

- Done.

- *P2L39 e.g. from ships are available*

- Done.

- *P2L57 Does 'earth' need to be capitalized?*

- The AMT guidelines on capitalization leave the decision in particular for 'earth' to the authors, as long as it is consistent. So we leave it as is. Instead we corrected 'Earth' > 'earth' at L21, L31 and L33.

- *P3L79 complex, non-spherical shape*

- Done.

- *P4L109 These findings are understood in the context of the results found for the SEVIRI aerosol product to observe*

- Done.

- *P4L119 Add a space after the comma.*

- Done.

- *P4L124 are publicly available*

- Done.

- *P5L133 reference: the sunphotometer*

- Done.

- *P8L251 of ±30min have been used*

- Done.

- *P9L257 distance angle less than 0.2°*

- Done.

- *P9L272 the analyses are*

- Done.

- *P10L303 Perhaps add 'to ensure the' after 'compensated for'? It means something slightly different but is what I think you meant to say here.*

- Done, thanks for the suggestion.

- *P12L373 Add a space after 'Table'.*

- Done.

• *P16L513 The wrong style of reference is used.*

- Done.

• *P16L518 I don't know what you meant to say by 'follow up'.*

- What is meant hear are overlapping pixels of MODIS images from consecutive Terra & Aqua scans. In the text we replaced 'follow up' with this description: "*To further investigate this point, MODIS collocations with the shipborne datasets are used to serve as random samples to study the AOD variability between successive overpasses. For each pixel of a MODIS image the corresponding SEVIRI AOD for every available SEVIRI image between overlapping MODIS images of consecutive Terra and Aqua overpasses was acquired to calculate the AOD variation.*"
We did this also to other occurrences of 'follow up' in this paragraph.

• *P20L627 has channels only at*

- Done.

• *P20L640 products can provide a*

- Done.

• *P21L672 with the next few years collocated with an imager and radar.*

- Done.

• *Tab.5 collocated data points. Listed*

- Done.

- *Fig.1 The aerosol classification method*

  - Done.

- *Fig.8 requirement for simultaneous . . . figure also shows CAMS RA*

  - Done.

Please also note the supplement to this comment:
https://www.atmos-meas-tech-discuss.net/amt-2019-321/amt-2019-321-AC1-
supplement.pdf

———————————————

---

## Author Comment (AC2) · 7 Feb 2020

Dear Editor and Reviewers,

We thank the editor and the three reviewers for their detailed reviews and thoughtful suggestions. We largely agree with their comments and have tried to address their concerns in the revised paper. In the following text, we give a point-by-point reply to the reviewer's comments. If changes are given in the answers with line, figure or table numbers, those numbers refer to the discussion article. Also the latexdiff file highlights the changes between the discussion article and the revised manuscript.

In order to separate the reviewer's comments and the author's response, we have printed the comments in black, and our response in blue.

We highly appreciate the detailed comments and suggestions, which have helped to improve the manuscript.

Sincerely, on behalf of all authors

Jonas Witthuhn

e-mail: jonas.witthuhn@tropos.de

Overview of changes made to the manuscript:

- Sect. 1:

  - The first three paragraphs have been merged to provide a shorter and more comprehensive introduction about why spectral aerosol observations over ocean are necessary.

- Sect. 2.1:

  - Included short description of MAN Microtops measurement protocol.
  - Included note of calibration procedures for GUVis and Microtops.
  - Included note of post processing in the GUVis introduction and restructured segments.
  - Rephrased the COMB dataset description.

- Sect. 2.2:

  - Added the wavelengths of the MODIS aerosol product.
  - Added note about referring to *MxD04* (MODIS) or *SEV_AER-OC-L2* (SEVIRI) when writing about MODIS or SEVIRI aerosol products.
  - Added a note on the increased side-scatter effect of non-spherical particles.

- Sect. 2.3:

  - Question accuracy of CAMS RA AOD under cloudy sky conditions.

- Sect. 3.1:

  - Change the thresholds of the aerosol classification method as suggested by Stefan Kinne. This effects all figures and tables related to aerosol type by changing number of datapoints but do not change the main conclusions.

  - Emphasize that the presented aerosol classification is an estimate of the dominant aerosol type of the current (mixed) aerosol situation.

- Sect. 4.2:

  - This section receives a major rework to account for changes in aerosol classification and the presentation of the statistics in Table 5., as well as to avoid several repetitions.
  - Added a note about the incompleteness of the analysis of the AOD variation between MODIS overpasses, and the additional value of high temporal resolution observations from SEVIRI, since morning and evening hours with potential aerosol growth are omitted.

- Sect. 4.3:

  - This section receives a rework due to Tables 6 and 7 are omitted or merged to Table 5 in the revised paper.

- Sect. 5:

  - Added a sentence that it has been shown, that the bias of SEVIRI AOD is dependend on AOD.
  - Reworded the paragraph about benefits of SEVIRI temporal resolution to emphasize more the targeted applications such as studies about aerosol plumes or frontal zones.

- Table 2 is updated due to the changes made to the aerosol classification.

- Table 4 is updated since outliers are no longer omitted from the calculation of GUVisE.

- Table 5 receives a major update as suggested by Stefan Kinne.

- – Results are presented for the comparison to MIC only.
- – The table focuses on 550 nm only.
- – The table includes the information about aerosol type and in case of CAMS RA additional information with and without AATSR.

- Figure 1 shows all aerosol types.

- Figure 3 shows all aerosol types.

- Figure 4 and similar Figures: added solid lines to connect the median values of each bin for clearer visualisation of the change in bias with increasing wavelength.

- All figures and tables are updated after changing the aerosol classification as suggested by Stefan Kinne.

- Minor changes and corrections to wording, grammar and typos throughout the manuscript as suggested by the reviewers.

Response to RC2 from Referee #3:

*This paper aims to evaluate the satellite (MODIS, SEVIRI) and reanalysis (CAMS) retrievals of AOD and Angstrom exponent over ocean by comparing them with moving ship-borne observations using Microtops sunphotometers and multi-spectral shadow-band radiometer GUVis-3511 during several cruises in the Atlantic Ocean. The results are re-evaluated for defined aerosol types, mostly maritime and desert dust.*

*Overall, the manuscript is well written and organized, although some improvements may be attained in the discussion of the results.*

*However, the manuscript is rather long enough and some parts may be significantly shortened without any effect in the general discussion and importance of the results, since there are several repetitions throughout the manuscript.*

- *There is a rather long discussion of aerosol direct and indirect effects in the beginning of the Introduction that is beyond the scope of the current research. I understand that authors initially discuss the role of aerosols on global climate and the necessity of accurate measurements of them, in a way to reduce the uncertainty in their climate response, but this part may be shortened in one paragraph (for example the first three paragraphs could be shortened and merged into one).*

- You are right. We have now merged the three paragraphs as suggested. The text is reduced to:
  "*Aerosol particles directly influence the earth's radiation budget through their interaction with solar and terrestrial radiation, and indirectly by modifying the optical properties of clouds (Boucher2013). Studies of aerosol effects on the climate system are based on radiative transfer models. Therefore, knowledge about the spectrally resolved optical properties of different aerosol types is essential. Over ocean, sea spray (Bellouin2005, Loeb2005, Yu2006, Myhre2007) and desert dust*

*(e.g., Tegen2003, Christopher2007, Nabat2015) are the major contributors to the direct radiative effect of aerosol. Observations of aerosol load and optical properties with global coverage are required to improve our understanding of climate-relevant aerosol processes.*"

- *Although a very analytic description is provided for satellite products, GUVis measurements, collocation procedures and so on, there is lack of information about uncertainties in the Microtops-II AOD retrievals, which may be high if the instrument is not exactly oriented to the sun's disk. Usually, 3-5 sets of measurements are taken from Microtops in order to select the best one via techniques described in previous papers (e.g. Sharma2014, Tiwari et al. 2018, Environ. Science Pollution Res.).*

- The Microtops data is processed within the Maritime Aerosol Network (MAN) framework within AERONET. There is a detailed description of the procedure available in (Smirnov2009), indeed the series data consists of the average of >5 consecutive scans. In the paper we simply state the uncertainty estimate of $\pm 0.02$ and cite the general article (Smirnov2009). For clarification, we added a short description in section 2.1.:
  "*The Microtops is a hand-held sunphotometer, which has to be pointed manually at the sun. To minimize uncertainties arise from manual pointing, more than five consecutive scans are averaged to form one measurement (Smirnov2009). The Microtops instrument measures the incident direct normal solar irradiance with a field of view of 2.5° (Porter2001). The MAN Microtops sunphotometers are calibrated against an AERONET master Cimel sunphotometer, which in turn is calibrated using the Langley-technique. [...] The uncertainty of Microtops AOD is estimated to be within $\pm 0.02$ (Smirnov2009).*"

- *In addition, during the W-ICARB cruise campaign over the Bay of Bengal, there was a comparison between Microtops-II and MODIS AODs revealing a very good*

*agreement between them, which may be mentioned in the paper and discussed against the current findings (Kharol2011)*

- Thank you for pointing out this relevant study, we now mention this reference in Sect.4.2:
  "*Nevertheless, the correlations found here agree well with the findings of Levy2013 considering the MODIS C6.1 aerosol product (0.937) and the 550 nm channel. A smaller dataset of Microtops observations was compared to MODIS aerosol products by Kharol2011, where a general overestimation of AOD, and a high correlation was found similar to our results.*"

- *Section 4.2 is composed of numerous relatively short paragraphs, whose meanings are not so distinguishable. This creates some difficulties in reading and understanding exactly the major issue (spirit) of each paragraph. Taking also into account the several repetitions, this becomes more problematic. What I recommend is to merge the paragraphs into longer ones discussing a define issue, for example results of the presented figures and tables and/or discussion on these results.*

- We have rewritten this section and refined the structure to increase the clarity of the text.

- *Special care should be taken throughout the manuscript on avoiding several repetitions. Some of these are emphasized below.*

- As we have rewritten this section we have removed several repetitions.

*Minor comments/corrections*

- *Line 51: Levy et al. (2013) estimated...*

- Done.

- *Line 73: Double use of "system" at the end of this sentence does not make good sense and should be revised.*

- Done.

- *Lines 205-206. I recommend to remove this sentence from this part of the manuscript. In case the reader would expect a better accuracy from MODIS, what's the reason to read the results of this study?*

- Done.

- *Lines 362-364 and lines 380-381. These sentences are just a repetition and one should be removed.*

- We stripped the first sentence to its bare minimum rather than deleting it, since it is used to describe the expectation. The second sentence now provide the explanation for the expectation.

- *Line 447....is presented here.*

- Done.

- Repetitions:

  – *Line 476-480. Since the data...MIC data. Such statements have been repeated several times in the manuscript and may be removed or significantly shortened.*
  – *Line487. This sentence, even rephrased has been stated several times in the manuscript.*
  – *Line493-494. Since the MODIS...accurately. Similarly, this has been stated several times in the manuscript.*

- As we have reworked section 4.2 we have removed several repetitions.

- *Line 541. This emphasizes...*

- Done.

- *Line 562. A slight increase...*

- Done.

- *Line 589. This is a similar statement as in line 571.*

- We deleted the first sentence (L571).

Please also note the supplement to this comment:
https://www.atmos-meas-tech-discuss.net/amt-2019-321/amt-2019-321-AC2-supplement.pdf

**Supplement:**

[revised manuscript text omitted]
 | $C_{PS83}$ | $k_{PS83}$ | $C_{PS95}$ | $k_{PS95}$ | $C_{PS98}$ | $k_{PS98}$ | $C_{PS102}$ | $k_{PS102}$ | $C_{PS113}$ | $k_{PS113}$ | $S$ |
|---|---|---|---|---|---|---|---|---|---|---|---|
| nm | - | $\frac{\mathrm{V\,m^2\,nm}}{\mathrm{W}}$ | - | $\frac{\mathrm{V\,m^2\,nm}}{\mathrm{W}}$ | - | $\frac{\mathrm{V\,m^2\,nm}}{\mathrm{W}}$ | - | $\frac{\mathrm{V\,m^2\,nm}}{\mathrm{W}}$ | - | $\frac{\mathrm{V\,m^2\,nm}}{\mathrm{W}}$ | - |
| 380 | 1.02 | 1.42 | 0.97 | 1.44 | 1.03 | 1.32 | 0.96 | 1.31 | 0.99 | 1.29 | 1.12 |
| 440 | 0.98 | 7.52 | 0.93 | 7.48 | 0.96 | 6.90 | 0.92 | 6.87 | 0.95 | 6.79 | 1.13 |
| 500 | 1.02 | 20.91 | 0.99 | 21.33 | 1.03 | 19.55 | 0.97 | 19.63 | 1.00 | 19.83 | 1.14 |

[revised manuscript text omitted]

---

## Author Comment (AC3) · 7 Feb 2020

Dear Editor and Reviewers,

We thank the editor and the three reviewers for their detailed reviews and thoughtful suggestions. We largely agree with their comments and have tried to address their concerns in the revised paper. In the following text, we give a point-by-point reply to the reviewer's comments. If changes are given in the answers with line, figure or table numbers, those numbers refer to the discussion article. Also the latexdiff file highlights the changes between the discussion article and the revised manuscript.

[Figure]

In order to separate the reviewer's comments and the author's response, we have printed the comments in black, and our response in blue.

We highly appreciate the detailed comments and suggestions, which have helped to improve the manuscript.

Sincerely, on behalf of all authors

Jonas Witthuhn

e-mail: jonas.witthuhn@tropos.de

[Figure]

Overview of changes made to the manuscript:

- Sect. 1:

  – The first three paragraphs have been merged to provide a shorter and more comprehensive introduction about why spectral aerosol observations over ocean are necessary.

- Sect. 2.1:

  – Included short description of MAN Microtops measurement protocol.
  – Included note of calibration procedures for GUVis and Microtops.
  – Included note of post processing in the GUVis introduction and restructured segments.
  – Rephrased the COMB dataset description.

- Sect. 2.2:

  – Added the wavelengths of the MODIS aerosol product.
  – Added note about referring to *MxD04* (MODIS) or *SEV_AER-OC-L2* (SEVIRI) when writing about MODIS or SEVIRI aerosol products.
  – Added a note on the increased side-scatter effect of non-spherical particles.

- Sect. 2.3:

  – Question accuracy of CAMS RA AOD under cloudy sky conditions.

- Sect. 3.1:

  – Change the thresholds of the aerosol classification method as suggested by Stefan Kinne. This effects all figures and tables related to aerosol type by changing number of datapoints but do not change the main conclusions.

– Emphasize that the presented aerosol classification is an estimate of the dominant aerosol type of the current (mixed) aerosol situation.

- Sect. 4.2:

  – This section receives a major rework to account for changes in aerosol classification and the presentation of the statistics in Table 5., as well as to avoid several repetitions.

  – Added a note about the incompleteness of the analysis of the AOD variation between MODIS overpasses, and the additional value of high temporal resolution observations from SEVIRI, since morning and evening hours with potential aerosol growth are omitted.

- Sect. 4.3:

  – This section receives a rework due to Tables 6 and 7 are omitted or merged to Table 5 in the revised paper.

- Sect. 5:

  – Added a sentence that it has been shown, that the bias of SEVIRI AOD is dependend on AOD.

  – Reworded the paragraph about benefits of SEVIRI temporal resolution to emphasize more the targeted applications such as studies about aerosol plumes or frontal zones.

- Table 2 is updated due to the changes made to the aerosol classification.

- Table 4 is updated since outliers are no longer omitted from the calculation of GUVisE.

- Table 5 receives a major update as suggested by Stefan Kinne.

- – Results are presented for the comparison to MIC only.
- – The table focuses on 550 nm only.
- – The table includes the information about aerosol type and in case of CAMS RA additional information with and without AATSR.

- Figure 1 shows all aerosol types.

- Figure 3 shows all aerosol types.

- Figure 4 and similar Figures: added solid lines to connect the median values of each bin for clearer visualisation of the change in bias with increasing wavelength.

- All figures and tables are updated after changing the aerosol classification as suggested by Stefan Kinne.

- Minor changes and corrections to wording, grammar and typos throughout the manuscript as suggested by the reviewers.

Response to RC3 from Stefan Kinne:

*Evaluation of satellite-based aerosol datasets and the CAMS reanalysis over ocean utilizing shipborne reference observations by J. Witthuhn et al.*

*Positives*

- *needed demonstration of (MAN) reference data over ocean*

- *efforts to evaluate more than just the aerosol column amount (AOD)*

*Concerns*

- *shadowband (GUV) data seem to lack processing maturity to serve as reference*

- *comparisons to two satellite data and one assimilation lack interpretation*

- *aerosol typing via AOD and Angstrom should be redone - more tailored of oceanic reg.*

*General comments*

*The paper evaluates the aerosol properties over oceans of two satellite retrievals (MODIS, SEVIRI) and one modeling effort (CAMS) that assimilated MODIS data. The evaluation is based on matches (in time and location) to shipborne samples of the direct solar attenuation and of solar scattering during multiple latitudinal Atlantic crossings with the German POLARSTERN research vessel. The accuracy of (handheld MICROTOPS) direct solar attenuation data is out of question and serves as the main reference in the evaluation. In contrast, the usefulness of the complementary shadowband radiometer (GUV) appears more limited (completely cloud-free situations are*

*needed) and the retrieval algorithm to retrieve the direct solar spectral irradiance component as not reached the needed maturity (still many ad hoc adjustments are needed to match solar attenuations of simultaneous MICROTOPS measurements). Thus recent efforts to improve the shadow-band references (GUV, GUVisE) are more something for the Appendix. Focusing on the sunphotometer data there are many nice aspects addressed (although other satellite data sets and models could be included – possibly in an additional paper). I am not so happy about the chosen aerosol classification (especially since many source types like biomass are unlikely to be observed over oceans). Based on the sunphotometer data for AOD and Angstrom value (use AE only if AOD550 is larger than 0.15) I would separate into 4 categories (see comments to Figure 1) and work from there. I really like the plots that investigate biases are function of AOD. That said I am really disappointed that this detail is missed in the statistical tables, as common features of (1) AOD overestimate at lower AOD and (2) AOD under-estimate at larger AOD lead the authors to claim via a linear fit 'overall good agreement' and 'low biases'. The paper is great contribution but I think it needs a major revision prior to a publication.*

Thank you for your valuable and detailed review of our manuscript. We largely agree with your suggestions, which motivated the following changes we made to the manuscript:

- Changes of the thresholds used for the aerosol classification.

- Table 5 has been changed, now covering the bias at different magnitudes of the AOD and for different aerosol classifications. Therefore, Tables 6,7 and Tables in the Appendix have been omitted.

- Section 4.2 has been modified according to the changes of the aerosol classification to present the statistics given in Table 5.

*Detailed comments*

- *1/17 if AE is overestimated then offer an explanation such as that large mineral dust size-events are missed or that the compositional mix is constrained by model assumptions*

- Yes, we reworded the sentence to:
  "*When considering aerosol conditions, an overestimation of AE is found for scenes dominated by desert dust for MODIS and SEVIRI products versus the shipborne reference dataset. As the composition of the mixture of aerosol in satellite products is constrained by model assumptions, this highlights the importance of considering the aerosol type in evaluation studies for identifying problematic aspects.*"

- *1/21+ the radiative effect/forcing introduction (although I am not sure if it is needed) could/should be sharper: In terms of radiative forcing (that is the impact by anthropogenic aerosol at all-sky conditions at TOA) the indirect effect RFaci overall has larger contributions than the RFari. And anthropogenic aerosol impacts are primarily caused by extra smaller aerosol sizes so that (mostly natural components) of seasalt and dust which are mainly observed over oceans are almost irrelevant even for RFari. The impact of these two components on Raci (seasalt to increase CCN, dust to increase IN) has been demonstrated but the overall impact strength remains unclear (for more on radiative effects and forcing have a look at the MACv2 paper)*

- You are right. Since RFari and RFaci are not further considered in the manuscript, we have decided to shorten and merge the three introduction paragraphs into one, also based on the commments RC2 by reviewer #3. The new paragraph only gives a general introduction of aerosol direct and indirect radiative effects and emphasizes the need of spectral aerosol observation over ocean for climatology studies. The text now reads as follows:

"*Aerosol particles directly influence the earth's radiation budget through their interaction with solar and terrestrial radiation, and indirectly by modifying the optical properties of clouds (Boucher2013). Studies of aerosol effects on the climate system are based on radiative transfer models. Therefore, knowledge about the spectrally resolved optical properties of different aerosol types is essential. Over ocean, sea spray (Bellouin2005, Loeb2005, Yu2006, Myhre2007) and desert dust (e.g., Tegen2003, Christopher2007, Nabat2015) are the major contributors to the direct radiative effect of aerosol. Observations of aerosol load and optical properties with global coverage are required to improve our understanding of climate-relevant aerosol processes.*"

- *2/37+ if you mean CERES (broadband radiation) data then say so, even though then the link of aerosol retrievals (they make a compositional and size assumptions to yield AOD estimates) to broadband fluxes is not straight forward and need further rad.transfer modeling.*

- No, this statement does not refer to CERES data specifically. This paragraph is just a statement about the capability of satellite observations to provide continuous and global coverage for climatological studies. As well as the need to validate such satellite retrievals.

- *2/39 Exactly, this is why you want the ship data: to evaluate and to test (retrieval) model assumptions.*

- Yes.

- *2/55 you may also consider to evaluate to (the newly re-processed) MISR data, to VIIRS data (designated as MODIS follow on), to SLSTR (the ATSR follow-on) data and to GRASP-type (e.g. MERIS) data.*

[Figure]

- While we in agree in principle that it would be very interesting to study a wider range of aerosol products from different satellite instruments, we chose here to evaluate "only" two satellite products, as representative approaches of a simple (two spectral channels) and a more advanced (seven spectral channels) retrieval of a geostationary and polar orbiting satellites respectively.

• *2/76 A separate evaluation for fine (and mostly anthropogenic) and coarse-mode AOD (as offered for the MICROTOPS data via the SDA approach) - in place of an Angstromparameter evaluation - would elevate a marine aerosol retrieval or modeling evaluation: Most global models and satellite retrieval models use bi-modal (by size) scheme, while Angstrom parameters depend spectral choice and becomes extremely noisy and probably meaningless at low AOD values in already one of the two spectral bands needed.*

- We acknowledge your point here, and yes, the AE evaluation for maritime aerosol is extremely noisy. We have therefore modified the AE comparison figures, as you suggest later, to filter out situations with low AOD. We have decided to keep the AE evaluation, as for other types than maritime aerosol, it is still a widely used method to extrapolate AOD to other wavelengths.

• *5/131 I would have used a different description of the Angstrom parameter : ANG = -ln(AOD1/AOD2) / ln(wavel1/wavel2)...as the negative spectral slope in ln/ln-space*

- As the Ångström relation is commonly known, it is a matter of taste how to present equation (1). We prefer the presentation we have chosen, as it clearly shows the relation of AOD and wavelength.

• *5/138 The AOD shadow-band radiometer requires more actions (leveling corrections) and is more restrictive (completely cloud-free over the time of a scan).*

- We have added one sentence describing the shadowband radiometer post processing (e.g., leveling correction):
"*The shadowband radiometer GUVis utilizes an entrance optic with a global field of view combined with a shadowband that performs a 180° sweep, while the global irradiance is measured at a high temporal frequency of 15 Hz. Several corrections are applied as post-processing to correct the influence of the ship motion, and to retrieve the direct spectral irradiance for later AOD calculation, as is described later. The measurement principle of the shadowband radiometer can be described as follows: While the global irradiance is observed with the shadowband in its lowest position between sweeps, the shadowband blocks a fraction of the incoming diffuse irradiance during its rotation, and will occlude the direct irradiance at a specific angle determined by instrument orientation and sun position. From the irradiance time series measured during the sweep, the global, diffuse and direct irradiance components can be inferred (Witthuhn2017).*"
It is however not entirely true that a completely cloud-free situation is needed for a successful retrieval. The direct irradiance can be inferred from the data even if clouds are present, unless they cover or are close to the sun. So I would prefer "sun-free" instead of "cloud-free", which is a similar requirement for the Microtops measurement protocol.

- *6/180 The empirical correction is covered in detail but only applied to the shadowband(GUV) instrument (and not to the sunphotometer). Considering that the GUV match statistics is so much sparser then for the sunphotometer (with useful data evaluation data at this stage only through sunphotometer matches) so that in the end only sun-photometer data are used in the evaluation, I wonder if all that GUV detail (table and figures) is not better added to the Appendix. In terms of the forward scattering correction I am worried about a linear approach - which ignores a dependence on aerosol size (over oceans certainly at larger AOD) size.*

- We partly agree with this point. Since we have adapted the GUVis retrieval within

the scope of this study, we have to provide the details of the processing and correction. This is not necessary for Microtops, since it is already well documented. In an internal draft version we had moved all the GUVis detail to a section in the Appendix. But since one key point of the results section is the inter-comparison of both shipborne instruments, we have decided to provide the details of the correction and the cross-calibration in the main section, and only describe the processing-update in the appendix.

- *6/179 COMB is a subset of GUV, where GUV outliers (inconsistent to MIC) removed -correct?*

- Not entirely. COMB consists of the mean of the collocated GUVisE and MIC products. Outliers were removed previously, but are now included in the analysis given in the revised manuscript. To clarify this aspect, we have rephrased this paragraph to:
"*The enhanced GUVis dataset (GUVisE) has been combined with the Microtops dataset to obtain a merged surface product, to test whether the combination can lead to further improvements in accuracy. This combined surface dataset (COMB) serves as third reference dataset for the evaluation of the satellite products. COMB consists of the mean of the collocated GUVisE and MIC AOD for this purpose. As shown in Table 2, the total amount of data points decreases to 1033 due to the collocation requirement.*"

- *7/194 mention also the AOD (I assume 550) wavelength of MODIS*

- We used the full dataset of *Effective_Optical_Depth_Average_Ocean*, so we added all seven channel wavelengths in the text:
"*Satellite based aerosol datasets over ocean considered here are obtained from both the MODIS and SEVIRI satellite instruments. The MODIS Collection 6.1 (C6.1) level-2 aerosol products MxD04_L2 (Levy2015) and MxD04_3K (Re-*

*mer2013) are used from both the Terra and Aqua satellites. This dataset includes the AOD at 470, 550, 660, 860, 1240, 1630 and 2130 nm.* "

- *7/208 there are three major solar retrieval problems for dust outflow:*
  *(1) dust is solar spectrally flat (like clouds) and large AOD events may be removed as clouds -> retrieved dust AOD too low*
  *(2) dust size is underestimated and with it the size related 'dust absorption' (with low SSA values) -> retrieval AOD will be too high*
  *(3) non-sphericity has increased side-scatter so that a retrieval model with coarse-mode and fine-mode spheres will interpret the extra side-scatter artificially with extra fine-mode spheres→wrong composition and likely size underestimate (see under 2).*

- Thank you for this insightful comment. As we don't evaluate the cloud masking of the satellites, the key property of desert dust influencing a valid AOD measurement is its non-sphericity, as it is stated in the text. We rephrased this sentence to mention the increased side-scatter as the result of the non-sphericity:
  " *A degraded accuracy for aerosol properties in the presence of desert dust in both satellite products is expected, since dust particles are non-spherical. This leads to an increased side-scatter effect compared to spherical particles which are assumed in both retrievals.*"

- *7/219...but cloud coverage may be wrong in global modeling...aside that the AOD might be different in clear-sky and cloud-sky regions of modeling*

- Yes, therefore we added a sentence to this paragraph:
  "*The advantage of utilizing CAMS RA over satellite observations is the availability of aerosol properties independent of factors such as cloud coverage or satellite orbit. Albeit the accuracy of AOD under cloudy sky conditions in the model might be questionable.*"

- *7/226 in CAMS only AOD is assimilated and in its bias correction the compositional mixture of the forecast model is maintained...so the composition may get worse (in comparison to a forecast without assimilations). By the way, the use of the (relatively sparse) ATSR data added little in a combined (ATSR & MODIS) assimilation - as the volume of MODIS data dominated*

- This is true, we found only slight differences in the comparison MIC & CAMS with and without AATSR (see Sect.4.3). We slightly rephrased the sentence regarding assimilation of MODIS products to emphasize, that only AOD is assimilated:
  "*It relies on the assimilation of global observational datasets into the Integrated Forecast System (IFS) from various satellites to provide a global picture. In terms of aerosol properties, the AOD from the products of the MODIS C6 from both Terra and Aqua are assimilated, while the composition mixture is maintained as given from the IFS.*"

- Comments on the aerosol classification

  – *8/228 the offered aerosol classification (based on AOD and ANG) is rather general and prone to failure if one of the two needed AOD values becomes low. For those events any ANG value becomes meaningless. And with respect to aerosol typing aerosol is always a mixture of many components. And unless surface winds are very calm, even marine aerosol has Angstrom parameters between 0.7 and 1.0. An ideal thing would be if simultaneous info on aerosol absorption would be available (as demonstrated in MACv2). To extract info an components, I would start with an AOD separation into fine-mode (r<0.5um) and coarse-mode (r>0.5um) contributions via the fine-mode AOD fraction (easily derived from the AOD spectral dependence) to separate seasalt/dust from pollution/wildfire aerosol (in a quantitative way!). The MAN data-base (where MICROTOPS data are stored) offers such derived (fine-mode-fraction) estimates!*

- – *8/240 aerosol is always a mixture of many components and what the simple 'Toledano' scheme does is simply identifying a likely dominant component in a qualitative way. I disagree though that marine aerosol has Angstrom parameters larger than 1.2 unless there are contributions from pollution and or biomass burning. AE values larger than 1.2 at should be linked fine-mode aerosol with contribution from sulfate / nitrate (less absorption), pollution and aged biomass burning (medium absorption) and fresh biomass burning (highly absorption and AE >1.5).*

- – *8/245 all classes are 'mixed'...but in terms of the mixed category here, how useful is a 'mixed' category if it could be a 10/90% or a 90/10% mixture?*

- You are right, the used aerosol classification is very simple and serves only as a rough estimate of the dominant type of aerosol. Of course all classifications are mixed to some extent, which implies that the mixed category has no obvious dominant aerosol type and could therefore also be named "unclassified". We rephrased and added a sentence to this paragraph:
"*The pair of AOD and AE values is checked against empirical thresholds to identify the dominant aerosol type of the current situation as being one of maritime background (AOD < 0.15), mineral dust transport (AE < 0.5, AOD > 0.15), continental transport (AE >1, AOD > 0.15), or mixed (0.5 < AE <1, AOD > 0.15) type. It should be noted, that all categories are expected to cover mixed aerosol types to some extent. Therefore, the mixed category consists of a mixture of aerosol without a dominant type.*"
For the purpose of this study, the presented aerosol classification enables the investigation of the performance of the satellite and CAMS RA products with regard to the predominant aerosol situation such as maritime or desert dust. As a future improvement, benefits of using a more complex aerosol classification including absorption, fine- and coarse mode aerosol should be evaluated. As you have suggested also in later comments, we have restricted the AE validation

to AOD(870nm)>0.05, and changed the thresholds of the aerosol classification based on your recommendation.

- *8/230 the AE overestimate in CAMS can be just related to the improper component mixture (too much fine-mode contributions) in the forecast model, which an (total) AOD assimilation cannot fix .. and in case of AOD adjustment during the assimilation process will pull away from likely expectations (such that mix of assimilated products is often worse than that of the forecast model)*

- We have rewritten this paragraph taking into account your comment:
  "*A first validation presented within (Inness2019) emphasizes the high quality of AOD in the CAMS RA system, as judged by a comparison to AERONET stations around the world. However, an overestimate of AE was shown during desert dust events, and was attributed to the fixed component mixture (e.g., less dust in CAMS RA) in the forecast model. Further evaluation with a focus on individual aerosol components as well as aerosol properties over ocean has been recommended (Inness2019).*"

- *9/262 why only so few matches between MIC CAMS-RA (as modeling has AOD data at any location and time)?*

- CAMS RA data has a temporal resolution of 3 h, which limits the available data-points from one CAMS RA grid cell during daylight to about 4 or 5. If the ship is able to move to another grid cell (one cell is 80x80km) within ±30min of a CAMS timestep, the number of possible collocation could increase. But since the top speed of RV Polarstern is about 30 km/h its unlikely that there are more than two collocation per CAMS timestep. If we assume very clearsky conditions and a fast moving ship, the number of collocation samples per day might reach up to 8. From all cruises ever done within MAN until 12.2016 (CAMS data was only available until this date), there are only 1590 days consisting of at least one valid AOD measurement (valid implies here that none of the AOD values of the channels

380,440,500,675 and 870nm are NaN). Therefore a total number of 2474 valid collocated datapoints seem reasonable.

- *10/298 Well it is sad that GUV retrieved AOD are often inconsistent to (trusted) MICROTOPS data – as apparent more algorithm work is needed.*

- Sure, since the methods of measurement are considerably different, it is hard to compare the direct sun products without extensive radiative transfer modeling. On the presented database, we tried to produce a fair comparison, trusting the Microtops data. As also emphasized in the manuscript, the deviation of both instruments on a majority of cases is within the uncertainty limits of both instruments. We are aware that the linear dependence found in the bias is most likely not applicable to all situations with varying aerosol type. The correction with the linear factor is only feasible to provide a fair comparison, closing the gap of different methods/field-of-view influences within the presented study. Longer term, we would like to come up with an improved correction for the GUVis data, which might be situation-dependent.

- *10/306 To achieve GNU vs MICROTOPS consistency a rather complex procedure is applied to GNU data based on comparisons to almost simultaneous MICROTOPS samples. Hereby the described method seems a bit arbitrary to me. To account for the forward scattering into the GNU 15deg field of view (isn't 15 deg just the width of the band so in effect the missed diffuse radiation is even larger than 15 deg?) a constant Cis defined...while in reality this forward scattering contribution depends (at completely cloud-free conditions) on aerosol size (!) and multiple scattering potential (AOD, air-mass)...so a constant seems a poor choice. And then there is even a second scaling factor for a temporal (do you mean sun-elevation?) correction!*

- Actually, the constant $C_i$ is applied for a calibration correction to compensate for differences in Microtops and GUVis calibration for the sake of comparability. This

is done separately for each cruise, since the instruments are calibrated between the cruises and instrumental response might drift over time. We have implemented here the method used for MFRSR shadowband radiometers introduced by (Alexandrov2002). This corrections is done on the irradiance measurements (Eq.5) and therefore appears as the term $\mu_0\,c_i$ in Eq. 10. In contrast, $S_i$ is introduced as a constant linear scale factor of the GUVis AOD in order to correct for the discrepancy attributed to forward scattering. (Sarra2015) introduced this method for shadow band radiometers, also suggesting a quadratic fit. Based on our results, a quadratic term does not lead to significant improvement in our case, so we chose to utilize the linear model to reduce the complexity (as stated in L338). As such, we disagree that the corrections are arbitrary.

- *12/376 a fixed bias of -0.02 (with the new processing) seems strange as I would expect a bias related to missed forward scattering to be function of AOD and size..? Actually Figure 3 shows such an AOD dependence of the bias. Generally: the entire correction explanations of 3.4 and of 4.1 (till line 400) including the comparisons of Table 4 are technical details which seem better suited for the Appendix as the focus should be on the comparison of satellite (and CAMS) data to trusted references.*

- Yes, you are right, the bias is not fixed, and is proportional to AOD in our case. This paragraph refers to the statistics calculated in Table 4, where we calculate the mean bias (which in this case is -0.02) and its dependency on AOD as the correlation factor R(D) (in this case about -0.7, which emphasize the linear dependence also seen in Figure 3). As you rightly explain, we have attributed this to forward scattering, knowing the difference in "field-of-view" of both instruments. Therefore we have chosen to correct this linear dependent bias to produce the GUVisE dataset.
As the comparison of MIC and GUV is part of the discussion, and we think that the automatic nature of GUV measurements offers advantages over the manual

MIC measurements, we have decided to not move this sections to the appendix.

- *12/383...the forward scattering missed radiation problem is more an issue for larger size and AOD (e.g. especially relevant for larger mineral dust sizes). You actually show in Figure 3a that this bias is near zero at low AOD and stronger at larger (likely dust) AOD...in that sense it is misleading to talk about a -0.02 bias (but I said this before).*

- This comment is covered by the answer of the previous comment.

- *13/412 how was the SEVIRI AOD at 550nm derived (was the MICROTOPS Angstromparameter used?)*

- The ICARE SEVIRI product (*SEV_AER-OC-L2*) is based on the wavelengths 630 and 810nm. The dataset is delivered with AOD at 500nm and AE. We used the AE reported for the SEVIRI product for the calculation of AOD at 550nm. We added "*Since the SEVIRI dataset does not provide AOD at 550 nm, it was calculated with Eq. (1) using the AOD of 630 nm and the AE from the SEVIRI dataset.*" to this paragraph.

- *13/417 The larges MODIS AOD outliers in the 0.3 bin vs GUVisE may be less meaningful since the sample number is fairly low. This raises the question, why the GUVisE dataset is so small and why this data-set only addresses selective dust and marine cases. Does it mean that the retrieval and correction method very frequently fails?*

- No, until now we only have data of five cruises with RV Polarstern, which are about 4-5 weeks of consecutive measurements, so about 170 days of measurements on the ocean. This means there have been about 340 overpasses (Terra & Aqua) during daytime to collocate the data to. Subtracting days in port or days with some clouds, a number of about 200 collocated datapoints does sound reasonable. I compared the number of collocated datapoints of GUVis and MIC from

[Figure]

the five cruises which operated the the GUVis (see table 1). The table shows that more collocations are achieved with GUVis. This is expected, since GUVis measures automatically and continuous, while Microtops is operated manually with individual breaks.

**Table 1.** The table presents the total amount of datapoints collocated from satellite to the ship-borne datasets of GUVis, Microtops, within a 30 and 5 min timeframe. The table compares data collected during certain cruises with *RV* Polarstern (PS83, PS95, PS98, PS102 and PS113).

| Reference | GUVis | MIC |
|---|---|---|
| total | 10412 | 1572 |
| Collocation [±min] | 30 / 5 | 30 / 5 |
| CAMS | 141 / 90 | 157 / 67 |
| SEVIRI | 1126 / 996 | 697 / 467 |
| *MxD04_L2* | 147 / 80 | 115 / 34 |
| *MxD04_3K* | 210 / 82 | 141 / 29 |

- *14/424 I do not see major differences between 4a and 4b unless low (and less meaningful) statistics is considered. the low GUV sample number remains a mystery to me (a few 'marine' and 'dust' cases).*

- Yes, there is no more a clear improvement from GUVisE to COMB. The paragraph was written with a comparison to GUVis (no forward scattering / calibration correction) as panel a) and we overlooked this detail in the internal review process. This paragraph has been omitted/merged in the new version of the manuscript with the leading and following paragraph addressing Figure 4.

- *14/435 you could compare if the SEVIRI AOD550 data would be different if the MICROTOPS 440/870 Angstrom had been used instead (also the SEVIRI band are spectrally much broader with some water vapor contamination especially in 810nm band).*
- Yes, as expected, the SEVIRI performance increases with the usage of AE from an external source like Microtops. Below in Table 2 we show the results as in Table 5 in the manuscript, but SEVIRI AOD at 550nm is calculated once with its original AE and once with MIC AE. With the usage of MIC AE the performance of SEVIR at 550nm is similar to the comparison at 630nm (lower bias, less proportionality of bias on AOD). Nevertheless, as it is the goal of the study to show certain shortcomings of the satellite products, we have calculated all statistics using only data of these datasets itself. High uncertainty of SEVIRI AE is expected can be shown in this study.

  Nevertheless, we have reworked Table 5 also to your recommendations later, including the usage of MIC AE for SEVIRI AOD at 550nm (see Sect.4.2 and Table5 in the manuscript).

**Table 2.** Statistics as in Table 5 SEVIRI versus MIC, but SEVIRI AOD 550 is calculated with AE from SEVIRI product (original) and with MIC AE for comparison.

| Instrument | N | channel | R | linear regression | R(D) | bias ± LOA | G1 | G2 |
|------------|-----|---------|------|-------------------|------|------------|-----|-----|
| - | - | nm | - | - | - | - | % | % |
| SEVIRI | 2*10060 | 550 | 0.90 | Y = 0.81 X-0.00 | 0.21 | 0.03±0.15 | 56 | 68 |
| original | | 630 | 0.89 | Y = 0.87 X-0.00 | 0.06 | 0.02±0.14 | 63 | 73 |
| SEVIRI | 2*10055 | 550 | 0.88 | Y = 0.85 X+0.00 | 0.08 | 0.02±0.15 | 60 | 71 |
| MIC AE | | 630 | 0.89 | Y = 0.87 X-0.00 | 0.06 | 0.02±0.14 | 63 | 73 |

- *14/439 I would say Figure 5 shows (opposite to your assessment) that SEVIRI AOD is usually higher than MODIS (at low to median AOD) and that at high AOD SEVIRI likely has a low bias, as (large AOD dust events with no solar spectral dependence)are probably removed during cloud screening.*

- We did not draw this conclusion from figure 5 but from figure A3, where SEVIRI AOD (630nm) is indeed slightly lower in all cases. But as we reworded this section this statement has become obsolete. Figure5 was updated to show the comparison of 550 and 630nm. As you have mentioned, SEVIRI overestimation is larger than MODIS for lower AOD values and turns into an underestimation at higher AOD bins.

- *14/445 Can you say this when samples of the two different MODIS products are different?*

- Our collocation method ensures that all pixels within a 0.2° radius are considered. The MODIS products may have different spatial resolutions, but are both built on the same radiance observations using the same algorithm and lookup tables. Because of their resolution both products differ only due to their method of combining pixels of the radiance measurement. Over ocean this means the number of good pixels for a valid retrieval is reduced for the 3km product (5 instead of 10 for 10 km product) and obviously the size of the pixel array of the retrieval box (6x6 instead of 20x20 for the 10 km product). Nevertheless, as the difference is negligible, this sentence is omitted in the new version of this manuscript.

- *15/471 Why did you apply SEVIRI Angstrom and not MICROTOPS Angstrom data to interpolate from 630 to 550nm?*

- As we want to evaluate the satellite products, we compared AOD and AE from the satellite to the reference data. As expected this leads to uncertainties for AE of the SEVIRI product, which is calculated based on the measurements of only two spectral channels. With the revision of this section, we have calculated AOD 550 for SEVIRI both ways (SEVIRI AE and MIC AE) to show the limitations of the SEVIRI AE.

- *15/474 Why would I want to use COMB if I have MIC (and much better statistics)*

- COMB ensures the comparability of both GUVis and MIC products, since both instruments utilize different methods and agree within their uncertainty limits. As

the true values of AOD are unknown, COMB can serve as a best estimate of the surface measurements. Unfortunately, the statistics are restricted to situations when both instruments are operated together, which means data are available only for four cruises and with a time resolution determined by MIC. As MIC was operated since 2004 in numerous cruises, statistical significane is obviously better. Extending this study in the future will lead to better statistics, and will also offer additional insights on the accuracy of both shipborne instruments.

- *15/478 the MICROTOPS 440/870 Angstrom is more reliable than the 670/870 Angstrom. (The label in Figure 7 says 440/870...but the captions say differently?). Anyway, you do make sure not to include AE data if AOD at 870nm is below 0.05 ... otherwise derived Angstrom parameters are likely close to 'garbage'.*

- Thank you. We corrected the label of Figure 7. AE is calculated for 440/870nm. Also we have redone the figure 7 and similar figures and restrict AOD870>0.05.

- *16/489 I agree that at low AOD satellite tend to overestimate AOD due to a more relaxed or insufficient cloud-screening (sometime there are many low clouds which are hardly visible)*

- Thanks.

- *16/495 the CAMS forecast model (and thus also MODIS AOD data assimilations) have a too high fine-mode fraction for the dust outflow region. I am not sure if the problem of the satellite retrievals is related to non-spherical shapes of dust: Assuming spherical shapes in retrieval models the higher side-scattering of non-spheres is compensated (then incorrectly) by an extra fine-mode fraction.*

- While this is an interesting interpretation, we decided to not include this in the manuscript, as we do not investigate fine and coarse mode separately.

- *17/530 Frequently the AOD over oceans is quite stable...as long as there are no changes in near surface winds and unless you enter/exit a big plume, as the Saharan dust outflow. Then AOD variations are much stronger than suggested here (possibly these cases were removed in this study a mixed aerosol type to not acceptable aerosol type). In particular for process studies these aerosol gradients (as few as they may be)are of interest.*

- Yes, we reworded the paragraph slightly to emphasise such process studies as an application for high temporal resolution products. In Sect.4.2:
  " *Continuous evaluation of the daily cycle of AOD are only possible with geostationary satellites such as SEVIRI. Nevertheless, with the high temporal resolution, the SEVIRI product may be needed for many applications, such as case studies of dust or smoke plume development, where high variability of AOD is expected.*"
  In Sect.5:
  "*Nevertheless, the SEVIRI AOD product provide valuable information on the temporal evolution of AOD fields where aerosol changes rapidly. Specific cases with high temporal variability are dust storms, plumes of volcanic ash or the passing of frontal systems.*"

- *17/549 the positive bias at low AOD is still there! You cannot use negative biases at higher AOD to suggest a lack of bias.*

- To cover this we reworded Sect.4.2 and recalculated Table 5 as you have recommended.

- *18/558 The (to MODIS) added use of ATSR data had a negligible impact, since MODIS data were dominating in volume.*

- Yes. Still, we see a slight difference favouring data with ATSR. But as the number of datapoints is low, this small differences are not that meaningful. As emphasized in L563.

- *18/575 I would not expect significant differences between MODIS and CAMS except that the aerosol typing (e.g. AE) becomes worse in CAMS, as AE in CAMS in model prescribed.*

- Yes, we found no significant differences. But we cannot verify if AE is worse in CAMS, it just looks more restricted.

- *19/610 this is not too convincing as along as a MICROTOPS reference is needed to assure accuracy.*

- In order to assure the comparability we have to correct the influence of forward scattering on the shadowband radiometer. Still, the shadowband provides good performance for irradiance measuruments. Therefore, we think that the GUVis works fine as an standalone instrument, but one has to keep the limitations for aerosol retrieval in mind, such as larger influence of forward scattering compared to a narrow field of view sunphotometer.

- *19/620 an important result is that AOD biases change with AOD strength*

- Thank you for pointing this fact out. We added this sentence to the paragraph: "*Also it was shown that the bias of SEVIRI AOD is dependent on the AOD, as the bias for AOD < 0.4 is larger than for the MODIS product, and the bias for AOD ≥ 0.4 turns negative (underestimation).*"

- *20/629 have you determined the MODIS Angstrom (if so which wavelengths) or was simply the available product from MODIS used?*

- The MODIS AE is determined from the MODIS AOD. For the AE comparison to MIC we have used the 470 and 860nm channels.

- *20/641 I disagree. There are biases and AOD retrievals make assumptions, which are not validated and often can be poor - affecting in turn the assigned AOD.*

- While we acknowledge and generally agree with the concerns raised in this comment on satellite retrievals, our results do confirm that the satellite products agree with the shipborne reference observations within the expected error limits. Hence, we believe our statement "our results confirm that satellite products can provide a global view of AOD" is well-founded, at least as long as the error limits are properly taken into account. To address the concern, we have modified the text as follows:
  "*Our results confirm that satellite products can provide a global view of the spatiotemporal aerosol distribution e.g for climate studies or model assimilation, as long as their error limits are properly taken into account, and spectral extrapolation of products is avoided.*"

- *21/644 I agree that AERONET sky capabilities and the use of available inversion data would be great. The MFRSR type GUV however is a different approach. AERONET inversions show that sky data offer aerosol size (-distribution) detail, while info on aerosol absorption requires large AOD cases. Anyway, those inversion capabilities with GUV need to be demonstrated first.*

- Yes, we plan to continue working on the GUVis processing and want to investigate further inversion methods in comparison to AERONET / MAN.

- *22/...A nice touch to mention EarthCare...although 3MI might be an aerosol satellite dedicated satellite sensor, which likely will launch and provide data sooner.*

- Yes, that might indeed be true. We mentioned MTG, EarthCARE and ESP-SG in no particular order.

*Figures and tables*

- *table 1 nice*

- Thanks.

• *table 2 I would rename the categories: maritime -> maritime background, mixed and continental -> continental transport, desert dust -> mineral dust transport. I would get rid of this biomass (near sources) component since aged transported biomass is more like continental transport. I also wonder why there are so many 'no class' events for GUVis – apparent outliers as they do not appear in COMB. It is disappointing that COMB matches are only 5% compared to the MIC matches*

- We now applied the aerosol classification as you suggested in the manuscript. Yes, the low number on datapoints do not lead to meaningful statistics, but this is expected, as we only have five cruises on the Atlantic ocean where we can collocate MIC and GUVis. As shown in the answer to your earlier comment to 13/417, the total amount of MIC datapoints during this five cruises is 1572.

• *table 3 the C-values without a few exceptions are very close to 1.0, so in effect only the lab calibration factors seem to matter (I am not sure how this V to radiance conversion was done, though). And S to correct for the forward scattering into the field of view should be a function of AOD and size (which is also a function of AOD since large dust AOD usually carry larger sizes).*

- This table presents the lab calibration factors k for the GUVis instrument along with the C-values to show the difference of calibration of GUVis and Microtops. All C-values are close to one, so both instruments are well calibrated (and the bias is therefore not caused by different calibrations). S is basically the slope for the linear correction of the bias caused by the forward scattering (see Eq.10). Therefore the correction is dependent on AOD. On our case study we found a simple linear correction model feasible, but we are aware, that this might not be true for cases with different aerosol type or mixtures.

• *table 4 I would move this into the Appendix.*

- As the comparison of MIC and GUV is part of the discussion, we have decided to not move this table to the appendix.

• *table 5/6/7 Maybe all the data can be compacted into a single table. I would work with 550nm statistics only (and use Microtops data to convert SEVIRI to 550nm). I would focus on MIC data and I would separate by 3 regions (dust outflow, continental outflow and maritime background) and there into lower and higher AOD. Also I would focus on bias and I do not know how to judge correlation. With low AOD cases (and more noise)correlations are meaningless.*

- We see the benefits of this condensed table you suggest. Therefore we have updated Table5 as you recommend and drop table 6/7 and the appendix tables. Nevertheless, we kept the correlation in, as it is widely used in validation studies and can therefore be compared to them. We are aware, that the correlation is meaningless for certain selections, such as maritime aerosol.

• *figure 1 great overview. But I cannot believe a dust domination in the southern hemisphere (as for the last cruise near 8S) - unless close to the Namib. I also would separate into transported dust (AE <0.5, AOD >0.15), maritime background (AOD <0.15),continental transport (AE >1, AOD >0.15) and mixed (AE 0.5 to 1.0, AOD >0.15) unless close to the Namib. For a clearer presentation I would compare the different reference data in one plot and MIC vs satellite and CAMS in another plot*

- We applied your suggested thresholds and wording of the aerosol classification and updated all figures and tables and implemented the following sentences to Section 3.1:
  "*The pair of AOD and AE values is checked against empirical thresholds to identify the dominant aerosol type of the current situation as being one of maritime background (AOD < 0.15), mineral dust transport (AE < 0.5, AOD > 0.15), continental transport (AE > 1, AOD > 0.15), or mixed (0.5 < AE < 1, AOD > 0.15) type. It*

*should be noted, that all categories are expected to cover mixed aerosol types to some extent. Therefore, the mixed category consists of a mixture of aerosol without a dominant type."*

- *figure 2 redo with the suggested typing of Figure 1 comments*

- Done.

- *figure 3 /4 something for the Appendix (why are counts of COMB sometimes higher than for GUVisE in figure 4?)*

- Counts for individual bins may differ between GUVisE and COMB occasionally when AOD values are close to the bin borders, since COMB consists of the means values between GUVisE and MIC. But as we looked at this detail, we found that the datapoint numbers presented in the Figure 5 and table 5 are lower as shown in Table 2. We found a bug in our code, which omits all datapoints from GUVisE which are not collocated to MIC. After the fix, we calculated Figure5 and table 5 again with the correct number of collocated datapoints.

- *figure 5 nice detail...but just because SEVIRI has likely (and missed dust event related) biases at higher AOD, while most cases (at low AOD) are larger than MODIS you cannot conclude that the SEVIRI overestimate (compared to MIC) is lower*

- We did not draw this conclusion from figure 5 but from figure A3, where SEVIRI AOD (630nm) is slightly lower in all cases. We admit, that this is easily overlooked, since the paragraph mainly discusses the findings from figure 5. We added "*(see Fig. A3 and Table 5)*" to the sentence L436-L438 for clarification.

- *figure 6 seeing this plot, who wants to use the Angstrom parameter from satellite remote sensing and even modeling? SEVIRI seems to use a very simple Aerosol model without allowing for large and small Angstrom parameters. Similarly CAMS*

*is constrained by composition, which does not allow for large Angstrom parameters. On the other hand very high Angstrom parameters with MICROTOPS may be associated with low AOD. Thus I recommend to show Figure 6 only for MIC data, when AOD at 870 is at least 0.1.*
*Actually I call a redo of Figure 6 via a Figure 5-type plot where Angstrom data are compared as a function of AOD bins.*

- We have restricted the comparison to AOD(870nm)>0.05, as you have recommended before. But we opt to stick with the scatter type of the figure, since it nicely shows the limitations of aerosol products in terms of usage of simple models or compsition restrictions as you have analysed. Nevertheless, Figure 1 (see last page) shows the AE comparison in a Figure5 type plot to compare AE as a function of AOD bins. Apart from conclusions already drawn from Figures 7/8 and 9 in the manuscript we see no improvement choosing this representation style.

• *figure 7/8/11 I would redo the aerosol types – see comments to Figure 1 (I assume all component samples fit within circles)*

- Done.

• *figure 9 interesting aspect*

- Thanks.

• *figure 10 nice*

- Thanks.

*resource*

*I placed relevant monthly 1x1 lat/lon gridded data of the MACv2 aerosol climatology on ftp in ascii and netcdf (for details on file naming look at README) ftp:// ftp-projects.*

*zmaw.de/ aerocom/ climatology/ MACv2_2018/ 550nm/ for_tropos/ there you find expected monthly average for the 450nm/850nm Angstrom parameter and aerosol single scattering properties at 450, 550, 650 and 850nm ...maybe this helps as general reference*

Within the scope of the current study, we focus on the instantaneous agreement of satellite products with the shipborne observations. Hence, we did not find a direct application of these data for this paper at this stage. Nevertheless, we thank You for providing these datasets, as in the future, we are planning to investigate the radiative effects of aerosols based on the shipborne measurements. A comparison with the monthly climatology and the additional information provided on the single scattering albedo might be interesting contributions of the MACv2 climatology to these plans. We have therefore added the following sentences to the Outlook:

- L647: "*It also offers the chance of an evaluation of the*" => "*to evaluate the direct ...*"

- and add => "*and these observations could contribute towards improving climatological estimates of the aerosol radiative effect (e.g. Kinne2019)*"

Please also note the supplement to this comment:
https://www.atmos-meas-tech-discuss.net/amt-2019-321/amt-2019-321-AC3-supplement.pdf

—————————————————

**Fig. 1.** A plot like Figure 5 but with AE difference and aerosol type

**Supplement:**

[revised manuscript text omitted]
 | $C_{PS83}$ | $k_{PS83}$ | $C_{PS95}$ | $k_{PS95}$ | $C_{PS98}$ | $k_{PS98}$ | $C_{PS102}$ | $k_{PS102}$ | $C_{PS113}$ | $k_{PS113}$ | $S$ |
|---|---|---|---|---|---|---|---|---|---|---|---|
| nm | - | $\frac{\mathrm{V\,m^2\,nm}}{\mathrm{W}}$ | - | $\frac{\mathrm{V\,m^2\,nm}}{\mathrm{W}}$ | - | $\frac{\mathrm{V\,m^2\,nm}}{\mathrm{W}}$ | - | $\frac{\mathrm{V\,m^2\,nm}}{\mathrm{W}}$ | - | $\frac{\mathrm{V\,m^2\,nm}}{\mathrm{W}}$ | - |
| 380 | 1.02 | 1.42 | 0.97 | 1.44 | 1.03 | 1.32 | 0.96 | 1.31 | 0.99 | 1.29 | 1.12 |
| 440 | 0.98 | 7.52 | 0.93 | 7.48 | 0.96 | 6.90 | 0.92 | 6.87 | 0.95 | 6.79 | 1.13 |
| 500 | 1.02 | 20.91 | 0.99 | 21.33 | 1.03 | 19.55 | 0.97 | 19.63 | 1.00 | 19.83 | 1.14 |

[revised manuscript text omitted]